# Spatial Transcriptomics to define transcriptional patterns of zonation and structural components in the mouse liver

Franziska Hildebrandt [1✉], Alma Andersson[2], Sami Saarenpää[2,7], Ludvig Larsson [2,7], Noémi Van Hul [3,7], Sachie Kanatani[1], Jan Masek [3,4], Ewa Ellis [5], Antonio Barragan [1], Annelie Mollbrink[2], Emma R. Andersson [3], Joakim Lundeberg [2] & Johan Ankarklev [1,6✉]

Reconstruction of heterogeneity through single cell transcriptional profiling has greatly advanced our understanding of the spatial liver transcriptome in recent years. However, global transcriptional differences across lobular units remain elusive in physical space. Here, we apply Spatial Transcriptomics to perform transcriptomic analysis across sectioned liver tissue. We confirm that the heterogeneity in this complex tissue is predominantly determined by lobular zonation. By introducing novel computational approaches, we enable transcriptional gradient measurements between tissue structures, including several lobules in a variety of orientations. Further, our data suggests the presence of previously transcriptionally uncharacterized structures within liver tissue, contributing to the overall spatial heterogeneity of the organ. This study demonstrates how comprehensive spatial transcriptomic technologies can be used to delineate extensive spatial gene expression patterns in the liver, indicating its future impact for studies of liver function, development and regeneration as well as its potential in pre-clinical and clinical pathology.

[1] Department of Molecular Biosciences, the Wenner-Gren Institute, Stockholm University, Svante Arrhenius Väg 20C, SE-106 91 Stockholm, Sweden. [2] Science for Life Laboratory, Department of Gene Technology, KTH Royal Institute of Technology, Tomtebodavägen 23a, SE-171 65 Solna, Sweden. [3] Department of Cell and Molecular Biology, Karolinska Institutet Stockholm, SE-171 77 Solna, Sweden. [4] Department of Cell Biology, Faculty of Science, Charles University, Viničná 7, 128 00 Prague 2, Czech Republic. [5] Department of Clinical Science, Intervention and Technology (CLINTEC), Karolinska Institutet, 141-86 Stockholm, Sweden. [6] Microbial Single Cell Genomics facility, SciLifeLab, Biomedical Center (BMC) Uppsala University, SE-751 23 Uppsala, Sweden. [7] These authors contributed equally: Sami Saarenpää, Ludvig Larsson, Noémi Van Hul. ✉email: franziska.hildebrandt@su.se; johan.ankarklev@su.se

The mammalian liver is a pivotal organ for metabolic homeostasis and detoxification. It has been ascribed a central role for the generation, exchange and degradation of essential biomolecules such as ammonium, fatty acids, amino acids, and glucose, as well as the conversion and eradication of various xenobiotic compounds and toxins[1].

In mice, the mature liver can be divided into four major lobes: medial, left (largest), right (bisected) and caudate[2]. Lobes are formed by repetitive units, termed liver lobules. In brief, the lobule, traditionally represented as a hexagon, has a portal vein (PV) at each junction with the neighboring lobules, through which nutrient-rich blood from the intestine enters the liver. Eventually, the nutrient- and oxygen-exhausted blood is drained in the central vein (CV)[3–5].

By volume, the majority of liver resident cells (80%) are parenchymal cells, i.e., hepatocytes[6]. The remaining tissue consists of liver non-parenchymal cells (NPCs), including liver endothelial cells (LECs), liver resident macrophages (Kupffer cells) and other immune cells, hepatic stellate cells (HSCs) and other stromal cells, biliary epithelial cells (cholangiocytes) and cell types of the vasculature (endothelial and smooth muscle cells), which together make up the heterogeneous functional lobular liver environment[7]. Liver resident cells execute distinct functions along the lobular axis based on their proximity to the CV or the PV[8–11]. In mice, this spatial division in metabolic functions, known as zonation, is primarily based on the differential expression profiles along the lobular axis and is classically divided into three zones (zone 1–3). Zone 1 is the region near the portal veins, while zone 2 is defined as the intermediate region between the portal and central veins, and zone 3 is the region near the central veins[11]. More recently these zones between the central and portal vein were divided into 9 concentric layers with layers 1–3 representing the central vein area, mid lobular layers 4–6 and layers 7-9 around the portal vein[12]. Recent findings from single-cell spatial reconstruction approaches suggest that smaller and less abundant NPCs also follow distinct spatial expression profiles based on their position along the lobular axis[13,14]. These reconstruction approaches: (I) provide an intricate image of the metabolic division of labor within the microenvironment of the liver lobule, (II) identify defining factors of zonation based on differentially expressed genes (DEG) along the lobular axis[12,14–16] and (III) represent a fundamental resource for the extensively studied concept of liver zonation[7]. However, all previous studies either performed laser capture microdissection[16] or used perfusion techniques[12,14], ultimately requiring tissue dissociation prior to sequencing, resulting in single-cell resolution but also altering the physiological transcriptional landscape[17–19].

Further, previous studies focused on identifying factors underlying zonation exclusively in the microenvironment of the liver lobule. Investigation of individual liver sections shows that studying the theoretical organization of the repetitive liver lobules is challenging, due to the 3-dimensional organization and the overall complexity of the complete organ. Lobules across the tissue are organized in a highly irregular manner and differ greatly in size and axial orientation. In addition, lobules are situated in varying proximities to the main sources of blood supply, namely the hepatic artery and the portal vein.

An additional layer of complexity in the study of liver tissues is introduced by their organization into several lobes[20,21]. The reason for this partitioning is not yet fully understood, however, certain functional differences of the lobes have been suggested[22–24]. Gene expression profiles may also vary between regions, defined by their distance to other lobes. Therefore, differential gene expression patterns among liver cells independent of the organization in individual lobules and in the extended

tissue context are poorly studied and vital for our full understanding of liver function in homeostasis and disease.

Spatial Transcriptomics (ST) enables high-resolution assessment of spatial gene expression across tissue sections, overcoming the limitations associated with tissue dissociation[17–19]. Hence, the generation of Spatial Transcriptomics data from liver sections in their bona fide tissue context, together with pre-existing knowledge of liver zonation enables the spatial annotation of structures consisting of small mixtures of cells in the liver microenvironment (lobule) and liver macroenvironment (tissue section). Moreover, performing ST across liver tissue sections has the capacity to reveal novel structures, which may be lost when using protocols that do not allow analysis in a spatial context — structures that may play crucial roles for the overall architecture of the liver.

Here, we perform ST on healthy, female mouse liver tissue sections, assessing spatial factors contributing to spatial liver heterogeneity at the transcriptional level. By designing and implementing a variety of computational methods, this study aims to resolve the spatial relationships of vascular components involved in liver zonation and explore previously uncharacterized structures based on their transcriptional profile and in their original tissue context. Our results support the concept that zonation represents a prominent factor contributing to spatial heterogeneity. Computationally tracing the expression levels of transcriptional markers linked to zonation along the lobular axis allows us to study zonation gradients in physical space, and to infer the identity of vascular structures based on their neighborhood expression profiles. We anticipate that our results from ST complement previous findings of different structures and cell types constituting the overall transcriptional landscape of liver tissue and enhance our current understanding of liver tissue organization.

## Results

**Unsupervised clustering defines spatial distribution of expression across liver tissues.** We used a total of 8 sections of wild type adult, female mouse livers from the caudate and right liver lobe for histological staining, library preparation and sequencing. After mapping, filtering, annotation and normalization of raw sequencing reads (Methods) we obtained curated expression data consisting of 19,017 genes across 4,863 individual capture locations (spots) on the ST arrays (summarized over all sections) and subjected the data to downstream computational analysis. Only spots under the tissue sections were considered for analysis and visualization (Fig. 1a). Each spot is covered by a small mixture of liver cells, not all necessarily of the same cell type. For a select set of cell types, we used immunofluorescence staining to estimate the number of cells present in a subset of projected spot areas in liver cryosections. We performed stainings for nuclei (Hoechst), hepatocytes (HNF4α), Kupffer cells (F4/80), and endothelial cells (CD31). Quantification of Hoechst+ nuclei revealed the range of cell count per spot is 10-60 cells with a mean value of 32.1 ± 8.73 cells per spot, out of which 56.9% ± 15.8% are hepatocytes, 12.7% ± 7.4% are Kupffer cells, and ~30.8% ± 17.0% endothelial cells (Supplementary Fig. 1). Subsequently, we integrated the spatial data of different samples using canonical correlation analysis (CCA) and clustered it in an unsupervised manner using a graph-based approach, which identified 6 clusters, exhibiting uniform distribution of unique transcripts (Fig. 1b, top panel, Methods for details, Supplementary Fig. 2). To put the clusters into context and assess their spatial organization, spots were projected onto the brightfield image of the same tissue section stained with Hematoxylin- and Eosin (H&E).

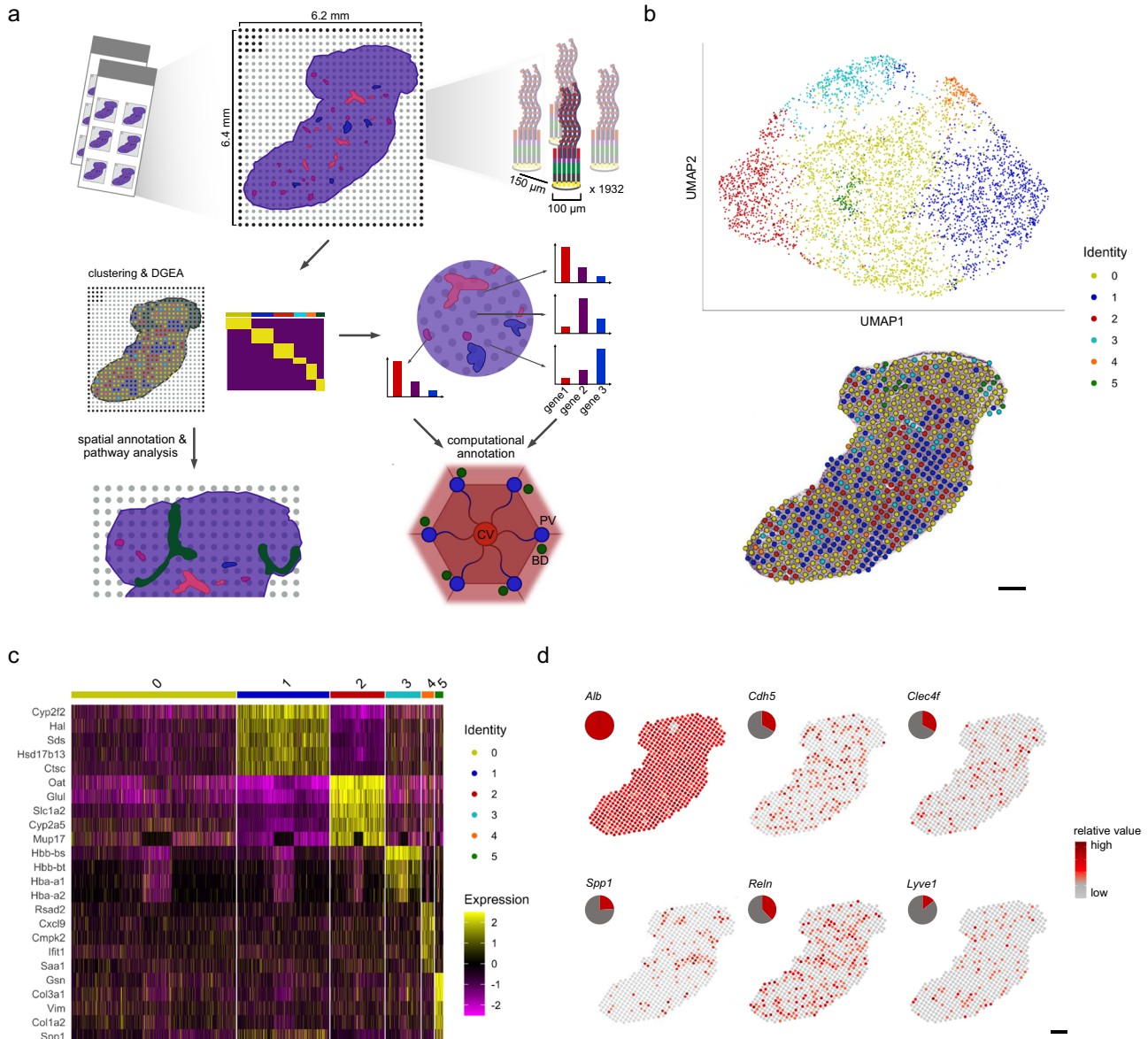

**Fig. 1 Study overview of Spatial Transcriptomics on murine liver. a** Spatial Transcriptomics was performed on a total of 8 murine liver tissue sections. The tissue sections were placed in one of six, 6.2 × 6.4 mm frames on the glass slide ST array. Each frame contains 1932 spots, with >200 M uniquely barcoded mRNA capture probes. The distance between centers of each neighboring spot is 150 μm (200 μm for spots in the same row). Initially, each tissue section was fixed, stained with hematoxylin and eosin (H&E) and followed by imaging. Then, tissue sections were permeabilized, followed by mRNA capture, tissue removal and sequencing. Thereafter, the count data was subjected to cluster- and differential gene expression analysis (DGEA). The results of the clustering and DGEA were further analyzed and spatially annotated at the global tissue context and down to the lobular level. For new spatial annotations, pathway analysis was performed. Liver lobules are classically described by a central vein (CV, red) surrounded by 6 portal nodes (PV, blue) with neighboring bile-ducts (BD, green). For lobular spatial annotations, clusters have been computationally annotated by comparing expression levels in a set of genetic markers linked to metabolic zonation along the lobular axis. **b** Canonical correlation analysis (CCA) was performed to integrate data of 8 liver tissue sections, the data was subsequently normalized and subjected to graph-based clustering in which 6 clusters were identified (see Methods). The integrated data was embedded in UMAP space (top) and depicted as an overlay of the spot cluster annotation across the tissue (bottom) (scale bar indicates 500 μm). **c** Heatmap depicting expression values of the five most variable genes for each cluster after subjecting the 6 clusters to DGEA, with the exception of cluster 3, which resulted in only four significantly differentially expressed genes and cluster 0 which did not result in any significantly differentially expressed genes with the given parameters (Methods). **d** Visualization of spatial distribution of reported expression markers of Hepatocytes (*Alb*), liver endothelial cells (*Cdh5*), Kupffer cells (*Clec4f*), Cholangiocytes (*Spp1*), hepatic stellate cells (*Reln*) and lymphatic liver endothelial cells (*Lyve1*) by spots under the tissue. Pie-charts indicate the respective proportion of cell type markers present in spots under the tissue (scale bar indicates 500 μm).

The projection showed a clear spatial segregation between spots belonging to certain clusters. Visual inspection showed that cluster 5 was localized to an exclusive region of the tissue section, while spots belonging to cluster 1 and cluster 2 aligned with the vascular structures in the liver tissue (Fig. 1b, lower panel, Supplementary Fig. 3). To further characterize the identified clusters, we performed differential gene expression analysis (DGEA) between them. In fact, DEGs in cluster 1 support periportal gene expression from previous studies while genes previously associated with pericentral gene expression are enriched in cluster 2, proposing that cluster 1 and cluster 2 denote regions around the portal and central veins,

respectively[12,14–16]. Cluster 3 is enriched for genes associated with hemoglobin, whereas cluster 4 shows enriched expression of genes involved in immune-related processes[25,26]. Cluster 5 displays enrichment for mesenchymal genes[27–29]. Cluster 0 does not show enrichment of any gene at the set log-fold change (logFC) threshold of 0.5 (Fig. 1c, Supplementary dataset 1).

Spots of cluster 3–5 are mainly surrounded by spots of different clusters, while cluster 0–2 form more cohesive groups of spots. Interestingly, spots of cluster 0, 3, and 4 seem to adjoin spots of cluster 0, 1, and 2 in descending order, implicating transcriptional profiles of most clusters are commonly surrounded by periportal rather than pericentral areas (Supplementary Fig. 4). The scattered spatial distribution of cluster 3 across sections can most likely be explained by the fact that the tissue was not perfused prior to freezing and sectioning, allowing us to detect blood cell populations throughout the liver. To assess replicability and the sensitivity of the method to detect the transcripts of different liver cells per spot, we examined the expression of genes, previously reported to be marker genes for common cell types in the liver across spots under the tissue.

In agreement with the histological evaluation of the tissue, non-zero expression of the hepatocyte marker *Alb* (expression value > 0) in 100% of spots indicated a global presence of hepatocytes. For LECs, 1594 out of 4863 spots showed expression of *Cdh5*[30,31] (~33%). Lymphatic liver endothelial cell and liver midlobular endothelial cell-marker *Lyve1*[32–34] showed expression in a smaller fraction of 698 spots (~14%). Kupffer cell-marker *Clec4f*[35–37] showed expression in 1723 spots (~35%) while hepatic stellate cell-marker *Reln*[38] was expressed in 1870 spots (~38%). *Spp1* is a marker for Cholangiocytes[39], expected to only be present in bile ducts, next to portal veins and is expressed in 1165 spots (~24%) (Fig. 1d). These results demonstrate that highly abundant, or bigger cells are widespread, while smaller and rarer cell types are found more scattered across the liver tissue.

While characteristic marker gene expression is a common way to extrapolate the presence of certain cell types, we wanted to include a larger set of genes constituting the expression profile of a specific cell type and compare it to our spatial data. *stereoscope*, presented by Andersson et al.[40] enables cell types from single-cell RNA sequencing (scRNA-seq) data to be mapped spatially onto the tissue, by using a probabilistic model. With *stereoscope*, we were able to spatially map 20 cell types annotated in the Mouse Cell Atlas (MCA)[41] on liver tissue sections (Supplementary Figs. 5–7). Notably, high proportion estimate values are obtained for periportal as well as pericentral hepatocytes in the MCA (Supplementary Figs. 5–7). Pearson correlation values between cell-type proportions across the spots show positive correlation, to be interpreted as spatial co-localization of nonparenchymal cells like LECs, epithelial cells and most immune-cells, as well as stromal cells (Fig. 2a). Interestingly, periportal and pericentral hepatocytes not only exhibit negative correlation, indicating spatial segregation between each other but also with most other cell types (Fig. 2a). A large fraction of spots is assigned to cluster 1 and cluster 2, while these cells only represent a very small fraction of the MCA data. This observed discrepancy implies that a relatively small cell type population identified by scRNA-seq can constitute a large proportion of the spatially profiled cells, illustrating the power of complementing single-cell transcriptome data with spatial gene expression data to thoroughly delineate liver architecture and the transcriptional landscape of liver tissue. Importantly, the spatial distribution of periportal and pericentral cell type proportions overlap with spatial annotations for cluster 1 and cluster 2, respectively (Fig. 2a (top right)). Moreover, Pearson correlations between spots exhibiting high proportions of periportal and pericentral hepatocytes and correlations between spots with portal and central annotations (cluster 1 and cluster 2)

show similar trends, advocating for a reliable integration of cell type annotations from scRNA-seq data and our ST data (Supplementary Fig. 8, Supplementary Tables 1–2).

**Heterogeneous spatial gene expression linked to pericentral and periportal zonation.** Spatial expression of common marker genes of periportal or pericentral zonation, as well as observed periportal and pericentral hepatocyte proportions from single-cell integration across the tissue imply co-localization of cluster 1 and cluster 2 with portal and central veins, respectively. To support this observation, venous structures in our sections were annotated as: a portal vein, central vein, or vein of unknown type (ambiguous). The annotations are based on the presence of bile ducts and portal vein mesenchyme or lack thereof. Comparison of the histological annotations and the corresponding clusters allowed us to annotate cluster 1 as the periportal cluster (PPC) and cluster 2 as the pericentral cluster (PCC) (Fig. 2b).

Pearson correlations between genes enriched in the PPC and genes enriched in the PCC show a negative trend, interpreted as spatial segregation (Fig. 2c, Supplementary Dataset 2). PCC genes exhibit positive correlations to all other marker genes present in the PCC, and PPC marker genes show positive correlations to other PPC markers, interpreted as spatial correlation (Fig. 2c). None or lower correlations can be observed between PPC or PCC marker genes and the remaining 4 clusters (cluster 0 and cluster 3-5) (Supplementary Fig, 9, Supplementary Dataset 2). The spatial gene expression's heterogeneity with respect to central and portal vein proximity is corroborated by the spatial autocorrelation of known marker genes (Methods, Supplementary Fig. 10, Supplementary dataset 3).

Visualization of representative pericentral (*Glul*) and periportal (*Sds*) marker expression in the UMAP embedding further demonstrate highest expression values of *Glul* or *Sds* in the pericentral or periportal cluster, respectively. When inspecting the expression of *Glul* and *Sds* in their spatial context, these genes show the highest expression in areas annotated as central or portal veins. In addition, no expression of *Sds* can be found in areas of elevated *Glul* expression and vice versa, indicating expression of genes present in the pericentral cluster 1 and periportal cluster 2 are spatially distinct and negatively correlated with each other (Fig. 2d). Based on these observations, we further investigated the zonation of reported marker genes in the context of reported immune zonation[42]. To this end, we investigated DEGs associated with immune system processes (GO:0002376) and found more genes with periportal than pericentral zonation (Supplementary Fig. 11).

**Transcriptional profiling of pericentral and periportal marker genes across tissue space enable computational annotation of liver veins.** To further investigate zonation in physical space, we first superimposed the spots under the tissue showing expression for two representative markers of central veins (*Glul*, *Cyp2e1*) and portal veins (*Sds*, *Cyp2f2*), onto histologically annotated veins (Fig. 3a). The gene *Glul* encodes the protein glutamine synthetase, the main enzyme in glutamine synthesis[15], while serine dehydratase (*Sds*) is a key factor for gluconeogenesis[43]. *Cyp2e1* and *Cyp2f2* both belong to the cytochrome P450 family involved in xenobiotic metabolism[44–46]. Pericentral expression of *Glul* is restricted to spots in very close proximity to the annotated central veins, while *Cyp2e1* is more evenly distributed across spots. Neither *Cyp2e1* nor *Glul* are detectable near annotated portal veins. The opposite pattern is observed for the expression of *Sds* and *Cyp2f2* around the portal vein. Including all marker genes of the PCC and the PPC and creating module scores (Methods) of expression of all DEGs of the respective cluster in the spots under

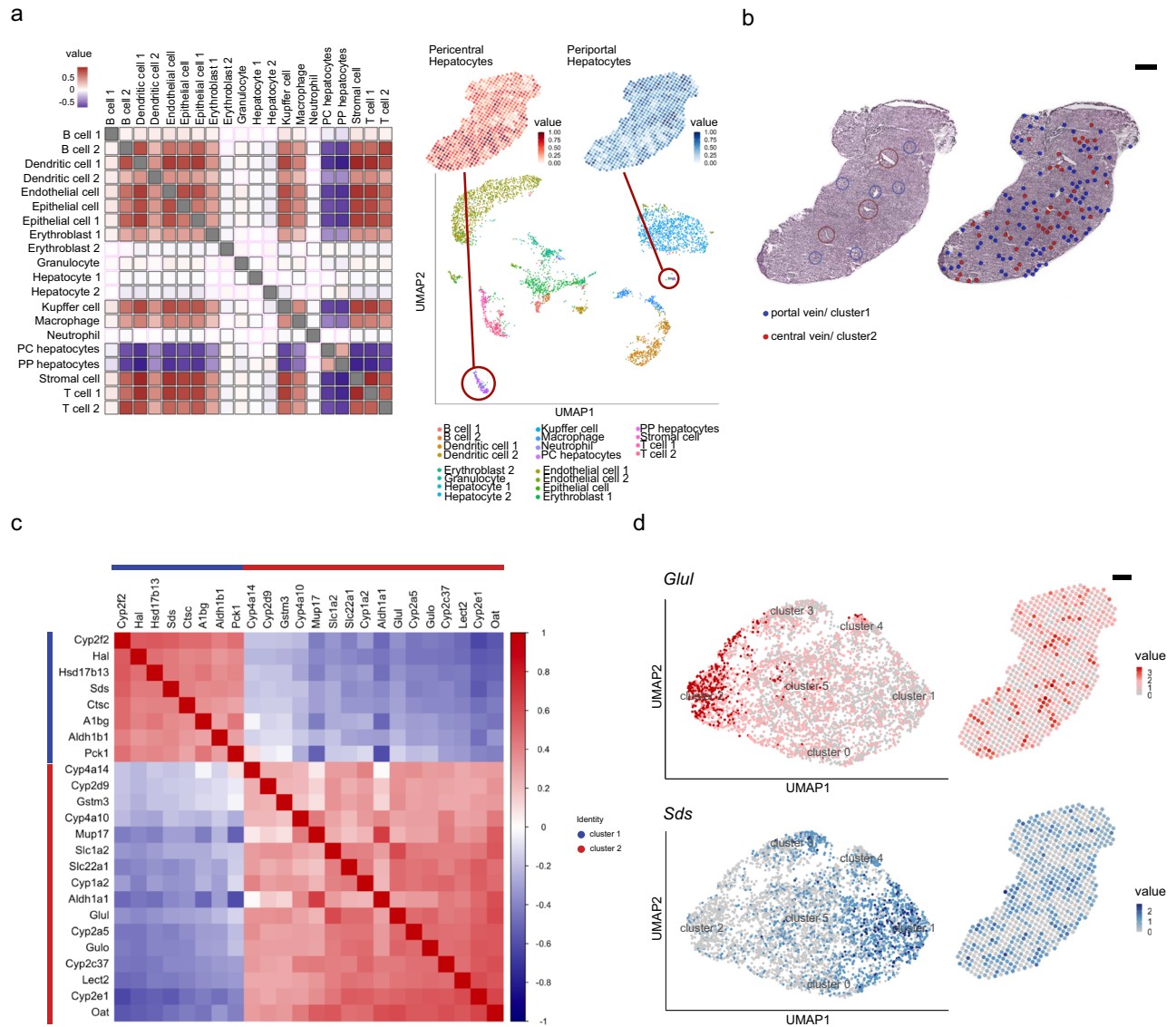

**Fig. 2 Clustering, spatial annotation and computational validation using established scRNA-seq data. a** Visualization of cell type co-localization by Pearson correlations (left). Positive correlation values indicate spatial co-localization of cell types while negative values represent spatial segregation. Non-significant correlations are highlighted with magenta borders. UMAP embedding of single cell data of the Mouse Cell Atlas (MCA)[41] grouped by annotated cell types (bottom right). Numeration behind the cell types represent annotation of MCA data (B cell-1: Fcmr high, -2: Jchain high, Dendritic cell-1: Cst3 high, -2: Siglec high, Epithelial cell-1: Spp1 high, -2: /, Erythroblast-1: Hbb-bs high, -2: Hbb-bt high, Hepatocyte-1: Fabp1 high, -2: mt-Nd4 high, T cell-1: Gzma high, -2: Trbcs2 high). Encircled clusters in the plot refer to pericentral or periportal hepatocytes of MCA data. Quantile scales of cell-proportions annotated as pericentral and periportal hepatocytes (Methods) are mapped on Spatial Transcriptomics spot data (top right). **b** Visualization of spots representing gene expression profiles of cluster 1 (portal vein, blue) and cluster 2 (central vein, red) on H&E stained tissue (right), compared with visual histology annotations of central- (red circles) and portal- (blue circles) veins (left) (scale bar indicates 500 μm). **c** Pearson correlations of genes expressed in cluster 1 and 2 ordered by their first principal component (Methods). Genes with high expression in the pericentral cluster (cluster 2) show negative correlation with genes highly expressed in the periportal cluster (cluster 1) and vice versa. Genes present within cluster 1 or cluster 2 exhibit positive correlation with genes in the same cluster. **d** Projection of selected marker genes for central venous expression (*Glul*, top) and periportal expression (*Sds*, bottom) in UMAP space and spots under the tissue (scale bar indicates 500 μm).

the tissue, we visualized the common expression gradient along the lobular axis (Fig. 3a, Supplementary Figs. 12–13). These described genes are associated with a small subset of liver metabolic processes. However, we were also able to confirm that a general trend for the enrichment of additional known zonated metabolic pathways[7,47] can be observed between the PPC and PCC (Supplementary Fig. 14).

Next, we wanted to assess whether gene expression was influenced by spatial proximity to the different vein types, as would be expected based on the study by Halpern et al.,

describing expression gradients over a total of 9 layers along the lobular axis[12]. For this purpose, we generated what will be referred to as *expression by distance* plots, which portray the normalized gene expression as a function of the distance to a given vein type. To construct these plots, for each spot and gene, we pair the observed expression value with the distance from the spot's center to the nearest vein border. Finally, to better capture the relationship between distance and expression, we smooth our observations with the *loess* method (Methods). Expression by distance plots were compiled for a selected set of five periportal

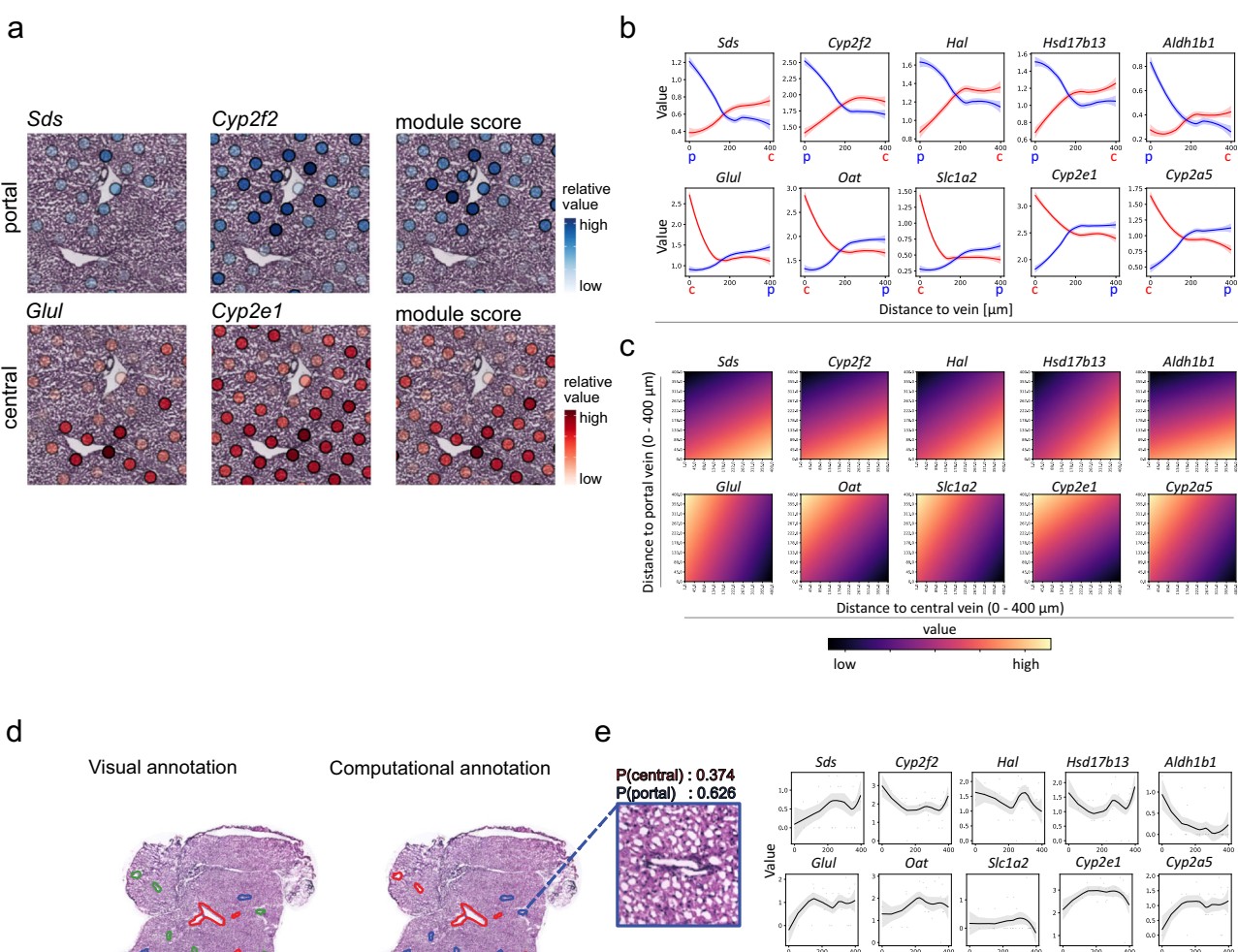

**Fig. 3 Expression gradient along the lobular axis and computational annotation of liver vein types. a** Enlarged view of a superimposed visualization of *Sds*, *Cyp2f2* expression in the portal vein module, consisting of selected DEGs of cluster one (Supplementary Dataset 1), all with high values around the histological annotation of a portal vein (top). Expression of *Glul*, *Cyp2e1* as representative marker-genes of the central vein module expression (Supplementary Dataset 1), consisting of DEGs of cluster 2 with high values around the histological annotation of a central vein (bottom). **b** Visualization of the average expression by distance to vein-type measured within 400 μm from the vein. The top row shows expression by distance of portal markers *Sds, Cyp2f2, Hal, Hsd17b13* and *Aldh1b1* to portal veins in blue and central veins in red, while the bottom row shows distances of central vein markers *Glul, Oat, Slc1a2, Cyp2e1*, and *Cyp2a5* to portal veins in blue and central veins in red (top panel). Red and blue ribbons around the fitted line represent the standard error of the gene expression within spots along the distance to respective vein type. **c** Visualization of influence of distance to both vein types on expression by *bivariate expression by distance* plots (Methods). Gene expression values are depicted in a gradient from low (dark) to high values (light). The distance of each gene to central veins between 0 and 400 μm is represented on the x-axis. Simultaneously, distances to portal veins for the same distance are depicted on the y-axis for each gene. High values in the bottom right corner indicate gene expression is predominantly observed close to portal veins and far from central veins, while high values in the upper left corner indicate the reverse observation (below graphs). **d** Visual histological annotations (left) of central (red) and portal (blue) veins, including ambiguous visual annotations (green), compared with computational prediction, using the 10 marker genes from **b** (right). The classification of vein types is based on a weighted (by distance) average expression of the genes' expression profiles in the neighborhood of each vein. In addition, the spatial expression data of spots neighboring uncertain morphological vascular annotations (green) can be used to predict periportal or pericentral vein types in the cases where visual annotations are ambiguous. **e** Expression by distance of portal—(top panel) and central—(bottom panel) markers. Probabilities for each class (central and portal) can be extracted from the logistic regression model, here given as P(central) or P(portal) (scale bar indicates 500 μm). Grey ribbons around the fitted line represent the standard error of the gene expression within spots along the distance to each of the depicted veins.

and pericentral marker genes with the highest positive logFC in the PPC and PCC (Supplementary dataset 1, Supplementary Fig. 15). Upon inspection of the plots, a clear dependency between distance and expression becomes apparent. Portal markers exhibit a gradual decline upon the increased distance from a portal vein. For central veins, certain genes (e.g., *Glul*, *Slc1a2*, *and Oat*) show a steep decrease in expression as the distance from a central vein increases, while others (e.g., *Cyp2e1* and *Cyp2a5*) display a more gradual decline (Fig. 3b). These results are in agreement with the observed expression gradients in spatially reconstructed layers in Halpern et al.[12], which are orthogonally validated by smFISH (Supplementary Figs. 16–17). Additionally, we transferred the mapped proportion values of the annotated periportal- and pericentral hepatocytes in the MCA single-cell data along the lobular axis, and observed the same inverse relationship with the distance of their associated vein types as for the marker genes (Supplementary Fig. 18).

The aforementioned expression by distance plots illustrate how the distance to either the portal or central vein influences the gene expression, and can thus be considered a univariate model. To also, simultaneously, assess how gene expression varies with the relative position to both vein types, we employed a bivariate linear model (Methods). Plots illustrating the results from fitting the bivariate model to the data will be referred to as bivariate expression by distance plots. Additionally, the parametric bivariate model facilitates evaluation of the relevance of each distance measurement (portal or central vein distance) for gene expression (Methods). For portal and central vein markers we observed a gradient from high expression at the portal or central vein border to low expression at the respective opposite vein (Fig. 3c). Our analysis revealed that the distance to both veins is equally influential for the expression of central and portal markers, as both univariate models are outperformed by the bivariate model (Supplementary Dataset 4). To further explore other processes determined by the zonation, we investigated the relevance of the central and portal distance for the expression of metabolic pathway markers[7,47] (Methods). We were able to determine genes for which distances to both veins or the distance to only central or portal veins were influential. However, we also found genes for which the distance to either vein represented an explanatory variable for expression (Supplementary Fig. 19, Supplementary Dataset 4).

The observed differences in gene expression along the lobular axis of different central and portal vein markers agree with concepts of dynamically expressed genes along the lobular axis of the gradient type and stable gene expression of genes of the compartment type directly at the central or portal vein borders[9,48,49]. Spatially stable expression of compartment type genes is exemplified by the genes *Glul*, and *Slc1a2*, important for glutamate transport at central veins[15]. Expression of *Sds*, and the histidine ammonia lyase (*Hal*), involved in ammonium production are distinctive for stable gene expression at portal veins[50]. The dynamic expression of gradient type genes is illustrated by *Cyp2e1* (pericentral) and *Cyp2f2* (periportal).

Given the strong association between the DEGs in the PPC and PCC, as well as the convincing demonstration of co-localization with histologically annotated central and portal veins, we aimed to explore whether veins could be computationally annotated solely based on gene expression (Fig. 3d). Computational annotation of veins as a complement to manual annotation is of relevance for multiple reasons. First, manual annotations sometimes prove to be difficult when only histological images of suboptimal quality or without immunohistological staining are available. Second, annotation is a labor-intensive process that requires thorough histology training. Thus, a computational model not only provides the possibility to validate the manual vein annotations but also to predict the type of unannotated veins based on their surrounding gene expression profiles. The model constructed in this study (Methods) corresponds convincingly to manually annotated central and portal veins based on the expression profile of their respective neighborhood across all sections from different biological origins (caudate and right liver lobe) (Supplementary Fig. 20). Based on the confident evidence of overlapping visual- and computational vein annotation, we continued to computationally annotate veins with ambiguous identity. With our method, we could assign the 72 ambiguous veins as being either portal-or central veins, only relying on the neighborhood expression profiles of 5 central and 5 portal vein markers (Fig. 3e, Supplementary Dataset 5). For proximate tissue sections of selected samples, we also show that the majority of computational predictions is supported by immunofluorescence staining for the respective central and portal protein markers GS and SOX9, serving as an orthogonal validation of our results (Supplementary Figs. 21–22). The prediction of vein types based on the spatial expression profile of surrounding spots demonstrates the potential to use spatial gene expression data for a variety of annotation-based applications.

**Exploration of components contributing to spatial heterogeneity across liver tissues**. Spots assigned to cluster 5 on the H&E images demonstrate exclusive spatial organization in one or two distinct regions across the tissue (Fig. 4a, Supplementary Fig. 23). Therefore, we asked how this cluster fits into the spatial liver organization based on its expression profile. Additionally, we wanted to assess whether the spatial organization of this cluster can give indications regarding the function of the underlying distinct structure in the tissue, which is characterized by morphologies resembling potential tissue partitioning.

DGEA identified *Gsn*, *Col1a2*, *Col1a3*, and *Vim*, as highly upregulated marker genes of cluster 5 (Supplementary Dataset 1). Spots not belonging to cluster 5 show no expression or low expression of these genes (Fig. 4b, Supplementary Fig. 24). In fact, pathway analysis of the cluster 5 marker genes demonstrates the strongest enrichment of genes belonging to the process "collagen and fibril organization" (Fig. 4c, Supplementary Dataset 6). Collagen fibrils have been reported to be the main component of the irregular connective tissue composing the Glisson's capsule in several animals, including rodents[51], giving first indications for a structural function of cluster 5. In addition, processes contributing to structural formation and development, such as "extracellular matrix organization" and "extracellular structure organization" and pathways related to innate immunity, namely "response to cytokine" and "antigen processing and presentation of peptide or polysaccharide antigen via MHC class II" show enrichment within cluster 5 (Fig. 4c, Supplementary Dataset 6).

Module scores of marker genes involved in "collagen and fibril organization" are highest in spots of cluster 5 and in their direct proximity in the tissue and show low scores across the remaining tissue. This is supported by additional unsupervised analysis using Spearman correlations (Methods), exhibiting negative correlation between increasing distance to cluster 5 and expression of *Gsn*, *Col1a2*, *Col1a3*, and *Vim* (Supplementary Fig. 25).

In contrast, module scores of marker genes involved in the response to cytokines (*H2-Eb1*, *Timp2*, *Timp3*, *H2-Aa*, *Cd74*, *H2Ab1*, *Spp1*, *Gsn*, *Col3a1*, and *Vim*) are more evenly distributed across the tissue (Fig. 4d, Supplementary Fig. 26). This result supports the higher significance of processes involved in structural formation and development of the tissue area at and around cluster 5. Moreover, we compared marker genes of cluster 5 with marker genes of annotated cell types of scRNA-Seq data

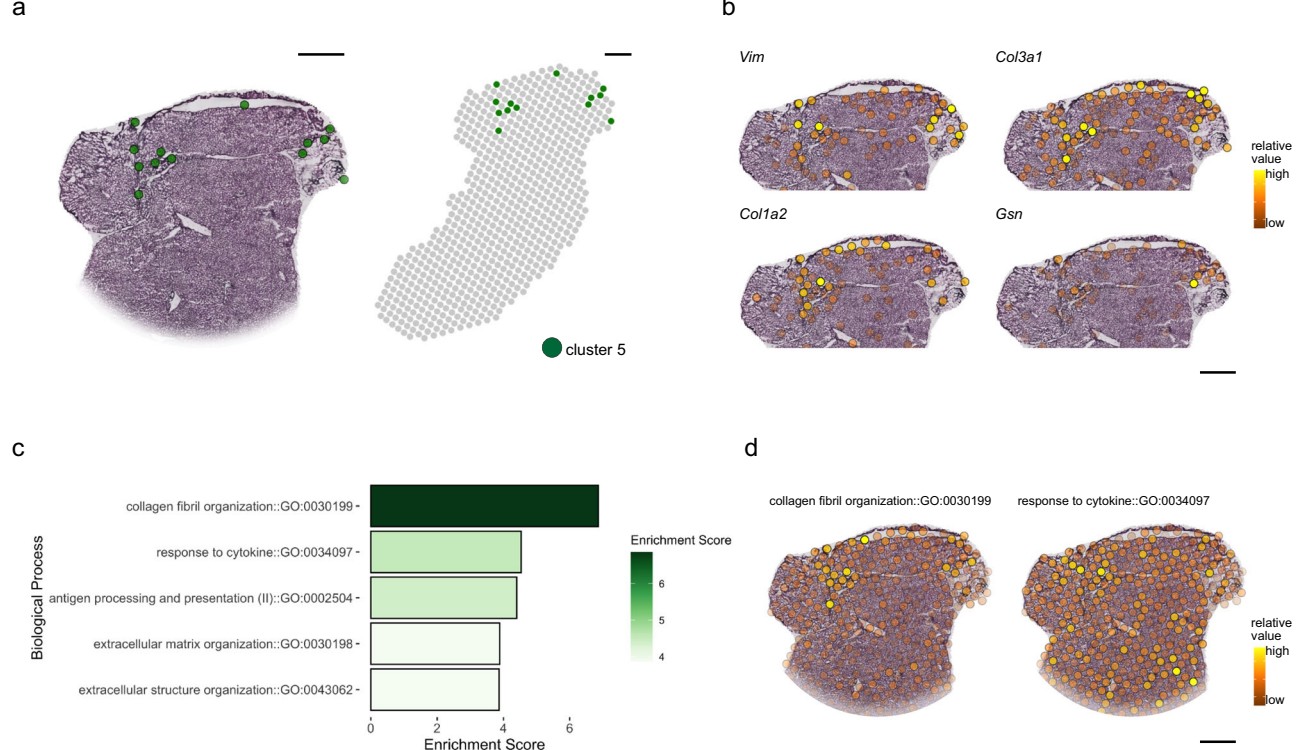

**Fig. 4 Identification of liver tissue regions with unique transcriptional patterns. a** Projection of spots including transcriptional patterns of cluster 5 in the UMAP and tissue (Fig. 1b), on the respective part of a histological section of the caudate lobe (left) and spot location in the entire tissue section (right). **b** Visualization of *Vim*, *Col3a1*, *Col1a2* and *Gsn* expression in spots of the same tissue section as in **a**. **c** Gene-ontology (GO:BP) enrichment for marker genes present in cluster 5. The Enrichment is given as the negative log10 algorithm of the adjusted p-value (g:SCS correction, Methods) of the differentially expressed marker genes in cluster 5. **d** Module scores of cluster 5 marker genes (Methods) associated with the two biological processes with the highest enrichment scores: "collagen fibril organization" and "response to cytokine" are visualized on spots across the tissue.

and found high overlap with mesenchymal cell types (fibroblasts, HSCs and vascular smooth muscle cells (VSMCs) (Supplementary Fig. 27).

Taken together, unsupervised correlation analysis of cluster 5 markers, histological morphology in the respective tissue area and high overlap with mesenchymal cell markers advocates for the spatial organization of cluster 5, independent of liver zonation.

## Discussion

Applying Spatial Transcriptomics to the mammalian liver represents a compelling venue to explore its transcriptional and functional heterogeneity while also complementing the previous data[7,18]. Recent scRNA-seq studies including integration of spatial information by reconstruction provide high-resolution information of single-cell transcriptomes[12,15,16], but the spatial organization of these cells within the same tissue is lost due to tissue dissociation, which additionally increases the risk of undesirable transcriptional changes[13,19]. In contrast, ST preserves the spatial information of the gene expression in its true tissue context, thus complementing single-cell transcriptomics approaches. The emerging possibilities of combining Spatial Transcriptomics data with single-cell and other omics data of the same tissue offer unprecedented levels of insight into the biology of the tissue[40,52].

Here, we estimated cell type information in the spatial data in two different ways. First, we assessed the expression of characteristic marker genes within a wide range of expression levels and second, we deconvolved gene expression profiles of the mixed cells in spots using *stereoscope*.

A recent study suggests predominant localisation of Kupffer cells in the periportal area of the liver lobule and neutrophil recruitment upon bacterial infection[42]. While our data does not indicate elevated Kupffer cell proportions in the periportal cluster compared to the remaining clusters, we found more genes related to immune system processes with periportal enrichment in comparison to the pericentral zone providing initial support for implications of previously proposed immune zonation[42]. The liver is constantly exposed to toxic and microbial threats from the periportal blood, requiring an efficient balance between the immune hyporesponsiveness and effective clearance of pathogens[53]. Therefore, it will be of high interest to perform Spatial Transcriptomics to study the effect of infection and inflammation on immune zonation.

Next, scRNA-seq integration shows that proportion values are highest for pericentral and periportal hepatocytes. The observed discrepancies between our and the MCA data may result from the different technical limitations that scRNA-seq and spatial data generation face, emphasizing the current limits of scRNA-seq data integration. For instance, transcriptionally highly active or physically large cells might mask cell types with moderate to low transcriptional levels in ST data. Therefore, technical and computational advances to enhance resolution may benefit transcriptional profiling of rare cell types within a tissue. Nevertheless, comparisons to scRNA-seq data confirm general trends observed in our ST data, highlighting the importance of combining ST with scRNA-seq data.

We annotated two clusters with anti-correlating spatial distributions and characteristic marker gene expression that align well with the visually annotated portal or central veins in the H&E image as periportal (PPC) and pericentral (PCC) clusters.

Overall, the spatial data generated in this study supports the hypothesis that the main source of spatial heterogeneity across liver tissue are transcriptional differences between zones along the lobular axis between the portal and central veins[12,14,15].

Moreover, the expression of central markers *Glul* and *Slc1a2* and portal markers *Sds* and *Hal* illustrate compartmentalization of gene expression for genes performing opposing tasks like glutamine and ammonium synthesis, necessary to prevent futile cycles[54]. We further affirm the established relevance of zonation of multiple metabolic pathways along the porto-central axis[5,7,9,11,12,14–16,55,56], by tracing expression gradients from outer vein borders and across physical space.

In addition, we investigate the relationships between the marker gene expression of both portal and central veins simultaneously. Marker gene expression across annotated veins in the tissue is insufficient to confirm the proposed schematic organization of the liver lobe of one central vein surrounded by six portal nodes. Nevertheless, the results illustrate the overall relationships of zonation markers, including metabolic pathway and immune markers with central and portal veins across the tissue, suggesting whether the distances to central and/or portal veins represent stronger explanatory variables for gene expression independent of the schematic organization of lobules in physical space.

Based on the convincing evidence for robust expression profiles of central and portal veins across the tissue we were able to generate a computational model to predict the vein type in cases where visual annotations were ambiguous, based on the expression profiles of neighboring spots. This computational model demonstrates the potential of ST to support morphological annotations, providing probability values for the certainty of the computational annotation of morphological structures at their natural tissue location by transcriptional profiling. We anticipate that this method will provide a multitude of applications in future spatial transcriptomics studies, e.g., linked to pathology or infection.

Cluster 5 consists of a small number of spots with distinct spatial localization, which exhibit expression of mesenchymal cell-marker genes[14,29] and are associated with "collagen fibril organization" pathways. We propose that cluster 5 might represent parts of the Glisson's capsule, composed of collagen fibrils together with its underlying mesothelium, representing the connective tissue encapsulating the liver and regions with thicker, hilar periportal mesenchyme. The capsule preserves the structural integrity of the loosely constructed liver and enables the division into lobes[51].

The mesenchymal cell-marker *Vim* is reported to maintain mesenchymal cell structure and serves as an indicator for cell proliferative activity in liver cells[27,57]. *Gsn* encodes the actin-binding protein gelsolin which has an anti-apoptotic role in the liver[58]. Anti-apoptotic effects and enrichment of connective tissue, possibly from the Glisson's capsule, might be crucial in fragile positions of the organ or close to connection positions of liver lobes. The two additional pathways involved in the structural integrity in cluster 5, namely "extracellular matrix organization" and "extracellular structure organization", further advocate for a structural function of cells in this cluster. Enrichment of gene ontologies associated with response to cytokines are observed in, but not limited to, cluster 5; hence they are contributing rather than defining components of the cluster's expression profile and function of the structure.

The functional basis for the division of the liver into multiple lobes and the establishment and maintenance of the organ's structural integrity are yet to be fully understood[20,59]. Considering the sample size used in this study, we can provide initial indications rather than general claims of the function of this

proposed structure. In addition to capturing and supporting previously observed trends of tissue heterogeneity in the mammalian liver, our study serves as a valuable resource to further investigate the spatial expression of structural components and gene candidates involved in the aforementioned processes.

In summary, this study presents an effective approach to investigate the transcriptional landscape of liver tissue through ST and innovative computational approaches. We designed and implemented computational tools allowing physical distance measurements and predictions of vein-type identity. In addition, we observe the presence of transcriptionally distinct structures in liver tissues, which have not been characterized by previous transcriptomic studies, likely due to the rarity of cells contributing to these structures.

With expected future advances in the spatial genomics field, the increased resolution will promote detailed investigations of rare cell types in tissue space. This study constitutes a compelling initial exploration of the benefits that spatial transcriptomics provides for studies of the liver and we consider it a valuable data resource for the hepatology field. We further anticipate that ST will be highly beneficial for future studies addressing liver development, sexual dimorphisms of liver zonation, immunity and general pathology in the mammalian liver, including humans.

## Methods

**Ethical statement**. The Regional Animal Research Ethical Board, Stockholm, Sweden, approved experimental procedures and protocols involving extraction of organs from mice (N135/15, N78/16 and 9707-2018), following proceedings described in EU legislation (Council Directive 2010/63/EU). The mice used in this study were kept at a 12 h night/day cycle and 22 °C ambient temperature, with free access to food and water under specific pathogen-free conditions at the Experimental Core Facility, Stockholm University and were euthanized between 8 and 12 weeks of age.

**Total RNA extraction**. To test for the RNA quality of the tissue for further downstream analysis, the tissue was sectioned and up to eight 10 μm sections were placed in Lysing Matrix D tubes (MPBiomedicals, cat.no.: 116913050-CF) containing Buffer RLT Plus (Qiagen, cat.no.: 1053393) and ß-Mercaptoethanol (Thermofisher, cat.no.: 31350010) and homogenized in a Fastprep instrument (ThermoSavant). The flowthrough was collected through a gDNA Eliminator column and 250 ul of pure ethanol was added. Total RNA was further extracted using the RNAeasy mini kit (Qiagen, cat.no.: 74104) according to manufacturer's instructions. The RNA integrity number (RIN) for each sample was assessed performing Bioanalyzer High Sensitivity RNA Analysis (Agilent cat.no.: 5067-1535).

**Collection and preparation of liver samples**. Female C57BL/6 mice (Charles River), were euthanized between weeks 8 and 12 and livers were collected, and four lobes were separated. Each lobe was segmented so cryosections would fit on the 6,200 × 6,400 μm areas of the Codelink-activated microscope slides and frozen in −30 °C 2-Methylbutane (Merck, cat.no.: M32631-1L). The frozen liver samples were embedded in cryomolds (10×10 mm, TissueTek) filled with pre-chilled (4 °C) OCT embedding matrix and frozen (CellPath, cat.no.: 00411243). For downstream experiments, the frozen samples were sectioned at 10 μm thickness with a cryostat (Cryostar NX70, ThermoFisher). Each subarray on the slide is covered with 1934 spots with a 100 μm diameter, containing approximately 200 million uniquely barcoded oligonucleotides with poly-$T_{20}$ VN capture regions per spot (Barcoded slides were manufactured by 10X Genomics Inc, probes were manufactured by IDT). The full protocol, including sequencing and computational analysis was performed for 8 sections across 3 samples. Sample 1 and sample 3 each include three sections of the caudate lobe. Sample 2 includes two sections of a liver piece of the right liver lobe. All sections of all samples have undergone the same treatment.

**Histological staining and annotations**. We performed the ST workflow according to Ståhl et al. and Vickovic et al., respectively[60,61]. After 10 minutes of formalin fixation of the tissue on the slides they were dried with isopropanol and stained with Mayer's hematoxylin (Dako, cat.no.: S330930-2) and bluing buffer (Dako, cat.no.: CS70230-2) followed by Eosin (Sigma-Aldrich, cat.no.: HT110216-500ML) (H&E), diluted in Tris/acetic acid (pH 6.0). The stained sections were mounted with 85% glycerol (Merck Millipore, cat.no.: 8187091000) and covered with a coverslip. Bright-field images were acquired at 20x magnification, using Zeiss AxioImager.Z2 microscope and the Metafer Slide Scanning System (Metasystems).

The liver images were assessed by a mouse liver expert (NVH) who annotated the portal (blue) and central (red) veins, based on the presence of bile ducts and portal vein mesenchyme (PV) or lack thereof (CV). When the quality of the sample did not allow for annotation, "ambiguous vein" (green) was reported.

**Immunofluorescence assay (IFA)**. For immunofluorescence labeling, liver sections were (co-)stained with antibodies directed against GS (1:1000; ab73593, Abcam), SOX9 (1/100; AB5535, Millipore), F4/80 (1:100; MCA497RT, BioRad), HNF4α (1/100; sc6556, Santa Cruz) or CD31 (1:200; MEC13.3, BioLegend). Secondary antibodies were either anti-rabbit AlexaFluor 488; or anti-rabbit, anti-goat or anti-rat AlexaFluor 647 (1:1000, Invitrogen). Hoechst (1:10 000, 62249, Thermo Scientific) was used to counterstain the nuclei. Liver sections were imaged using the Zeiss AxioImager.Z2 microscope.

**Permeabilization, cDNA synthesis, tissue removal, and probe release**. Next, the slides were put in mask holders (ArrayIT) to enable separated on-array reactions in each chamber as described previously[61]. Each tissue section was pre-permeabilized using Collagenase I for 20 min at 37 °C. Permeabilization was performed using 0.1% pepsin in 0.1 M HCl for 10 min at 37 °C. cDNA synthesis was performed overnight at 42 °C. Tissue removal from the arrays prior to probe release was performed using Proteinase K in PKD buffer at a 1:7 ratio at 56 °C for 1 h. Last, the surface probes were released and cDNA library preparation followed by sequencing was performed.

**cDNA library preparation and sequencing**. Released mRNA-DNA hybrids were further processed to generate cDNA libraries for sequencing. The sequencing libraries were prepared as described in Jemt et al.[62]. In short, the 2nd strand synthesis, cDNA purification, in vitro transcription, amplified RNA purification, adapter ligation, postligation purification, a second 2nd strand synthesis and purification were done using an automated MBS 8000+ system. To determine the number of PCR cycles needed for optimal indexing conditions a qPCR was performed, as described previously[61]. After the determination of the optimal cycle number for each sample, the remaining cDNA was indexed and amplified. The indexed libraries were then purified using an automated system as previously described[63]. The average length of the indexed cDNA libraries was determined with a 2100 Bioanalyzer using the Bioanalyzer High Sensitivity DNA kit (Agilent, cat.no.:5067-4626), concentrations were measured using a Qubit dsDNA HS Assay Kit (Thermofisher, cat.no:Q32851) and libraries were diluted to 4 nM. Paired-end sequencing was performed on the Illumina NextSeq500 platform, with 31 bases from read 1 and 46 bases from read 2 resulting in the generation between 15 and 32.1 million raw reads per sample. To assess the quality of the reads fastqc (v 0.11.8) reports were generated for all samples.

**Spot visualization and image alignment**. The staining, visualization, and imaging acquisition of spots printed on the ST slides were performed as previously described[60]. Briefly, spots were hybridized with fluorescently labeled probes for staining and subsequently imaged on the Metafer Slide Scanning system, similar to the previous acquisition of the H&E images. The previously obtained brightfield of the tissue slides and the fluorescent spot image were then loaded in the web-based ST Spot Detector tool[64]. Using the tool, the images were aligned and the spots under the tissue were recognized by the built-in recognition tool. Spots under the tissue were slightly adjusted and extracted.

**Computational analysis**

*Mapping, gene counting and demultiplexing*. Processing of raw reads was performed using the open-source ST Pipeline (v 1.7.6)[65]. In short, quality trimming was performed and homopolymer stretches longer than 15 bp were removed. The reads were subsequently mapped to the annotated reference genome (GRCm38 v86 and corresponding GENCODE annotation file) using STAR (v 2.6.1e)[66]. After filtering, PCR duplicates were removed and gene count matrices were generated.

*Dimensionality reduction and clustering*. Main computational analysis of spatial read-count matrices was performed using the STUtility package (v 0.1.0)[67] in R (v 4.0.2). The complete R workflow can be assessed and reproduced in R markdown (see code availability section). First, count matrices and metadata were loaded, translating Ensembl IDs to gene symbols simultaneously. Reads of individual samples were filtered to keep only protein-coding genes and subsequently normalized using the *SCTransform* function in Seurat[68,69]. The created objects were then integrated using the canonical correlation analysis (CCA) with the *MultiCCA* function provided in https://github.com/almaan/ST-mLiver[70]. Normalization of integrated data was performed, regressing out sample identities using the *SCTransform* function in Seurat. Thereafter, the CCA vectors were subjected to shared-nearest-neighbor (SNN) inspired graph-based clustering via the *FindNeighbors* and *FindClusters* functions. For modularity optimization, the louvain algorithm was used and clustering was performed at a resolution of 0.3 for clustering granularity.

*Visualization and spatial annotation of clusters*. To visualize the clusters in low-dimensional space and on the spot coordinates under the tissue, non-linear dimensionality reduction was performed using UMAP with the CCA vectors as input. Visualization and annotation of identified clusters in UMAP space, on spot coordinates as well as superimposed on the H&E images was performed using the Seurat and STUtility package.

*Differential gene expression analysis (DGEA) and expression programs*. Differential gene expression analysis of genes in identified clusters was performed using the function *FindAllMarkers* from the Seurat package. Following the default option of the method, differentially expressed genes for each cluster were identified using the non-parametric Wilcoxon rank-sum test. Initial thresholds were set to a logarithmic fold-change of 0.25 to be considered differentially expressed in a cluster and to be present in at least 10% of the spots belonging to the same cluster. Representative markers for each cluster were further selected, by choosing genes with a positive logarithmic threshold above 0.5 and an adjusted *p* value below 0.05. *P* value adjustments are based on Bonferroni correction using all genes in the dataset.

After the identification of marker genes of the individual clusters, we identified expression programs of genes (module scores) for clusters we identified to have spatial distribution in our data. These were cluster 1 (periportal cluster), cluster 2 (pericentral cluster) and cluster 5. The creation of expression programs was performed using the *AddModuleScore* function in Seurat. In brief, we stored the marker genes of each cluster in a list to serve as input for the function. From this input, the average expression of each program (list of markers) was calculated for each spot under the tissue and subtracted by the aggregated expression of a control gene set. Here, the control gene set included all genes present in our data. All analyzed genes were binned based on averaged expression and with the default number of 24 bins for the function, and 100 control genes of the control feature set were randomly selected from each bin. Higher scores indicate more marker genes of the program to be highly expressed in a spot, while lower scores indicate that no or only a small number of genes is expressed at low levels in the spot.

*Zonation based DGEA of immune system process and metabolic pathway markers*. To infer on DEG of immune system processes and metabolic pathways (ha-ras, chronic hypoxia and pituitary hormone metabolic pathways) between cluster 1 and cluster 2, DGEA between genes associated with immune system processes (GO:0002376) and metabolic pathways (KEGG) was performed. Each resulting list was cross-referenced with the normalized spatial data and the expression matrix was subset according to the respective cell type followed by DGEA between cluster 1 (portal) and cluster 2 (central) with a logFC threshold of 0.01 and significance (p_val_adj) below 0.05. The same sets of genes were used to perform bivariate expression by distance analysis on (see "Bivariate expression by distance analysis" and "Model comparison").

*Spatial autocorrelation*. To explore the correlation between spatial distribution and expression of all genes in our data spatial autocorrelations using the *CorSpatialGenes* function of the STUtility package was performed. The method is based on building a connection network from the spot-coordinates for each spot and the four surrounding neighbors at a maximum distance of 150 μm. Thereafter, individual connection networks are combined to a tissue-wide connection network to compute autocorrelations for the whole dataset. Based on the neighbor groups of each spot, lag vectors for all input features are calculated, essentially being the sum expression of the respective feature in the neighbor spots. This considered, neighboring spots with high spatial autocorrelation of features demonstrate similar expression levels. This allowed us to compute the correlation score between the lag vector and the actual expression vector to estimate spatial autocorrelations.

*scRNA-seq data*. Publicly available scRNA-seq data were analyzed to compare and complement the spatial data in our studies. For this purpose, we used the scRNA-seq dataset of cells originating from liver tissue from the Mouse Cell Atlas (accessed 2020-10-06)[41].

For comparative analysis and visualization, scRNA-seq data of the Mouse Cell Atlas was analyzed using the Seurat package (v 3.2.2). The count-data was first filtered for mitochondrial genes and normalized using the *SCTransform* function. Dimensionality reduction was performed using PCA and graph-based clustering was performed using the *FindNeighbors* and *FindClusters* function with a resolution of 0.8 for clustering granularity. Visualization of the clusters in low-dimensional space was performed using non-linear dimensionality reduction (UMAP). Clusters were grouped by the cell type annotations provided by the metadata of the single-cell dataset. The second dataset used for comparative analysis was extracted from single-cell spatial reconstruction data[15]. Differential gene expression data between layers of zonation was compared to markers for pericentral or periportal zonation in our dataset using R (v 4.0.2).

*Correlation analysis*. Correlation analyses between genes of clusters were performed using Pearson correlation, establishing linear correlations between differentially expressed genes of the clusters in base R. Visualization of correlation values was carried out using the corrplot package (v 0.84). The correlation coefficients of

the matrix were ordered using the method "FPC", describing the first principal component order of the correlation coefficients.

To explore the correlation relationship between single cells (assigned to the classes "pericentral hepatocytes" and "periportal hepatocytes") and the spatial transcriptomics "pericentral (cluster2)" and "periportal (cluster1)" clusters, Spearman rank correlation coefficients were calculated. First module scores of genes assigned to each cluster were calculated for each dataset: ST and scRNA-seq of the Mouse Cell Atlas. Notably, not all genes present in one dataset were present in the other, therefore only genes present in the respective dataset were considered. Thereafter, Spearman rank correlation between the scores for all groups ("pericentral hepatocytes", "periportal hepatocytes", "periportal (cluster1)", "pericentral (cluster2)") were performed. The relationships were visualized using the corrplot package, with values ordered in the original input order.

To identify genes that exhibited spatial zonation with respect to cluster 5, we employed the concept of modelling feature values (e.g., expression) as a function of the distance to a given structure (e.g., veins), see methods section "Features as a function of distance" for a more elaborate account of these ideas. In this analysis, we let cluster 5 represent our reference structure, and "measured" the minimal distance for every spot to the reference. As cluster 5 represents a small cluster with distinct localization in a specific region of the tissue and to include more data points for the reliable investigation of gradual expression from this cluster a higher threshold of 800 µm (TN = 284 pixels), compared to 400 µm for vein-distance analyses was chosen. Next, for every gene, we calculated the Spearman correlation between the spots' expression values and distances to cluster 5. Genes with a positive Spearman correlation value have an elevated expression as the distance to cluster 5 increases, while the opposite is true for genes with a negative correlation value. Notably, the relationship between distance and expression is not necessarily linear as the Spearman correlation — in contrast to Pearson correlation — assesses monotonicity rather than linearity. The Spearman correlation does not assume normality, and accurate p-values can be analytically derived, but must (in our case) be corrected for multiple hypothesis testing, for this purpose we use the Holm–Šidák method (implemented in the *statsmodels* submodule *stats.multitest* function *multipletests*). Having corrected the p-values, genes with a significant (adjusted *p* value < 0.05) correlation value could be extracted and further examined.

*Pathway analysis.* Functional enrichment analysis of marker genes of clusters was performed using g:Profiler2 (v 0.1.0). For the analysis, we extracted the gene symbols of each cluster and stored them in a list. The function "gost" of the g:Profiler2 package was then used to perform gene set enrichment analysis on input marker gene lists. In short, the function maps genes to known functional information sources and detects statistically significantly enriched terms. Since our data consists of murine liver sections, the organism was set to *mus musculus* and the source was set to Gene Ontology (GO) biological processes. Visualization of the 5 most significantly enriched processes for cluster 5 was performed using ggplot2 (v 3.3.2). Significance was adjusted using g:SCS (Set Counts and Sizes), as originally described by the authors of the g:Profiler package[71]. Enrichment scores are represented as the negative log10 algorithm of the corrected p-value. For visualization of functional enrichment on the tissue coordinates, marker genes of cluster 5 were referenced against all genes belonging to the gene ontology (GO) terms for "collagen fibril organization (GO:0030199)" and "response to cytokine (GO:0034097)", extracted from the GO browser of the Mouse Genome Informatics database. Gene expression programs were generated for genes belonging to each GO term as described before and visualized on the spots.

*Single-cell data integration* (stereoscope). The spatial data were integrated with the MCA dataset using *stereoscope*, a probabilistic method designed for spatial mapping of cell types[40]. In short, *stereoscope* assumes that both single cell and spatial data follows a negative binomial distribution, learns cell type-specific parameters from the (annotated) scRNA-seq data, and then uses these parameters to deconvolve the gene expression profile associated with each spot into proportion values of each cell type. *stereoscope* uses a stochastic gradient descent approach, leveraging the PyTorch framework, to obtain the maximum likelihood/maximum a priori estimates of both the parameter estimates and proportion values. In both steps (parameter estimation and proportion inference) a batch size of 2048 and 50000 epochs were used, a custom list of highly variable genes—see next section for details—was used rather than the full expression profiles; default values were used for all other parameters. *stereoscope* can be accessed at https://github.com/almaan/stereoscope, where more detailed documentation regarding the parameter values is provided. The *stereoscope* version used in the study was v.0.3 (commit: aacd5f775b73b138e504c35ff0cb3ffafbfc78ff).

The cell type proportion values were overlaid on the tissue section images by using the *FeatureOverlay* function in the STUtility package. To make our visualization more robust to outliers, we scaled all the proportion values using what we refer to as *quantile scaling*. Here, this procedure was performed in two steps: First, all values larger than the 0.95 quantile are changed to this quantile value (i.e., the data is clipped); then, within every cell type and section we divide the clipped values by their maximum, effectively mapping them to the unit interval [0,1]. Thereafter, the proportion values for all 20 cell types in the single-cell dataset were plotted on the spot coordinated and overlaid on the H&E stained tissue sections.

*Pearson Correlation of cell-type proportions.* The estimated cell-type proportion values do not comply with most of the assumptions to analytically compute confidence intervals for (e.g., normality and heteroskedasticity). Therefore, we used a bootstrap approach to compute confidence intervals (CI), and thus able to call signals as significant (zero not being included in the CI) or not (zero being included in the CI). For each pair of cell types, we generated 10,000 bootstrap samples and let the mean of these samples constitute a representative correlation value, while a 95% confidence interval was constructed around this by using the 2.5th and 97.5th percentiles as lower and upper limits. Pairs where the confidence interval overlaps with zero, i.e., being nonsignificant, are indicated with a magenta border.

*Selection of highly variable genes for* stereoscope. Seurat (v 3.2.2) was used to extract a set of highly variable genes from the MCA single-cell data, following the procedure recommended in the online Seurat *Clustering Tutorial* [https://satijalab.org/seurat/v3.2/pbmc3k_tutorial.html]. To elaborate, the following two steps were applied to the MCA single-cell dataset in sequential order: data normalization (*NormalizeData*, default parameters), and identification of highly variable genes (*FindVariableFeatures*, selection.method = "vst", features = 5000). The complete set of extracted genes used in the *stereoscope* analysis are listed in Supplementary dataset 7.

*Cluster interaction analysis.* To gauge the extent to which the expression-based clusters interacted in the spatial data, we constructed a simple interaction analysis based on a nearest neighbor approach. First, for every spot within each cluster, the cluster identity of the four nearest neighbors within a distance threshold are identified. To avoid confusion, note how these distances refer to the separation of spots in the ST array and not in gene expression-space. The distance threshold is used to ensure that only spots in the actual physical neighborhood are included in the count, as might not be the case for spots near the edge otherwise. Second, once neighbor identities have been registered for all members of a cluster, we convert these integer values to a fraction by dividing them with the total number of neighbors associated with the cluster. Thus, for any given cluster we have a set of $n\_cluster$ values representing the total fraction of neighbors that belong to each one of the clusters. Since spots need to be positioned somewhere in space, clusters with a large member count will by default neighbor more spots than a cluster with low member count. Hence, to assess whether an interaction seems to be present or not, one must account for cluster size and spatial organization of the spots; here done by random permutation of the cluster labels (100 times) followed by re-calculation of the same neighborhood fraction values. This allows us to put the observed neighborhood fractions into context, to what might be expected by random chance given the cluster cardinalities and spot organization. It is by this approach that Supplementary Fig. 4 was generated, where each bar represents the observed values, the dashed black line the empirical mean value from the permutation analysis, and the magenta envelopes filling the area of two standard deviations from the mean.

*Features as a function of distance.* To examine how certain features of interest (e.g., gene expression or proportion values) were influenced by the physical proximity to morphological structures (e.g., central and portal veins) in the tissue samples, an approach to model these values as a function of the distance to said the structure was devised. This procedure is described in detail below:

Using the brightfield H&E-images, a mask was created for each morphological structure. These masks covered all pixels considered to belong to the structure. Each structure was assigned an individual (numerical) id, and one or more class attributes related to it (e.g., "*vein type*"). As the spots' (capture locations) positions relate to pixel coordinates in the H&E-image, it was possible to—computationally—measure the distance from a spot to each of these structures.

The distance ($d(s,t)$) from a spot $s$ to a structure $t$ was here defined as *the minimal (euclidean) distance from the center of spot $s$ to any pixel $p$ belonging to the mask of $t$*. In other words, if $M_t$ is the set of all pixels in the mask belonging to structure $t$ then:

$$d(s, t) = argmin_{p \in M_t} d(s, p) \qquad (1)$$

The same procedure was used when determining the distance to a specific class attribute (e.g., vein type), except that the *union of all masks associated with a structure of said class* was used instead of only a single mask. That is, if $M_C$ is the set of all pixels belonging to any structure of class $C$, then the distance ($d(s,C)$) between spot $s$ and class $C$ is:

$$d(s, C) = argmin_{p \in M_C} d(s, p) \qquad (2)$$

*Univariate expression by distance analysis.* Once distances were determined, for a feature $x$ of interest (e.g., expression value) and a structure $t$, a tuple ($d(s,t),x_s$) was formed for each spot $s$; i.e. the distance for every spot was associated with the value of the feature. This set of distance-feature tuples could then be visualized in graphs, in order to depict the feature values' dependence on their distance to the structure.

To better capture general trends in the data, *scatterplot smoothing* (using the *loess* function from the scikit-misc package, v 0.1.3, default values for all parameters), was applied to generate smoothed estimates. The smoothed values would then serve as an approximation of a function $f$ such that $x_s = f(d(s,t))$, to be interpreted as if the feature value was a function of the distance to the structure. Plotting the smoothed values against their associated distances results in a

visualization of the function approximation over the distance domain; these plots are referred to as "*feature by distance*" plots; where the feature for example could be *expression* or *proportion*.

Unless otherwise stated, feature-distance tuples across all sections were aggregated when generating features by distance/distance-ratio plots. The envelopes encapsulating the smoothed approximation represent one standard error (SE) as given by the loess algorithm.

*Bivariate expression by distance analysis.* To better account for synergies between structures of various classes we also introduce a bivariate model, where we model the gene expression as a function of the distance to multiple reference structures. Rather than using a nonparametric function approximation—as in the univariate case—we favored a bivariate linear model, which would allow us to compare the influence of each independent variable (vein distances) on the explanatory power of the model. We define what will be referred to as the *full* model as:

$$y_{sg} \sim \beta_{0g} + \beta_{1g} d_{sc} + \beta_{2g} d_{sp} \tag{3}$$

Where $y_{sg}$ is the gene expression of gene $g$ in observation $s$, while $d_{sc}$ is the distance to the nearest central vein for observation $s$, and $d_{sp}$ is the distance to the nearest portal vein. $\beta_{ig}$ are coefficients specific to each gene, which are to be estimated from the data. For parameter estimation we use the *OLS* class from the *statsmodels* package submodule *regression.linear_models*, no regularization was used.

To visualize the results, we create a grid over the domain of distances to the central and portal veins that we seek to survey. Then for every node $s$ in the grid, we apply Eq. (3) to get a prediction of the gene expression. Finally, we plot the grid and let the intensity of the nodes be proportional to the predicted value, using linear interpolation between the grid points to color the whole domain.

*Model comparison.* The bivariate model also allows one to test whether the inclusion of covariates such as distance to either vein-type significantly improves the model's performance, or if a reduced model is sufficient to use. We thus introduce the three following *reduced* models:

$$y_{sg} \sim \beta_{0g} + \beta_{1g} d_{sc} \tag{4}$$

$$y_{sg} \sim \beta_{0g} + \beta_{2g} d_{sp} \tag{5}$$

$$y_{sg} \sim \beta_{0g} \tag{6}$$

We refer to the models described in Eqs. (4)–(6) as reduced since they are all nested with the full model, meaning that their performance can be compared with a likelihood ratio test; using one degree of freedom for the two reduced models with one distance covariate, and two degrees of freedom for the intercept-only model. The outcome of the likelihood ratio test (presented as a $p$ value), states whether the full model significantly ($p$ value < 0.05) outperforms the reduced model, accounting for the additional model parameters.

We selected genes to be subjected to bivariate expression by distance analysis in two different ways. First, in the case of metabolic pathway gene markers[12] for glucagon and Wnt targets, we extracted 12 known Wnt pathway markers genes with most elevated expression levels in the central cluster (cluster 2) in the spatial data. For the glucagon targets we chose 10 known marker genes with the most elevated levels in the portal cluster (cluster 1) and 2 genes with highest up- or downregulation (*Mup20, Mdm2*) in glucagon deficient mice[47]. Secondly, for the remaining bivariate expression by distance analyses of gene markers (immune system process, ha-ras, chronic hypoxia, pituitary hormones), we selected 2–3 genes exhibiting most elevated expression levels for each, the central (cluster 2) and the portal cluster (cluster 1). These markers were identified as described in the Methods section: "Zonation based DGEA of cell type and immune system process and metabolic pathway markers".

*Expression-based classification.* To assess whether the gene expression of a structure's (e.g., central or portal vein) neighborhood held sufficient information to infer its class, we constructed a classifier designed to predict structure-class based on gene expression data. The steps of data processing and explicit details for the classification procedure are described below:

First, a *neighborhood expression profile* (NEPs) was created for each structure, representing weighted (by distance) average expression of a set of features (here genes) in the neighborhood of a structure. The neighborhood ($N(t)$) of a structure $t$ was defined as the set of spots with a distance less than a threshold $T_N$ to $t$. That is:

$$N(t) = \{s \in S | d(s, t) < T_N\} \tag{7}$$

Where $S$ is the set of all spots, while distances between spots and structures (d(s,t)) are defined and computed as described in the section "*Features as a function of distance*" above. In this study, we set the distance threshold ($T_N$) to 142 pixels. This threshold equals 400 μm and represents the longest distance between three consecutive spot centers in the same row. Having formed the neighborhoods, their associated expression profiles for a feature ($x_{N(t)}$) were assembled accordingly:

$$x_{N(t)} = \sum_{s \in N(t)} w_{ts} x_s \tag{8}$$

Where $w_{ts}$ are the distance-based weights given by:

$$w_{ts} = \frac{\hat{w}_{ts}}{\sum_{k \in N(t)} \hat{w}_{tk}}, \hat{w}_{ts} = exp(-d(s, t)/\sigma) \tag{9}$$

In this analysis, σ was set to 20. As multiple features are used, NEPs are represented by a vector of $N$ (the number of features used) elements, denoted as $\boldsymbol{x_{N(t)}}$. Each NEP was then given a class label, *portal* or *central*, based on the associated structure's annotations. The task of predicting class labels from the NEPs then surmounts to a *multivariate binary classification problem*, for which a logistic regression model was employed. Implementation-wise the logistic regression was performed by using the *LogisticRegression* class from *sklearn's* (v 0.23.1) *linear_model* module, a *l2* penalty was used (regularization strength 1), the number of max iterations was set to 1000, default values were used for all other parameters.

In short, the logistic model considers the class label ($z_t$) of a structure $t$ as Bernoulli variable conditioned on the NEPs, i.e:

$$z_t \sim Ber(p_t), \log\left[\frac{p_t}{1 - p_t}\right] = \beta x_{N(t)} + \beta_0 \tag{10}$$

Fitting the model equates to finding the maximum likelihood estimates of β and $\beta_0$ given the observed data and regularization terms. Once fitted, the class of a structure $t$ is taken as class 1 if $p_t \leq 0.5$ and class 2 if $p_t > 0.5$. However, $p_t$ is a continuous value that also can be interpreted as the probability of a structure belonging to each class (low values indicate more similarities with class 1 and vice versa)—offering a form of *soft* classification.

To validate performance, cross-validation strategies were implemented at two different levels: *section* and *sample*. In the former $K$-sections were set aside forming a *test set* while the remaining sections constituted the *training set*. The model was trained on the *training set* and evaluated on the $K$ sections in the test set. This procedure was iterated for all combinations of the pairs. Cross validation on the sample level was conducted in a similar fashion, but splitting w.r.t. samples rather than individual sections; here setting aside a sample is equivalent to excluding *all sections associated with the given sample*.

As the number of samples—and thus structures—were fairly low, using the complete expression profiles (i.e., all genes) would likely have led to an overfitted model (n_features ≫ n_samples). Thus, a reduced set of genes were used to construct the NEPs, extracted from the set of marker genes identified in the previously described differential gene expression analysis—this set of genes can be found in Supplementary Fig. 15.

**Statistics & Reproducibilty.** Samples were chosen according to the number of the available experiments. Each Spatial Transcriptomics slide contains 6 sub-arrays. All samples resulting in cDNA libraries of correct size and concentration of all performed experiments were considered. No statistical method was used to pre-determine sample size. Only data that was handled the same way during samples freezing and library preparation was considered for consistent final analysis. The experiments were not randomized and the investigators were not blinded to allocation during experiments and outcome assessment.

**Reporting summary.** Further information on research design is available in the Nature Research Reporting Summary linked to this article.

## Data availability

The datasets generated during and/or analyzed during the current study are available in the doi-minting zenodo repository "Spatial Transcriptomics to define transcriptional patterns of zonation and structural components in the liver [https://zenodo.org/record/5595907][72]. The raw expression data and spot files can be accessed at the Gene Expression Omnibus database with the accession code "GSE165141". The data used for comparative analysis of previously published data can be accessed at 12 and Gene Expression omnibus (accession code "GSE84498") as well as at [41][http://bis.zju.edu.cn/MCA/]" with raw data accessible at Gene Expression omnibus (accession code "GSE108097") and[29] with raw data accessible at Gene Expression omnibus (accession code "GSE137720"). Where applicable, GO terms extracted from the gene ontology browser of the "Mouse Genome Informatics database [http://www.informatics.jax.org/]". All relevant data supporting the key findings of this study are available within the article and its Supplementary Information files or from the corresponding author upon reasonable request. Source data are provided with this paper.

## Code availability

All code to reproduce the analysis can be accessed in a "GitHub repository [https://github.com/almaan/ST-mLiver]"[70], and further instructions to reproduce the data analysis have also been deposited to a "doi-minting repository [https://zenodo.org/record/5595907]"[72]. Functions and classes pertaining to the *feature by distance* and *classification* analysis have been assembled into a Python module (*hepaquery*), while the workflow used to produce the results is given in a set of notebooks. A CLI program to prepare the data for the distance-related analysis once masks have been created is also provided. See the repository documentation for more information regarding reproduction of the analyses.

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

## Acknowledgements

We would like to thank Carina Olivera and Emily C. Ross, for providing excellent technical support and expertise with the animal work for this study. We thank Keren Halpern, Ross Dobie, and John R Wilson-Kanamori for sharing additional data of their scRNA-Seq studies with us. The computations and data handling were partially enabled by resources provided by the Swedish National Infrastructure for Computing (SNIC) at Uppmax and thereby partially funded by the Swedish Research Council through grant agreement no. 2018-05973. This research project was generously funded by grants from: the Swedish Society for Medical Research (SSMF), The Swedish Research Council (VR) to J.A, J.L and E.R.A; The Jeansson Foundation to J.L, the Swedish Foundations Starting Grant/Ragnar Söderberg Foundation, and Karolinska Institutet to E.R.A.; The Sven and Lily Lawski Foundation to F.H., and the European Association for the Study of the Liver (EASL) and PRIMUS/21/SCI/006 project, funded by Charles University Grant Agency to J.M.

## Author contributions

J.A. conceived the study; J.L. and J.A. supervised the project; F.H., A.M., and S.K. optimized the ST methodology for liver; F.H. and S.S., designed and carried out ST experiments; F.H., L.L. and A.A. developed computational methods and analyzed data; L.L. generated the CCA function; F.H. and A.A. discussed the expression by distance, cluster distribution, and vein classification analysis, while A.A. formulated the models and implemented these in code. N.V.H. performed immunostainings histological annotations and manual cell count analysis; E.R.A., N.V.H., J.M. provided essential background on liver biology; A.B., E.R.A., N.V.H., J.M., and E.E. provided technical aspects on liver assays; F.H. and J.A. wrote the manuscript with the help of A.A., E.R.A., and J.L. All authors edited and gave critical input on the manuscript.

## Funding

## Competing interests

S.S., L.L., A.A., A.M., and J.L. are consultants for 10X Genomics Inc holding the IP for the ST technology. The remaining authors declare no competing interests.

## Additional information

**Peer review information** *Nature Communications* thanks Shalev Itzkovitz and the other anonymous reviewer(s) for their contribution to the peer review this work. Peer reviewer reports are available.

