## [Peer Review File · Nature Communications]

Reviewers' Comments:

Reviewer #1:

Remarks to the Author:

In this manuscript, Hildebrandt and colleagues use Spatial Transcriptomics (ST) to study liver zonation. Liver zonation has been recently studied with landmark-gene guided single cell RNAseq or with laser capture microdissection sequencing. The ST approach is a complementary technique, which provides important new insights into the intriguing phenomenon of liver zonation. While spatial resolution is relatively low (150um between spots), the complete spatial coverage of the tissue with thousands of spots enables analyses that preserve the combined distances from both central veins and portal veins, as well as the distances from the liver capsule. This leads to some interesting discoveries, e.g. the fibrotic cells at the Glisson capsule (cluster 5). I particularly liked the way in which the authors overcame the spatial resolution limit with their 'expression by distance' method. This is a kind of 'super-resolution' approach that provides continuous detailed spatial expression profiles. I have a few suggestions for additional analyses to be addressed in a revision:

1) A major power of the approach is the ability to simultaneously record the distances of each spot from both the portal vein and the central vein. This enables assessing how the expression of each gene is affected by these two distances, potentially leading to novel insights into regulation of hepatocyte zonation. Hepatocyte zonation is shaped by both morphogens secreted by the non-parenchymal cells that surround the central vein (e.g. Wnt and Rspo3) as well as blood-borne factor such as hormones, nutrients and oxygen (for a list of some of these putative targets see PMID 28166538, Supplementary Table 4). It would be interesting to perform multi-linear regression of the expression of each gene vs. both the distance to the closest portal node and to the closest central vein. One could then use Bayesian Information Criterion or other methods to assess which of these distances is the stronger explanatory variable. I would expect classic Wnt target pericentral genes such as Axin2, Lgr5 and Cyp2e1, to be strongly affected by the distance to the central vein, classic glucagon targets should be periportal (PMID 29555772), whereas for some genes such as Cyp2f2, which is both a Wnt-inhibited gene and a RAS-activated gene (periportal signal) identifying the dominant distance could be very interesting.

2) The fibrotic cluster 5 at the liver capsule is very interesting. Are there other genes that show zoned expression towards the liver capsule? This could be assessed systematically with the present data by seeking genes that are significantly correlated or anti-correlated with the distance from the capsule. Particularly interesting genes are markers of liver capsule macrophages (PMID 30222169).

3) The paper discusses pericentral hepatocyte genes and periportal hepatocyte genes, what about the mid-lobule-peaking genes such as Hamp, Hamp2 and Igfbp2. Are they expressed in the 'intermediate' cluster 0? In a related note, Figure 1C does not show any markers for cluster 0, is this a real cluster or simply spots with low number of reads? A violinplot of log10 (numbers of reads) for each cluster + umap such as Figure 1b colored by log10(number of reads per cell) would be highly informative here. If cluster 0 spots consist of low-read spots I think they should be removed.

4) The liver consists of zoned non-parenchymal cell populations (NPCs). The authors should attempt to produce an expression by distance plot for these as well. For example, highly zoned endothelial genes (PMID 30222169) include the pericentral genes Thbd and Cdh13 and the periportal Efnb2 and Ltbp4. Highly zoned hepatic stellate cell genes (PMID 31722201) include the pericentral Adamts12 and Sox4 and the periportal Ngfr and Tagln. The authors should explore whether these genes show up in the ST data and produce their zonation plots. In addition, the authors should explore the concept of immune zonation (PMID 33239787) – is there a higher summed expression of immune genes in periportal spots (this would be expected based on the increased abundance of periportal immune cells previously reported? Are there zoned Kupffer cell genes? One approach to address the zonation of NPCs would be to extract genes specific to the particular NPC population of interest and then perform DGE between spots in cluster 1 and 2 over the expression matrix normalized by the sum of these NPC-specific genes (for example see PMID 32814046).

Minor comments:

- 1) Row 762 – what are the units of 210 and why was this threshold chosen?
- 2) Figure 3b – the humps in Figure 3b are not real (e.g. see smFISH validations for genes such as *Cyp2e1* and *Cyp2f2* in PMID 28166538, these are clearly monotonic). The authors should consider a computational method to remove these artefacts.
- 3) *Lyve1* is considered in the manuscript a marker of lymphatic endothelial cells, is it really distinct from endothelial cells? Please examine the correlation with *Cdh5*.
- 4) For the deconvolution the authors may want to consider other papers establishing cell-type references from single cell datasets that had more comprehensive coverage of liver NPCs compared to Tabula Muris, e.g. PMID 31398325, PMID 30222169 and PMID 31722201.
- 5) Line 117- Something in these numbers does not add up, how can each spot contain 30 cells, yet only 5-10 of them are hepatocytes? Hepatocytes take up 80% of liver mass but 60% of the number of cells. Please check.

Shalev Itzkovitz

Reviewer #2:

Remarks to the Author:

With their work entitled Spatial Transcriptomics to define transcriptional patterns of zonation and structural components in the liver, Hildebrand et al make an important contribution to the basic understanding of the zonation of the liver via using Computational models. By applying the latest methods in the field of systems biology using spatial transcriptomic analysis, the authors were able to show a very deep insight into the expression signature depending on the spatial allocation of hepatocytes using cryosections of liver material. The assignment of structural functions to specific regions in the parenchyma is particularly interesting.

However, the work also has some small weaknesses that I think should be taken into account in a Revision process.

1. liver tissue from female mice was used as material. This fact is not included in the discussion of the results. Since it is known that the dimorphism of the liver can also have a strong influence on the zonation (also on the morphology within the parenchyma), this would be important to discuss.
2. it would be good if the authors would carry out some additional experiments to confirm some of their results with regard to the uncharacterised structures found.
3. it would be important to know whether this method could also be applied to human material in a relatively high throughput manner. Such a statement would certainly encourage other researchers in the field of liver zonation to apply this method to answer still open questions.

Reviewer #3:

Remarks to the Author:

This paper performed the spatial transcriptomics on a total of 8 sections of wild type adult mouse livers consisting of 19,017 genes across 4,863 spots (each covering 5-10 hepatocytes and up to 30 cells), to study the zonation of liver lobules. In this paper, the unsupervised clustering on spot-level gene expression revealed six unique clusters, where two correspond to periportal and pericentral regions, respectively, and one is claimed to be a previously uncharacterized liver structure. The further cell type identification using scRNAseq-derived cell type signatures found the co-localization of multiple cell types, but differing from the previous scRNAseq-based observation, the liver is found to be predominantly constituted by zoned hepatocytes. Moreover, authors observed the dependency between spatial distance and gene expression along lobule axis, which further motivated the computational prediction of portal and central veins from spatial gene expression.

This study is non-trivial. However, the low resolution (non-single cell level) of spots is the major concern, which may result in suspicious conclusions when conducting all clustering, biological interpretation and cell type identification. Further, this paper lacks in-depth analysis and validation, and some of the observations are over-interpreted, making the result and conclusion

less solid. Followed are the detailed comments.

1) Low cell resolution of spots:

The technical artifacts of “doublet” in scRNAseq leads to suspicious biological conclusion, while this phenomenon is much more severe in the spatial transcriptome of this paper, where “each spot contains between 5-10 hepatocytes and up to 30 cells in total per spot”. It may largely bias the unsupervised clustering, resulting in suspicious cluster interpretation.

1a) As the very first step, the authors are suggested to demonstrate the frequency of cell count in spots, e.g. histogram or density plot.

1b) It is crucial to investigate whether the cell count in spots can influence the clustering. The authors are suggested to overlay the cell count in spots on the spot clustering. It is expected that the clusters with small cell count may give the more confident and accurate cluster interpretation, compared to those with high cell count, which are of the average expression of multiple cell types.

1c) To further reduce the bias, authors are suggested to also directly use the spots with low cell count to conduct clustering, to see how many clusters can be made, followed by cell type identification and biological interpretation.

2) previously uncharacterized structure Cluster 5

Based on unsupervised clustering, authors found the cluster 5, which may suggest an uncharacterized liver structure. However, given the above concern about the low resolution, the cluster may not be a novel structure, but result from the mixed existing cell population within spots. Thus, the average expression of various cell types drive the high similarity within cluster 5 and make them different from the others.

3) Spot gene expression normalization

Considering multiple cells in each spot, the Reviewer is curious about the gene expression normalization? Does it normalize to the cell count? This is not a standard step in the traditional single cell RNAseq gene expression normalization, but if not done, the gene expression of spots might be misestimated, i.e., overestimated for spots with higher cell count and underestimated for spots with lower cell count. This may cause problems when conducting differential gene expression analysis.

4) Gene sets of interest

It would be interesting to see if the well-known gene set enrichment can be reproduced along the lobule axis, for example, the gradually decreasing Nutrient- and oxygen-rich condition, Mitochondrial β -oxidation, Gluconeogenesis, Glycogen synthesis, and the gradually increasing WNT signaling, Glycolysis, Lipogenesis from zone1 to zone3. The similar expectation also includes the well-known cell types along lobule axis.

5) inappropriate statistic

In Fig 2a), authors calculated the Pearson correlation between cell type proportions that do not follow normal distribution. This is inappropriate, since the Pearson correlation might be largely biased by the outlier values from the data.

6) overstatement of correlation analysis

Due to mixed cells in spots, authors used the scRNAseq-derived signatures to estimate the cell type proportions for each spot, followed by the correlation between those proportions. As mentioned in this paper, “Pearson correlation scores between cell type proportions across the spots show positive correlation, to be interpreted as spatial co-localization of non-parenchymal cells”.

However, the low values of cell type proportions may be only the noise, that is, those cell types do not exist in the spot. Meanwhile, the correlation herein, may be largely driven by the similarity between the scRNAseq-derived signatures, the gene sharing or co-expression among signatures. So, correlation may not be because of the real spatial co-localization, but just mathematical similarities. The above possibilities cannot be excluded with no solid validation.

As a negative control, the same method is suggested to apply to a single cell RNAseq data. If no similar observation is made, then the co-localization could be partially supported.

7) Discrepancy between spatial transcriptome and MCA

As mentioned in the paper, "A large portion of spots is assigned to cluster 1 and cluster 2, and 100% of the spots contain hepatocyte markers, showing that - spatially - the liver is predominantly constituted by zoned hepatocytes, while these cells only represent a very small fraction of the MCA data. This discrepancy illustrates the power of complementing single cell transcriptome data with spatial gene expression data to thoroughly delineate liver architecture and the transcriptional landscape of liver tissue, while simultaneously demonstrating the limits of scRNA-seq data integration."

However, the discrepancies may only result from the limit of spatial transcriptome rather than the limits of scRNA-seq. The spatial transcriptome spot is not at the single cell level, and cluster1/2 may cover both zoned and non-zoned hepatocytes. Thus, it is possible that, even with only a very small proportion of zoned hepatocytes, cluster1/2 is still good enough to be distinguished from the clusters comprising non-zoned hepatocytes and cells other than hepatocytes. With that said, "A large portion of spots is assigned to cluster 1 and cluster 2" cannot exclude the possibility that the liver has only a small proportion of zoned hepatocytes, as suggested by MCA. More validation is needed to make the conclusion.

8) Prediction accuracy of portal and central veins

The authors predicted the portal and central veins based on gene expression. Although conducted cross validation, the Reviewer cannot find the relevant performance evaluation, failing to see how good the prediction is.

First, regarding the prediction evaluation, the authors are suggested to provide the ROC curve; Second, regarding the prediction result, it is better to overlay the statics of prediction, eg., the log ratio on Fig 3c, so that others can see how confident the vein prediction is.

9) Immunostaining of human liver tissue

Considering the above concerns, in addition to H&E staining, some other Immunostaining with antibodies against liver zonation and cell types are also suggested, for example, CD73 for pericentral zonation and E-cadherin for periportal zonation. These might be overlaid on all clustering, cell types and vein prediction, which may serve as the orthogonal validation for multiple observations.

“Spatial Transcriptomics to define transcriptional patterns of zonation and structural components in the liver”

-

Responses to the reviewers

We would like to first thank all reviewers for their constructive feedback provided to the manuscript entitled “Spatial Transcriptomics to define transcriptional patterns of zonation and structural components in the liver” and hope we have addressed all their comments adequately and to their satisfaction.

We kindly note that several of the questions and comments provide rather extensive responses. The extent of these responses is explained by the fact that we have partially included main figures of the original manuscript as well as figures included in the revised manuscript to the respective responses for ease of inspection and interpretation and to clarify our strategy to the reviewers beyond what is written in the responses. Throughout the responses all figures are referred to as “Review Figure”. At the end of each response we clarify if and how we modified the text in response to each comment and specify which figures we included and how they will be referred to in the revised manuscript.

Literature references are included in each individual response and respective links to these references can be found at the end of each response.

Reviewer #1

1) model gene expression and test against reduced models

A major power of the approach is the ability to simultaneously record the distances of each spot from both the portal vein and the central vein. This enables assessing how the expression of each gene is affected by these two distances, potentially leading to novel insights into regulation of hepatocyte zonation. Hepatocyte zonation is shaped by both morphogens secreted by the non-parenchymal cells that surround the central vein (e.g. Wnt and Rspo3) as well as blood-borne factors such as hormones, nutrients and oxygen (for a list of some of these putative targets see PMID 28166538, Supplementary Table 4). It would be interesting to perform multi-linear regression of the expression of each gene vs. both the distance to the closest portal node and to the closest central vein. One could then use Bayesian Information Criterion or other methods to assess which of these distances is the stronger explanatory variable. I would expect classic Wnt target pericentral genes such as Axin2, Lgr5 and Cyp2e1, to be strongly affected by the distance to the central vein, classic glucagon targets should be periportal (PMID 29555772),

whereas for some genes such as *Cyp2f2*, which is both a Wnt-inhibited gene and a RAS-activated gene (periportal signal) identifying the dominant distance could be very interesting.

We would like to express our gratitude towards the reviewer for an excellent suggestion which helped us to elevate the quality of our work substantially. Initially, we sought to convey information about the (potentially) synergistic effects between portal and central veins by use of our log-ratio plots, but the suggested multi-linear regression approach is to some extent superior to the log-ratio approach. However, as a consequence of the reviewer's comment we have implemented a new approach that we believe is more appropriate to convey this information; we refer to this approach as the bivariate model and will describe it and our results more thoroughly below:

Bivariate Model: Description

As suggested (by the reviewer) we decided to model the (normalized) gene expression as a function of the two independent variables d_c (distance to nearest central vein) and d_p (distance to nearest portal vein). In addition, we included an intercept term in this model - to represent a "baseline" expression level. Since the model is small and only utilizes two covariates, we decided to not apply any regularization scheme to our model. Now, if we let y_g represent the expression of gene g , our model then reads:

$$y_g \sim \beta_{0g} + \beta_{1g}d_c + \beta_{2g}d_p$$

Once the coefficient values are estimated, we can *predict* the expression level for any combination of distance values. Here, we chose to use the same distance threshold as for individual vein distances of 400 μm . This revised threshold describes the longest distance between adjacent spot centers in the same row. (Please refer to minor comment 1 for more detailed information on the threshold). While the reviewer kindly provided us with suggestions of genes to examine, we first sought to confirm that this approach gave the expected results when applied to the central and portal marker genes that we've already studied more carefully. Thus, we fit the model to our data and predicted expression values over a domain of various distance values. Although the formula above constitutes a 2D plane in 3D space, it can be visualized in 2D plots by letting the color intensity represent the predicted "height" (expression values) of a given distance combination.

Bivariate Model: Evaluation

In order to assess whether this model produced reasonable results that aligned with our expectations, we first applied it to the known marker genes for central and portal veins. The results - displayed as 2D plots - for these 10 (5+5) portal and central marker genes are displayed in Review Figure 1, while a guide for interpretation of these plots is found in Review Figure 2.

Review Figure 1 | Heatmaps representing predicted normalized gene expression value for each of the 5+5 portal (top) and central (bottom) marker genes. The position in the xy-plane represents the distance to the nearest vein of a given type, while the intensity is proportional to the normalized gene expression. We will refer to these plots as *bivariate expression by distance plots*. Relative expression values for each gene are depicted in a color gradient ranging from low (dark) to high (light).

Spatial Domain

Distance Domain

Review Figure 2 | Guide to interpretation of the 2D plots given in Review Figure 1. The leftmost image represents the observed spatial domain and the colored crosses are certain reference positions, the right plot indicates the (approximate) position of these reference positions in the central vs portal distance plots.

As can be observed, all marker genes displayed the expected patterns, i.e. highest intensity in the upper left corner and bottom right corner for central and portal genes, respectively. We interpret this as a positive affirmation of the model's validity. Evidently, visual inspection can provide several insights such as the influence of a certain covariate on the gene expression, but we still sought to quantify these effects in a more formalized manner.

Bivariate Model: Testing Model Performance

Hence, in line with the reviewer's suggestion of testing the performance between a more complex model including both distances as covariates (full model) and a reduced model (including only one distance as covariate) we also implemented a likelihood ratio test (LRT). Using the LRT we test both reduced models (dropping the central alternatively portal distance covariate) against the full model, to better understand which covariates that were informative and which were not. In the LRT the full model was (in both cases) taken as the model where both covariates (distance to central and portal vein) were included while the three reduced models were obtained by excluding either the central, portal vein or both distance covariates. That is:

$$\text{Full model : } y_g \sim \beta_{0g} + \beta_{1g}d_c + \beta_{2g}d_p$$

$$\text{Reduced model 1 : } y_g \sim \beta_{0g} + \beta_{1g}d_c$$

$$\text{Reduced model 2: } y_g \sim \beta_{0g} + \beta_{2g}d_p$$

$$\text{Reduced model 3: } y_g \sim \beta_{0g}$$

To briefly explain how to interpret the result from the likelihood ratio test where the full model *does not* explain the data significantly better than any of the reduced models (accounting for the additional extra variable(s)):

- If the full model does not outperform reduced model 1, distance to the portal vein does not have a significant impact on the gene expression of the gene of interest
- If the full model does not outperform reduced model 2, distance to the central vein does not have a significant impact on the gene expression of the gene of interest
- If the full model does not outperform reduced model 3, distance to neither of the vein types is informative of the gene's expression. Effectively, this means that a gene expression is constant with respect to vein distances.

When examining the expression of portal and central maker genes, the full model was favored in all cases. These results are presented in Appendix Table 1. Having established confidence and value in the use of these bivariate plots together with the LRT, we felt comfortable to start using them in order to answer questions where we did not know the answers *a priori* to the analysis.

Bivariate Model : Further analysis

Encouraged by this and inspired by the reviewer's comments, we looked at additional sets of genes which are known to influence hepatocyte zonation. These included selected gene sets for morphogens such as wnt and ha-ras target genes [1,2,6] as well as gene sets which are known to be regulated by blood-borne factors, such as hormones (pituitary hormone targets)[3,6], nutrients (glucagon targets) [4] and oxygen (chronic hypoxia targets) [5,6].

We extracted a selection of genes for each set; wnt, ha-ras, glucagon, pituitary hormone, and chronic hypoxia targets and intersected them with the list of differentially expressed genes between the central and portal cluster in our spatial data. For glucagon targets we extracted a list of periportal genes which showed differential expression in glucagon deficient mice (*Gcg* ^{-/-}) [4].

We then examined the full model of the two genes showing the highest upregulation (*Mup20*) or downregulation (*Mmd2*) in glucagon deficient mice, described by Cheng et al. [4], and an additional 10 genes displaying differential gene up-regulated gene expression with the highest log-fold change in the periportal area of our spatial data (*Sds*, *Hal*, *Ctsc*, *Aldh1b1*, *Hsd17b6*, *Etnppl*, *Slc7a2*, *Apoa4*, *Gls2* and *Cyp17a1*).

To study the influence of the distance to central and portal veins on wnt expression, we followed a similar approach, by intersecting genes considered to be wnt target genes [1,6], and our spatial data. For the bivariate plots we then chose genes depicting a high log-fold change in the pericentral area, which have not been shown in other bivariate plots, as well as the wnt target genes suggested by the reviewer (*Axin2*, *Lgr5* and *Cyp2e1*).

For ha-ras, pituitary hormone and chronic hypoxia targets we selected the 2 genes with the highest log-fold change (indicating enrichment around the central vein) and the 2 genes with the lowest log-fold change (indicating enrichment around the portal vein). The multi-linear regression analysis for these suggested gene sets obtained the following results:

- 1) Glucagon target genes

For the investigated glucagon target genes, the majority shows highest expression in close proximity to the portal vein and further distance to the central vein illustrated by the highest expression value in the lower right corner in the bivariate plots (Review Figure 3). *Mmd2* expression is decreasing in further distance to the portal vein and the results of the LRT of *Mmd2* show no superior performance of the full model over reduced model 2 (portal vein distance only) (Review Figure 3, Appendix Table 2). This indicates that the distance to the portal vein is sufficient to explain the variation in spatial gene expression of *Mmd2* expression. For the spatial expression of *Mup20* both reduced models and the intercept-only model perform better than the corresponding full models. Thus spatial gene expression of *Mup20* seems to be expressed at relatively constant levels w.r.t vein distances and a model consisting of a single intercept would be a sufficient explanatory variable. In all other cases, where the highest expression close to the portal vein is

observed (*Sds*, *Hal*, *Ctsc*, *Aldh1b1*, *Hsd17b16*, *Etnppl*, *Slc7a2*, *Apoa4*, *Gls2* and *Cyp17a1*), the full model performs superior to both reduced models (central and portal) (Appendix Table 2). Hence, both distances are informative to predict spatial gene expression of these glucagon targets, i.e. increased expression in close proximity to the portal vein and increasing distance to the central vein.

2) Wnt target genes

In contrast to the expression of glucagon targets, most selected wnt target genes exhibit highest expression in closest proximity to the central vein and long distance to the portal vein (upper left corner in the bivariate plots) (Review Figure 4). The LRT results for all genes exhibiting this expression pattern show superior performance of the full model in comparison to the single covariate models, suggesting that distance to the central vein and the portal vein are informative to predict the expression of the wnt targets *Slc1a2*, *Cyp2a5*, *Cyp2e1*, *Gulo*, *Slc22a1*, *Lect2*, *Cyp2c37*, *Alh1a1* and *Cyp1a2* (Appendix Table 3). In the case of *Axin2* and *Lgr5*, the highest expression is observed along the portal axis and in close proximity to the central vein. The LRT shows inferiority of the full model in comparison to both reduced models (portal and central) and the full intercept for *Lgr5* expression. This observation indicates that *Lgr5* expression can be predicted to be expressed relatively constant across the tissue. For *Axin2* expression the full model outperforms the reduced portal model, suggesting that the distance to the central vein seems to be more influential in the prediction of *Axin2* zonation along the lobular axis. However, the reduced intercept-only model outperforms the full model at a significance threshold of 0.05, suggesting that although the distance to the central vein is more influential for *Axin2* expression, it is to a fairly small extent.

Review Figure 3 | Bivariate expression by distance plots for a set of periportal glucagon targets. The distance thresholds are set to a threshold of 400 µm on the x- and y-axis. Expression values are depicted as a gradient from low (dark) to high (light). Numbers in curly brackets after the gene name indicate that {1} the full model does not perform significantly better than the reduced portal model, {2} the full model

does not perform significantly better than the reduced central model, {3} the full model does not perform significantly better than either of the reduced models to explain gene expression along the lobular axis, {4} the full model does not outperform the most reduced model (intercept only), i.e., the gene expression can be taken as constant across the tissue w.r.t vein distances (p-value < 0.05).

Review Figure 4 | Bivariate expression by distance plots for a set of wnt targets. The distance thresholds are set to a threshold of 400 µm on the x- and y-axis. Relative expression values are depicted as a gradient from low (dark) to high (light). Numbers in curly brackets after the gene name indicate that {1} the full model does not perform significantly better than the reduced portal model, {2} the full model does not perform significantly better than the reduced central model, {3} the full model does not perform significantly better than either of the reduced models to explain gene expression along the lobular axis, {4} the full model does not outperform the most reduced model (intercept only), i.e., the gene expression can be taken as constant across the tissue w.r.t vein distances (p-value < 0.05).

3) Ha-ras, chronic hypoxia and pituitary hormone target genes

For selected ha-ras target genes we found that *Cyp2f2* (also observed as a portal vein marker) and *Apoa4* exhibit the highest expression closest to the portal vein and in further distance to the central vein. *Oat* shows the highest expression closest to the central vein and far from the portal vein, while *Mup17* shows the highest expression far from the portal vein but remains expressed along the central axis (Review Figure 5a). All expression profiles along the centrilobular axis, except for *Mup17* can be explained significantly better by the full model (Appendix Table 4). For *Mup17* the univariate model of the portal vein outperforms the full model, suggesting that the central vein represents the stronger explanatory variable for *Mup17* expression along the centrilobular axis than the portal vein.

The chronic hypoxia targets *Hal* and *Pck* exhibit the highest expression in close proximity to the central vein and in distance to the portal vein. However, the bivariate

model for *Pck* is outperformed by the reduced portal model as the distance to the portal vein alone explains the expression of *Pck* more accurately. In the case of the hypoxia targets *Gstm3* and *Slc1a2*, highest expression is observed closest to the central vein, with the bivariate model being superior to both univariate models (Review Figure 5b, Appendix Table 5).

The pituitary hormone targets *Fmo3* and *Igfbp2* display the highest expression closest to the portal vein, however the reduced portal model of *Fmo3* expression outperforms the full model, indicating the distance to the portal vein explains spatial expression of *Fmo3* better. For *Igfbp2* expression the bivariate model is not performing significantly better than either univariate model, indicating that the reduced models for the central and portal vein influence *Igfbp2* expression along the lobular axis but to a fairly small degree. For *Cyp4a10* and *Slc22a1* the highest expression can be observed closest to the central vein and far from the portal vein and the bivariate model performing significantly better for both genes, making both distances informative for the the expression of *Cyp4a10* and *Slc22a1* along the lobular axis (Review Figure 5c, Appendix Table 6).

Review Figure 5 | Bivariate expression by distance plots for a set of **a)** ha-ras [1], **b)** chronic hypoxia [2] and **c)** pituitary hormone targets [3]. The distance thresholds are set to a threshold of 400 μm on the x- and y-axis. Relative expression values are depicted as a gradient from low (dark) to high (light). Numbers in curly brackets after the gene name indicate that {1} the full model does not perform significantly better than the reduced portal model, {2} the full model does not perform significantly better than the reduced central model, {3} the full model does not perform significantly better than either of the reduced models to explain gene expression along the lobular axis (p -value < 0.05).

Taken together, we would like to thank the reviewer again for the suggestion to include a bivariate regression model with an associated test to interpret the influence of the distances to central and portal veins on gene expression simultaneously.

The results obtained from this additional analysis represent a substantial contribution to the manuscript and - in our opinion - explain the influence of different vein types on gene expression far better than the previously performed log-ratio plots.

In general we observed that many classical glucagon targets with differential gene expression in the portal area are best explained by both the distance to the central and portal vein. The same trend was observed for classical wnt targets. For instance, the wnt targets *Cyp2f2* and *Cyp2e1* are most highly expressed in close proximity to the portal or central vein, respectively. For both genes the central and portal vein are informative, as indicated by the LRT, suggesting synergistic effects of portal and central vein distance on the expression of these genes. For other wnt targets, such as *Axin2* and *Lgr5*, our results demonstrate only one or even no dominant explanatory variable for gene expression along the centrilobular axis, respectively. These results are very informative for the investigation of genes which are known to be influenced by morphogenes in opposite venous regions, such as *Cyp2f2* regulation by ha-ras and wnt [7].

As we consider the analysis and results suggested by the reviewer as superior to the log-ratio analysis, we are exchanging all log-ratio plots described in the original manuscript with the bivariate plots (Figure 3c) and include all tables for the statistical LRT analysis as well as representative bivariate plots for the investigated pathways in the supplementary information.

Figure 3b and the respective figure legend was changed to figure 3c in the manuscript, based on the reviewer's suggestions:

Line [1057 - 1064]

“Figure 3c | Visualization of influence of distance to both vein types on expression by bivariate expression by distance plots (see methods for details). Gene expression values are depicted in a gradient from low (dark) to high values (light). The distance of each gene to central veins between 0 and 400 μm is represented on the x-axis. Simultaneously, distances to portal veins for the same distance are depicted on the y-axis for each gene. High values in the bottom right corner indicate gene expression is predominantly observed close to portal veins and far from central veins, while high values in the upper left corner indicate the reverse observation (below graphs).”

Consequently, figure 3c has been changed to 3d and figure 3d has been changed to 3e in the revised version of the manuscript.

We modified the manuscript text referring to the newly obtained results accordingly in the results section from line [262 - 278] in the revised manuscript.

Further we consider the results of the bivariate expression analysis along the PC-PP axis of zonation and metabolic pathway markers very relevant and a valuable addition to our study. Therefore, we made the following changes to the manuscript text and added representative

bivariate plots and the LRT test results for all investigated metabolic genes in the Supplementary information (Supplementary Figure 14 and Supplementary table 5).

We further refined the discussion of the manuscript, including conclusions drawn from the additional metabolic zonation results presented here as follows:

Line [395 - 398]

“We further affirm the established relevance of zonation of multiple metabolic pathways along the porto-central axis^{5–7,10,11,13–15,53,54}, by tracing expression gradients from outer vein borders and across physical space.

”

and

Line [403 - 407]

“Nevertheless, the results illustrate the overall relationships of zonation markers, including NPC, metabolic pathway and immune markers with central and portal veins across the tissue, suggesting whether the distances to central and/or portal veins represent stronger explanatory variables for gene expression independent of the schematic organization of lobules in physical space.”

Lastly, we have added a description of the bivariate expression analysis to the methods section of the manuscript from line [776 - 839].

[1] <https://pubmed.ncbi.nlm.nih.gov/24214913/>

[2] <https://pubmed.ncbi.nlm.nih.gov/24535843/>

[3] <https://pubmed.ncbi.nlm.nih.gov/19943135/>

[4] <https://pubmed.ncbi.nlm.nih.gov/29555772/>

[5] <https://pubmed.ncbi.nlm.nih.gov/20103700/>

[6] <https://www.nature.com/articles/nature21065>

[7] <https://pubmed.ncbi.nlm.nih.gov/19275>

2) Investigate correlated/anti-correlated genes towards the liver capsule, zonated expression towards the liver capsule

The fibrotic cluster 5 at the liver capsule is very interesting. Are there other genes that show zonated expression towards the liver capsule? This could be assessed systematically with the present data by seeking genes that are significantly correlated or anti-correlated with the distance from the capsule. Particularly interesting genes are markers of liver capsule macrophages (PMID 30222169).

We are delighted to hear that the reviewer also considers cluster 5 of interest, and welcome the suggestion to further investigate the character of the tissue by looking at potentially zonated genes w.r.t. this structure. Thus, we constructed similar *feature-by-distance* plots as to what previously have been presented for the portal and central veins, but now using cluster 5 as a reference rather than the vein structures.

First we assessed zonation patterns of marker genes associated with the liver capsule macrophages (LCMs), as suggested by the reviewer. To this end, we created a short list of intersecting marker genes of LCMs extracted from PMID 30222169 and markers of cluster 5 attached here (Appendix Table 7). As can be seen in Review Figure 6, all the marker genes show a decreasing expression as the distance increases, implying the presence of zonation towards cluster 5.

Review Figure 6 | feature-by-distance plots of the liver capsule macrophages, as given in PMID 30222169. Each dot represents one spot, while the green line represents the smoothed curve of all observations within the set distance threshold of 800 μm .

Next we acted upon the second request by the reviewer, to find genes with potential zonation patterns in a more *unsupervised* manner using measures of correlation. We chose to use *Spearman* correlation here as we are not only interested in linear relationships but rather monotonic ones, and p-values to test for significance are easier to calculate analytically as the Spearman correlation has no assumption on normality. To briefly describe the process we implemented:

We selected all spots that were within a given distance threshold of 800 μm (which equals distance between four adjacent spot centers in the same row) to cluster 5. Next we calculated the Spearman correlation between the expression value of each gene and the distance to cluster 5. Since we are looking at a large set of genes and do not have an initial hypothesis that we test, we adjusted the p-values (for the Spearman correlation value) using the Holm-Šidák method. We removed all genes with an adjusted p-value larger than 0.05, and ranked the genes based on their correlation values. Genes whose expression level exhibit a positive Spearman correlation with the distance to cluster 5 increase in expression the further one gets from cluster 5 and vice versa. We display the top 15 genes with highest negative and positive Spearman correlation values in Review Figure 7 (negative Spearman correlation) and Review Figure 8 (positive Spearman correlation).

Review Figure 7 | feature-by-distance plot using cluster 5 as reference, here the top 15 genes with largest negative magnitude of the Spearman correlation between expression and distance to cluster 5. Each dot represents one spot, while the green line represents the smoothed curve of all observations within the set distance threshold of 800 μm .

Review Figure 8 | feature-by-distance plot using cluster 5 as reference, here the top 15 genes with largest positive magnitude of the Spearman correlation between expression and distance to cluster 5. Each dot represents one spot, while the green line represents the smoothed curve of all observations within the set distance threshold of 800 μm .

Finally, to get a better understanding of what sort of biological processes the genes that seem to have a higher expression in the vicinity of cluster 5 are related to, we subjected the top 15 genes to functional enrichment analysis using g:Profiler's web service (<https://biit.cs.ut.ee/gprofiler/gost>), querying against GO:BP (GO Biological Processes). The results are shown in Review Table 1 below, where several of the enriched processes are associated with ECM function/structure and immunity.

This independent GO:BP analysis performed here supports the analysis performed in the original manuscript (Figure 4c, attached here as Review Figure 9), using marker genes of cluster 5 identified by DGEA (Appendix Table 7). For instance, the GO:BP term "collagen fibril organisation" (GO:0097435), shows high enrichment in the results of both analyses.

Following the same strategy, we were interested in illuminating the underlying biological processes of genes that seem to be more highly expressed upon higher distance to cluster 5

(Review Table 2). Interestingly, the first 10 genes with the lowest adjusted p-value are associated with a variety of different metabolic processes, a hallmark function of liver tissue.

Review Table 1 | GO:BP functional enrichment of the 15 genes shown in Figure 7.

Database	Term Name	Term Id	adjusted p-value
GO:BP	supramolecular fiber organization	GO:0097435	8.280×10 ⁻³
GO:BP	Collagen fibril organization	GO:0030199	9.454×10 ⁻³
GO:BP	response to oxygen-containing compound	GO:1901700	1.402×10 ⁻²
GO:BP	response to organic substance	GO:0010033	2.175×10 ⁻²
GO:BP	response to cytokine	GO:0034097	3.867×10 ⁻²
GO:BP	sequestering of actin monomers	GO:0042989	4.967×10 ⁻²

Review Figure 9 | Figure 4c, original manuscript : Gene-ontology (GO:BP) enrichment for markers present in cluster 5. The Enrichment is given as the negative log₁₀ algorithm of the adjusted p-value (g:SCS correction, see methods) of the differentially expressed marker genes in cluster 5.

Review Table 2 | GO:BP functional enrichment of the 15 genes shown in Figure 8.

Database	Term Name	Term Id	adjusted p-value
GO:BP	arachidonic acid metabolic process	GO:0019369	2.547×10 ⁻⁹
GO:BP	lipid metabolic process	GO:0006629	5.924×10 ⁻⁹
GO:BP	fatty acid metabolic process	GO:0006631	3.778×10 ⁻⁸
GO:BP	long-chain fatty acid metabolic process	GO:0001676	5.210×10 ⁻⁸
GO:BP	olefinic compound metabolic process	GO:0120254	7.823×10 ⁻⁸
GO:BP	icosanoid metabolic process	GO:0006690	1.254×10 ⁻⁷
GO:BP	unsaturated fatty acid metabolic process	GO:0033559	1.254×10 ⁻⁷
GO:BP	carboxylic acid metabolic process	GO:0019752	3.374×10 ⁻⁷
GO:BP	oxoacid metabolic process	GO:0043436	3.833×10 ⁻⁷

We observe overlapping GO terms associated with structural tissue formation and integrity between two independently conducted analyses. This further strengthens our claim that cluster 5 is associated with mesenchymal structures and/or the Glisson’s capsule.

Taken together, the results presented here suggest that cluster 5 may represent an important transitional structure between the architectural integrity of the liver tissue (including liver mesenchyme and the Glisson’s capsule) and the remaining metabolically diverse and highly active liver tissue. The additional results shown here reveals apparent zonation of LCM markers (e.g., *H2-Eb1*, *Crip1*, *Tmsb4x*, *H2-Aa*, *H2-Ab1*, *Cd74*), towards the structure underlying cluster 5, further advocate for the presence of tissue from the Glissons’ capsule and the mesenchyme, as LCMs are often found in the mesenchyme and Glisson’s capsule [1,2]. Thus, the observed enrichment of immune related processes such as “response to cytokine” and “antigen processing and presentation of peptide or polysaccharide antigen via MHC class II” in both GO-term analyses can most likely be explained by the enrichment of LCM markers.

To recognize the - in our opinion - very important suggestion for the additional characterization of cluster 5 in our data, we will include the results of this correlation analysis in the

supplementary material as Supplementary Figure 17. To refer to these results, we edited the main text in the manuscript as follows to describe and interpret the results accordingly:

Line [330 - 334]

“Expression scores of markers involved in “collagen and fibril organization” are highest in spots of cluster 5 and in their direct proximity in the tissue and show low scores for the remaining tissue. This is supported by unsupervised Spearman correlation results, exhibiting negative correlation to the distance to cluster 5 and expression of Gsn, Col1a2, Col1a3 and Vim (Supplementary figure 18).”

In addition, we describe the correlation analysis in an additional paragraph of the methods from line [635 - 670].

[1] <https://pubmed.ncbi.nlm.nih.gov/20637938/>

[2] <https://pubmed.ncbi.nlm.nih.gov/27569723/>

3) Investigate the significance of cluster 0 and mid-lobule-peaking genes/number of reads/cluster

The paper discusses pericentral hepatocyte genes and periportal hepatocyte genes, what about the mid-lobule-peaking genes such as *Hamp*, *Hamp2* and *Igfbp2*. Are they expressed in the ‘intermediate’ cluster 0? In a related note, Figure 1C does not show any markers for cluster 0, is this a real cluster or simply spots with low number of reads? A violin plot of log10 (numbers of reads) for each cluster + umap such as Figure 1b colored by log10(number of reads per cell) would be highly informative here. If cluster 0 spots consist of low-read spots I think they should be removed.

Response:

The reviewer raises a valid point by mentioning genes belonging to the previously described mid-lobule zone between the periportal and pericentral zone. These genes include as stated by the reviewer: *Hamp*, *Hamp2* and *Igfbp2* [1,2]. We understand the reviewer's concern that cluster 0 does not represent a cluster annotation based on expression but differences in read number. Hence, we have tried to address the reviewer's concerns about the annotation of cluster 0, by executing an analysis in concordance with the provided suggestions.

As observed by the reviewer Figure 1c in the original manuscript attached here as Review Figure 10, DGEA between clusters did not result in any markers for cluster 0. Based on this

note, we would like to thank the reviewer for bringing this missing information in the figure legend to our attention we changed it accordingly:

Line [1012 - 1015]

“Figure 1c | Heatmap depicting expression values of the five most variable genes for each cluster after subjecting the six clusters to DGEA, with the exception of cluster 3, which resulted in only four significantly differentially expressed genes and cluster 0 which did not result in any significantly differentially expressed genes with the given parameters (Methods).”

In our previous analysis only genes with a log-fold change larger than 0.5 and significant adjusted p-value were considered as differentially expressed, in order to produce distinct sets of marker genes that would promote a less ambiguous assignment of identities to the clusters. However, genes exhibiting lower *positive* log-fold change values than the threshold value of 0.5 could still be considered as differentially expressed, but with less prominent changes than the genes we chose to consider as marker genes.

Thus, to further address the reviewer's comments, we performed a DGEA with a more permissive log-fold change threshold value - 0.25 instead of 0.5 - to also include weakly differentially expressed genes.

For cluster 0, this adjustment resulted in 3 significantly DEGs, being: *Hamp2*, *Hamp* and *Cyp3a44* (Review Table 3, Review Figure 11). Indeed, two of the genes, *Hamp2* and *Hamp* exhibit elevated mid-lobule expression, as expected by the reviewer.

Review Figure 10 | Figure 1c, original manuscript : Heatmap depicting expression values of the five most variable genes for each cluster after subjecting the six clusters to DGEA, with the exception of cluster 3, which resulted in only four significantly differentially expressed genes and cluster 0 without any significantly differentially expressed genes using defined parameters (see methods, original manuscript).

Table 3 | Cluster 0 markers with adjusted effect size. Results of DGEA of the spatial gene expression data with a lowered log-fold threshold identified 3 genes with elevated expression in cluster 0 in comparison to the remaining clusters (*Hamp2*, *Hamp* and *Cyp3a44*). The effect size of elevated expression ranges from 0.26 to 0.30 (avg_log2FC) and is significant for all 3 genes (p_val_adj < 0.05).

	p_val	avg_log2FC	pct.1	pct.2	p_val_adj
Hamp2	7.41E-70	0.26	1.00	1.00	7.05E-66
Hamp	8.26E-57	0.30	1.00	1.00	7.85E-53
Cyp3a44	5.25E-22	0.26	0.67	0.56	4.99E-18

Review Figure 11 | Heatmap showing the 3 most variable genes of cluster 3 resulting from DGEA with an adjusted log-fold change of 0.25. Expression values of the 3 genes with elevated expression in cluster 0 (*Hamp2*, *Hamp* and *Cyp3a44*) are depicted as a color gradient from purple (low) to yellow (high).

To further confirm the validity of cluster 0, we followed the reviewers suggestion to visualize the number of reads across clusters by violin plots and in a UMAP projection. To compare orders of magnitude in the amount of reads we used the log 10 of the total number of reads for the visualization in both cases. Both visualizations display a comparable and even number of reads across all clusters and an uniform distribution, without a higher number of reads present in cluster 0 (Review Figure 12 a,b), indicating cluster 0 is based on differential gene expression and not on read count.

Based on the reviewers question on the presence of the mid-lobule-peaking genes *Hamp*, *Hamp2* and *Igfbp2*, we regarded it appropriate to elucidate further on the distribution of these genes within our data and investigate their distribution across the annotated clusters. Thus, we visualized the distribution of expression levels across clusters in violin plots. As expected from the adjusted DGEA of cluster 0, *Hamp* and *Hamp2* show the highest expression in cluster 0, with almost equally high expression in cluster 3 and cluster 4. Meanwhile, expression in cluster 1 and cluster 2 is reduced for both, *Hamp* and *Hamp2*. Furthermore, *Hamp* also shows reduced expression in cluster 5 (Review Figure 12 c,d). However, *Igfbp2* generally shows lower overall expression in comparison to the other two genes, with the lowest expression observed for cluster 2 and an otherwise equal expression across clusters (Review Figure 12 e).

To summarize, we conclude that cluster 0 represents a real cluster based on differential gene expression based on the following 4 observations:

- 1) Cluster 0 does not exhibit a lower number of reads in comparison to the remaining 5 clusters.

- 2) Reads across all spots show a uniform distribution independent of their cluster annotation.
- 3) Differentially expressed genes can be extracted from Cluster 0, but with a lower fold change than for the remaining clusters.
- 4) The majority of identified genes with elevated expression in cluster 0 are also marker genes for mid-lobule layers, as would be expected by the spatial distribution of cluster 0 within the tissue.

The distinction of cluster 0 is less clear than that of the remaining clusters, as the effect sizes of the markers of cluster 0 are not large enough to be detected by the DGEA performed in the original manuscript. This observation, together with the fact that cluster 0 represents the cluster comprising the largest assembly of spots - in our opinion - do not represent enough evidence to confidently annotate cluster 0 as the mid-lobule layer, but rather suggests that cluster 0 includes the mid-lobule layer since it still exhibits DGE of classic mid-lobule peaking genes [1,2]. Based on the small effect size of differentially expressed markers in cluster 0, the multidimensional structure of the investigated tissue section and the gradual nature of zoned expression, we believe that extensive characterisation of cluster 0 is outside of the scope of this study, but is nevertheless very interesting and could be a potential objective in future research projects. Thus we believe including the extended DGEA of cluster 0 should not be included in this study.

However, the reviewers comments on the nature of cluster 0 are very valuable and the results from the analyses suggested by the reviewer provide an initial interpretation of cluster 0 and confirm its validity in our spatial dataset. We hope the reviewer agrees with us that we present interesting results that could act as inspiration for future studies on the mid-lobule regions in liver tissue. These will be critical to extend our current knowledge on the importance of the transcriptional landscape in the space between the portal and central area.

To highlight the relevance of the analysis suggested by the reviewer, we are including violin plots showing the distribution of transcript numbers per across clusters in Supplementary figure 1 of the original manuscript and refer to the analysis in the manuscript as follows:

Line [122-125]

“Subsequently, we decomposed the data into correlation vectors by canonical correlation analysis (CCA) and clustered it in an unsupervised manner using a graph-based approach, which identified 6 clusters, exhibiting uniform distribution of unique transcripts (Figure 1b, top panel, Methods for details, Supplementary figure 1)”

Review Figure 12 | Distribution of the number of unique transcripts and transcript counts of genes of interest across clusters. **a** The logarithmic number of unique transcript (base10) for each cluster reveals equal numbers of unique transcripts for each cluster. **b** The log10 of unique transcripts exhibits a uniform distribution in the UMAP embedding of spatial data and across cluster annotations ranging from low (light) to high (dark). **c** Violin plot showing the normalized expression level of *Hamp* for each cluster, depicting slightly elevated levels in cluster 0. **d** Violin plot showing the normalized expression level of *Hamp2* for each cluster, depicting slightly elevated levels in cluster 0. **e** Violin plot showing the normalized expression level of *Igfbp2* for each cluster, depicting uniform distribution of expression for each cluster and relatively low expression values.

[1] <https://www.ncbi.nlm.nih.gov/pmc/articles/PMC5321580/>

[2]

<https://science.sciencemag.org/content/371/6532/eabb1625?elqTrackId=d6c890a6e9b1447ca229282b1c3814a8>

4) Expression by distance plots for non-parenchymal marker genes

The liver consists of zoned non-parenchymal cell populations (NPCs). The authors should attempt to produce an *expression by distance* plot for these as well. For example, highly zoned endothelial genes (PMID 30222169) include the pericentral genes *Thbd* and *Cdh13* and the periportal *Efnb2* and *Ltbp4*. Highly zoned hepatic stellate cell genes (PMID 31722201) include the pericentral *Adamts12* and *Sox4* and the periportal *Ngfr* and *Tagln*. The authors should explore whether these genes show up in the ST data and produce their zonation plots.

In addition, the authors should explore the concept of immune zonation (PMID 33239787) – is there a higher summed expression of immune genes in periportal spots (this would be expected based on the increased abundance of periportal immune cells previously reported? Are there zoned Kupffer cell genes?

One approach to address the zonation of NPCs would be to extract genes specific to the particular NPC population of interest and then perform DGE between spots in cluster 1 and 2 over the expression matrix normalized by the sum of these NPC-specific genes (for example see PMID 32814046).

Response:

We would like to thank the reviewer for the highly constructive suggestions to include additional analysis on non-parenchymal cells (NPCs) and to elaborate further on the concept of immune zonation. We believe that the suggestions contribute a substantial amount of additional information to our study.

As suggested by the reviewer we explored the zonation of NPCs, namely endothelial and hepatic stellate cells. Previously described highly zoned endothelial cells included the pericentral genes *Thbd* and *Cdh13* and the periportal genes *Efnb2* and *Ltbp4* [1]. *Adamts12* and *Sox4* have been described as zoned pericentral markers for hepatic stellate cells, while *Ngfr* and *Tagln* show periportal expression in this cell-type [2]. Therefore, we created *expression by distance* as well as *bivariate expression by distance* plots for the requested marker genes.

We were only able to generate the requested plots for 6 out of the 8 genes, as *Cdh13* and *Efnb2* are not present in our data. The results of the expression by distance plots do not show obvious distinct differences in expression along the lobular axis for these genes as expected by the reviewer (Review Figure 13 a). However, upon investigation of the bivariate plots, taking distances to portal and central veins into account simultaneously the expression trends for some genes become more obvious. For instance, expression of *Ltbp4* can be better explained as the highest expression is observed in close proximity to both vein types and superior performance of the bivariate model in comparison to the reduced models. Hence, *Ltbp4* seems to be expressed between veins, located within 400 μm of each other. Spatial *Adamts12* expression is sufficiently explained by the reduced central model, indicating the distance to the central vein represents the stronger explanatory variable for *Adamts12* expression (Review Figure 13 b, Appendix Table 8). Surprisingly, we find *Sox4* expression along the central axis, in close proximity to the portal vein and not as expected close to the central vein and the LRT suggests

expression is explained better by the univariate portal model, although to a weak amount as the full model is also outperformed by the single intercept models. These observed differences to published data make *Sox4* an interesting target for further studies on NPC zonation across tissue. The reported portal marker *Ngfr* shows highest expression furthest from the portal vein and in closer proximity to the central vein, which is in contrast to the expectations of the reviewer and the literature [2]. However, the bivariate model is not able to explain the observed distances to either vein better than the reduced models and similar to *Sox4*, expression of *Ngfr* is quite low compared to other zoned genes in our data. Merely, *Thbd* expression shows an expected trend with high expression far from the portal vein and slightly closer to the central vein. However, the LRT of the portal and central vein distances do not perform better in comparison to the central and portal reduced models, making the observations for the bivariate models less relevant for the interpretation of spatial expression along the lobular axis of these genes.

Conclusively, zonation patterns of markers genes suggested by the reviewer (pericentral *Thbd*, *Cdh13*, *Adamts12*, *Sox4* and periportal *Efnb2*, *Ngfr*, *Tagln*, *Ltpb4*) in our spatial data deviate from previous observations. This most likely can be explained by two reasons. 1) All suggested genes show either no or very low expression along the lobular axis, making assumptions on their expression profile unreliable. As the suggested genes result from scRNA seq data, they are more efficiently captured by the nature of this method, e.g. through enrichment by sorting. Hence, transcript abundance of certain genes is relatively low in comparison to the remaining liver transcripts 2) ST encompasses expression profiles across whole liver sections, zoned or unzoned expression of lowly abundant genes might be enriched or present in limited locations within the tissue.

However, we wanted to investigate further on the DGEA of NPCs as suggested by the reviewer. To this extent we used publicly available single-cell data from Halpern et al. [1] with a more detailed description, generously provided by the authors of the paper. Using the provided data, we performed DGEA between the annotated cell types (Endothelial cells, T cells, Kupffer cells, B cells, Plasmacytoid dendritic cell (pDCs) and Neutrophils). This analysis resulted in a list of marker genes for each annotated non parenchymal cell (NPC) type. We then cross-referenced this list with our normalized spatial data and subsetted the expression matrix according to the cell type list of choice.

We considered the normalization method suggested by the author but believe it more appropriate for bulk RNA-seq data rather than data of the same character as ours. Differences between gene expression of NPCs are accounted for by the normalization approach used in this study (see materials and methods section in the main manuscript).

Therefore, we directly performed DGEA between cluster 1 (portal) and cluster 2 (central). We used a log-fold change threshold of 0.01 to also detect elevated gene expression with a relatively small fold change, in both spatial clusters. We obtained differential expression results for the following NPC types:

- 1) Liver endothelial cells

For portal endothelial cell markers our analysis identified *Sepp1*, *Aass* and *Ctsl* with the highest significantly elevated expression levels. Central endothelial cell markers are higher in number and include *Ndrp1*, *Lifr* and *Tsc22d1*, exhibiting the highest significantly elevated expression levels in the central zone (Review Figure 13 c (left), Appendix Table 9, Appendix). For *Sepp1* including the distance to the central and portal vein to assess zonation is preferred, showing the highest expression close to the portal and far from the central vein (Review Figure 13 c (right)). *Ass1* expression however can be explained better by the univariate model of the portal vein as the highest expression is observed close to the portal vein and along the central axis (Appendix Table 10). Even Though *Ctsl* expression is observed highest close to the portal vein and far from the central vein, the LRT of the bivariate model does not outperform neither the portal nor central model. Hence, the orientation within the lobular axis does not seem to be a significant explanatory factor for *Ctsl* expression.

2) Plasmacytoid dendritic cells (pDCs) and Neutrophils

pDCs exhibit elevated central zonation for *Dirc2*, *Upb1*, *Ctsh*, *Rnf187* and *Lgals1*. Portal zonation is observed for *Atp1b1* (Appendix Table 11). In addition, we also observe high expression elevation of the pDC markers *Mpeg1*, *Lgals*, *Atp1b1* and *Rnf187* in cluster 5 (Review Figure 13 d, (left)). The small number of neutrophil markers (*Grina*, *Dgat2* and *Gsr*) show highest elevation in the central zone (Review Figure X+1d) (Review Figure 13 d, (right), Appendix Table 12).

Taken together we can observe zoned expression of NPC marker genes along the lobular axis. The zoned NPC markers suggested by the reviewer are either not detected at all (*Cdh13*, *Efnb2*) at very low levels (*Thbd*, *Ltbp4*, *Adamtsl2*, *Sox4*, *Ngfr* and *Tagln*) in our data. Therefore, we are not able to reliably use these genes to confirm NPC zonation. However, we determine a number of zoned NPC markers for multiple cell types (Endothelial cells, Plasmacytoid dendritic cells (pDCs) and Neutrophils). Based on the sparsity of marker genes for these cell types in the central and portal area of spatial data, we can only identify a small number of zoned genes for each cell type. Nevertheless the results presented here complement previous studies with additional zonation markers for endothelial cells, pDCs and Neutrophils.

Kupffer cells, the liver resident macrophages, represent an additional NPC type in the liver. They are a vital part of the immune response, which was reported to be organized in a zoned fashion [3]. Inspired by the reviewer's highly relevant comment on immune zonation within the liver we gladly performed additional analyses to investigate this concept. To this end we investigated the potential zonation of Kupffer cell genes between the portal and central gradient across spots. We extracted Kupffer cell markers as described for other NPC populations above and performed DGEA between zone 1 (portal) and zone 3 (central) (Review Figure 14 a (left), Appendix Table 13). Selecting the three DEG with the highest significant log-fold change we find *Ctsc*, *Igf1* and *Ctsb* to be highly expressed in close proximity to the portal vein. For *Ctsc* and *Igf1*, increasing

distance to the central area is associated with elevated expression levels. *Ctsb* is expressed along the central axis but always in close proximity to a portal vein, making the distance to the central vein a relevant explanatory variable for gene expression.

For the DEG in the central area *Hpgd*, *Creg1* and *Apoe*, our data suggests the highest expression of the first two genes in close proximity to the central vein and far from the portal vein. *Apoe* is expressed close to the central vein and along the portal vein. However the results of the multivariate model suggest that distances to the central or portal vein do not perform significantly better than either both (*Hpgd*, *Creg1*) or only the central model (*Apoe*) (Review Figure 14 a (right), Appendix Table 14). This suggests that the distances along the lobular axis are not as informative to predict gene expression of these markers as for Kupffer cell markers, which were found to be differentially expressed in the portal area.

In addition to exploring the contribution of Kupffer cells to immune zonation, we analyzed whether established genes involved in immune responses are zoned in our spatial data. Therefore, we extracted genes belonging to the GO-term “immune system processes” and performed DGEA between cluster 1 and cluster 2 (as described above for cell type zonation analysis) (Appendix Table 15). We found that more genes show upregulation in the portal area compared to the central area, which is in line with the previously reported dominant role of the portal area for immune zonation (Review Figure 14 b (left)). Comparing the expression by distance the three most significant DEG of the portal and central cluster, DEG genes of the portal area (*Arg1*, *C9* and *Hc*) share the distance to the portal vein as a significant explanatory variable. In the case of *Arg1* and *C9* the distance to the central vein also represents a significant factor to predict their expression along the zonation gradient. The immune markers *Psm1* and *Mb11* with elevated expression levels in the central cluster are expressed along the portal axis. Only in the case of *C4bp* the multivariate model suggests the distance to the central vein as the dominant explanatory factor for gene expression along the lobular axis (Review Figure 14 b (left), Appendix Table 16). Hence, the results of the DGEA and bivariate expression by distance supports the dominant role of the portal area for zonation of immune related processes.

Collectively, our results of the investigation of immune zonation further support the hypothesis of immune zonation and that the distance to the portal area represents the stronger explanatory variable for Kupffer cells and general immune related processes. We would like to thank the reviewer again for his excellent suggestion to explore NPC and immune zonation further using our spatial data. We believe these additional results - especially in regard to immune zonation - represent an important addition to our study. Therefore, we are including the heatmap results of differential gene expression of NPCs and immune system processes in Supplementary Figure 8. Further we will include results of the bivariate expression by distance analysis of NPCs and immune system processes in Supplementary Figure 14 and Supplementary table 5. We also performed the following modifications to the main text in the manuscript:

Line [216-222]

“Based on these observations, we further investigated the zonation of reported markers of NPCs 13,27 (endothelial cells, HSCs, plasmacytoid dendritic cells, neutrophils and Kupffer cells)

in our data (Methods) and found several differentially expressed cell type markers between the PPC and PCC (Supplementary Figure 8). In the context of reported immune zonation ⁴⁰, we also investigated DEG of immune system process associated genes (GO:0002376) and found more genes with periportal than pericentral zonation (Supplementary Figure 8)."

We further discuss the observed results in regard to NPC zonation and immune zonation as follows:

Line [360-362]

"First, we assessed expression of characteristic marker genes within a wide range of expression levels and investigated zonation patterns of established cell type markers ^{13,27}."

and

Line [365 - 375]

"A recent study suggests predominant localisation of Kupffer cells in the periportal area of the liver lobule and neutrophil recruitment upon bacterial infection ⁴⁰. While, our data does not indicate elevated Kupffer cell proportions in the periportal cluster compared to the remaining clusters, we found a number of Kupffer cell marker genes exhibiting portal but also central zonation. In addition, we found more genes related to immune system processes with periportal enrichment in comparison to the pericentral zone and colocalization of neutrophils and periportal hepatocytes, already in unperturbed conditions, all supporting implications of previously proposed immune zonation ⁴⁰."

In addition, we modified the methods section of the manuscript to include the description of the performed analysis in detail:

Line [594 - 605]

"Zonation based DGEA of cell type markers

To infer on DEG of NPCs between cluster 1 and cluster 2, DGEA between annotated cell types (Endothelial cells, T cells, Kupffer cells, B cells, Liver Capsular Macrophages (LCMs), Plasmacytoid dendritic cell (pDCs) and Neutrophils of data and respective annotations, generously provided by Halpern et al. [cite], as well as LCMs, provided by Dobie et al. [cite] (see scRNA-seq data) was performed. This analysis resulted in a list of marker genes for each

annotated non parenchymal NPC type. Each resulting list was cross-referenced with the normalized spatial data and the expression matrix was subsetted according to the respective cell type followed by DGEA between cluster 1 (portal) and cluster 2 (central) with a log-fold change threshold of 0.01 and significance (p_{val_adj}) below 0.05. ”

Line [622 - 626]

“Three datasets were considered for this purpose. The scRNA-seq data set of cells originating from liver tissue from the Mouse Cell Atlas was (accessed 2020-10-06) 39 and the detailed gene expression data of annotated cell types from the single cell spatial reconstruction of mouse liver 13 and expression data on mesenchymal liver 27 was generously shared as a Seurat object by the authors of the respective publications ”

Line [836 - 846]

“We selected genes to be subjected to bivariate expression by distance analysis in two different ways. First, in the case of metabolic pathway gene markers for glucagon and Wnt targets, we extracted 12 known Wnt pathway markers 11 genes with most elevated expression levels in the central cluster (cluster 2) in the spatial data. For glucagon target we chose 10 known marker genes with the most elevated levels in the portal cluster (cluster 1) and 2 genes with highest up- or downregulation (*Mup20*, *Mdm2*) in glucagon deficient mice 45. Secondly, for the remaining bivariate expression by distance analyses of gene markers (Endothelial cells, Kupffer cells, HSCs, immune system process, ha-rasm chronic hypoxia, pituitary hormones), we selected 2-3 genes exhibiting most elevated expression levels for each, the central (cluster 2) and the portal cluster (cluster 1). These markers were identified as described in the Methods section

“Zonation based DGEA of cell type and metabolic pathway markers. ”

Review Figure 13 | Zonation and expression by distance of selected NPC markers. **a** Expression by distance of selected NPC markers of central endothelial cells (*Thbd*) and liver capsular macrophages

(LCMs) (*Sox4* and *Adamts2*) as well as portal endothelial cells (*Ltbp4*) and LCMs (*Ngfr*, *Tagln*) within 400 μm distance to each vein border. The blue line describes expression by distance from the border of the portal vein and the red line exhibits expression by distance from the border of the central vein. Ribbons around the lines indicate standard deviations of the smoothed curves. **b** Bivariate expression of genes the same as in **(a)** within 400 μm distance to the portal (y-axis) and the central vein (x-axis). Numbers in curly brackets after the gene name indicate that {1} the full model does not perform significantly better than the reduced portal model, {2} the full model does not perform significantly better than the reduced central model, {3} the full model does not perform significantly better than either of the reduced models to explain gene expression along the lobular axis, {4} the full model does not outperform the most reduced model (intercept only), i.e., the gene expression can be taken as constant across the tissue w.r.t vein distances. Relative expression values for each gene are depicted in a color gradient ranging from low (dark) to high (light). **c** Heatmap depicting differentially expressed endothelial cell markers [1] (left). Genes exhibiting highest expression elevation in either the central (cluster 2) or the portal area (cluster 1) are surrounded by a red box. These genes were selected to perform expression by distance (middle) and bivariate expression by distance (right) analysis as described in **(a)** and **(b)**. **d** Heatmaps of DGEA between cluster 1 (portal) and cluster 2 (central) of spatial data for markers plasmacytoid dendritic cells (pDCs) (left) and neutrophil (right) markers.

Review Figure 14 | Immune zonation exemplified by DGEA and expression by distance of Kupffer cell and immune system process markers. **a** Heatmap displaying DE Kupffer cell markers between cluster 1 (portal) and cluster 2 (central) (left). Markers with highest expression elevation in cluster 1 or cluster 2 are

surrounded by a red box. This gene selection was subjected to expression by distance analysis (middle) and within 400 μm of the vein border. The blue line shows expression by distance from the portal vein border while the red line shows expression from the central vein border. Ribbons around the lines indicate standard deviations of the smoothed curves. bivariate expression by distance analysis was performed within the same distance (400 μm) to the portal vein (y-axis) and central vein (x-axis) simultaneously (right). Numbers in curly brackets after the gene name indicate that {1} the full model does not perform significantly better than the reduced portal model, {2} the full model does not perform significantly better than the reduced central model, {3} the full model does not perform significantly better than either of the reduced models to explain gene expression along the lobular axis, {4} the full model does not outperform the most reduced model (intercept only), i.e., the gene expression can be taken as constant across the tissue w.r.t vein distances. Relative expression values for each gene are depicted in a color gradient ranging from low (dark) to high (light). **b** Heatmap displaying DEG genes associated with the GO term “immune system processes” (GO:0002376). Markers with highest expression elevation in cluster 1 or cluster 2 are surrounded by a red box (right). This gene selection was subjected to expression by distance analysis (left, top panel) and bivariate expression by distance analysis (left, right panel).

[1] <https://pubmed.ncbi.nlm.nih.gov/30222169/>

[2] <https://pubmed.ncbi.nlm.nih.gov/31722201/>

[3] <https://pubmed.ncbi.nlm.nih.gov/33239787/>

Minor comments:

1) Row 762 – what are the units of 210 and why was this threshold chosen?

Response:

We thank the reviewer for drawing our attention to the distances and chosen thresholds for these distances. To elaborate on the reviewer’s comment we provide more explanatory details on the *expression by distance* plot construction in the following:

“In this study, we set the distance threshold (T_N to 210). Having formed the neighborhoods, their associated expression profiles for a feature ($x_{N(t)}$) were assembled accordingly”

For the generation of expression by distance and the expression-based classifier we set distance thresholds from the borders of each respective neighbourhood. The original units refer to pixels from the high-resolution Hematoxylin and Eosin (H&E) images of the tissue and the images of the fluorescently labeled probes under the tissue after tissue removal. The process of image alignment is described in further detail in the materials and methods section:

Line [534 - 541]

“The staining, visualization and imaging acquisition of spots printed on the ST slides were performed as previously described [1]. Briefly, spots were hybridized with fluorescently labeled probes for staining and subsequently imaged on the Metafer Slide Scanning system, similar to the previous acquisition of the HE images. The previously obtained brightfield of the tissue

slides and the fluorescent spot image were then loaded in the web-based ST Spot Detector tool [2]. Using the tool, the images were aligned and the spots under the tissue were recognized by the built-in recognition tool. Spots under the tissue were slightly adjusted and spots under the tissue were extracted.”

As the pixel and the actual physical distances can be extracted from the fluorescent image of the probes in the spots under the tissue, the actual physical distances between spots can be converted from pixel to μm on the tissue.

To determine the neighbourhoods of each morphological structure we created masks covering all pixels belonging to each structure of interest.

Line [767 - 768]

“Using the brightfield H&E-images, a mask was created for each morphological structure. These masks covered all pixels considered to belong to the structure.”

The conversion of pixels to μm resulted in a conversion factor of 0.28. When generating the objects for the further expression by distance analysis, we stored all images (HE-images and respective masks) at 10% of their original size to keep necessary computing power for the downstream analysis minimal. Therefore the conversion factor was multiplied by a factor of 10, to reflect the actual distances in the tissue, resulting in a factor of 2.8. We would like to thank the reviewer for drawing our attention to these conversion factors, since we noticed the lacking multiplication by the scaling factor only during the revision process and have therefore changed Figure 2 and Figure 4 of the original manuscript accordingly:

Figure 3 | a) Enlarged view of a superimposed visualization of *Sds*, *Cyp2f2* expression in the portal vein module, consisting of selected DEGs of cluster one (supplementary table 1), all with high values around the histological annotation of a portal vein (top). Expression of *Glul*, *Cyp2e1* as representative marker-genes of the central vein module expression (supplementary table 1), consisting of DEGs of cluster 2 with high values around the histological annotation of a central vein (bottom).

b) Visualization of the average expression by distance to vein-type measured within 50 μm from the vein. The top row shows expression by distance of portal markers *Sds*, *Cyp2f2*, *Hal*, *Hsd17b13* and *Aldh1b1* to portal veins in blue and central veins in red, while the bottom row shows distances of central vein markers *Glul*, *Oat*, *Slc1a2*, *Cyp2e1* and *Cyp2a5* to portal veins in blue and central veins in red (top panel). **c)** Visualization of influence of distance to both vein types on expression by bivariate expression by distance plots (see methods for details). Gene expression values are depicted in a gradient from low (dark) to high values (light). The distance of each gene to central veins between 0 and 400 μm is represented on the x-axis. Simultaneously, distances to portal veins for the same distance are depicted on the y-axis for each

gene. High values in the bottom right corner indicate gene expression is predominantly observed close to portal veins and far from central veins, while high values in the upper left corner indicate the reverse observation (below graphs). **d)** Visual histological annotations (left) of central (red) and portal (blue) veins, including ambiguous visual annotations (green), compared with computational prediction, using the 10 marker genes from 3b (right). The classification of vein types is based on a weighted (by distance) average expression of the genes' expression profiles in the neighborhood of each vein. In addition, the spatial expression data of spots neighboring uncertain morphological vascular annotations (green) can be used to deduce periportal or pericentral vein-types in the cases where visual annotations are ambiguous. **e)** Expression by distance of portal - (top panel) and central - (bottom panel) markers. Probabilities for each class (central and portal) can be extracted from the logistic regression model, here given as $P(\text{central})$ or $P(\text{portal})$ (scale bar indicates $500\mu\text{m}$).

Consequently, we also changed the distance threshold in the plots from 210 pixels (which referred to $50\mu\text{m}$ in the original plots) to 142 pixels (referring to $400\mu\text{m}$ in the revised plots). This threshold of 210 pixels was originally chosen since it referred to the radius of one capture region and showed the biggest difference in expression along the lobular axis. The revised threshold was chosen because it represents the longest distance between adjacent spot centers in the same row. In theory this threshold would depict the expression between the two furthest neighbouring spots, if the border of the vein would always go through the spot center (Review Figure 15).

Review Figure 15 | Schematic of spot-distance for lowest chosen distance threshold in expression by distance plots. Spots on ST arrays are arranged in a hexagonal pattern. The longest distance between spot-centers in the same row measures approximately $200\mu\text{m}$, referring to 71 pixels (px) in the image files, used for image alignment. Measuring the distance to two adjacent spots in the same row therefore results in a distance of approximately $400\mu\text{m}$, the chosen threshold for *expression by distance* analysis for veins.

We revised the method section in the original manuscript to correct the pixel threshold and added the unit of the threshold so that it now reads:

Line [862 - 864]

“In this study, we set the distance threshold (T_N to 142 pixels). This threshold refers to 400 μm and represents the longest distance between adjacent spot centers in the same row.”

[1] <https://pubmed.ncbi.nlm.nih.gov/31501547/>

[2] <https://pubmed.ncbi.nlm.nih.gov/29360929/>

2) Figure 3b – the humps in Figure 3b are not real (e.g. see smFISH validations for genes such as *Cyp2e1* and *Cyp2f2* in PMID 28166538, these are clearly monotonic). The authors should consider a computational method to remove these artefacts.

We appreciate the reviewer pointing out this flaw in our previous attempt to illustrate the combined influence of the portal and central vein distance, we fully understand - and agree - that the “humps” easily can lead to erroneous conclusions. We have therefore replaced the log-ratio plots with the bivariate expression by distance plots (based on multivariate regression), as suggested by the reviewer in Major Comment 1. These plots are not burdened with the same introduction of artificial signals, and - we believe - better represent the information we sought to convey. We hope that the reviewer agrees with us and is satisfied with the changes.

3) *Lyve1* is considered in the manuscript a marker of lymphatic endothelial cells, is it really distinct from endothelial cells? Please examine the correlation with *Cdh5*.

Response:

We thank the reviewer for drawing our attention to the use of *Lyve1* as a lymphatic endothelial marker.

As correctly observed by the reviewer, *Lyve1* is reported as a lymphatic endothelial cell marker in the literature but also as a marker for midlobular endothelial cells [1-4]. However, we consider it expressed more rarely, i.e. in a subset of endothelial cells compared to the common endothelial cells with *Cdh5* as a common marker. We would also like to emphasize that the data presented in our study is not suitable to establish distinct cell type annotations, since each capture location (spot) consists of a small mixture of cells, but rather to explore marker gene expressions of cell types across liver tissue.

To further elucidate whether we can support the assumption that *Lyve1* expression is distinct from the expression of the endothelial cell marker *Cdh5*, we performed two additional analyses according to the reviewer’s suggestion:

1) To see whether the relationship between *Lyve1* and *Cdh5* expression is monotonic, we performed a two-sided Spearman rank test between *Lyve1* and *Cdh5* across the spots under the tissue. Spotwise correlation between these two genes resulted in correlation of 0.043 (p-value = 0.0032), indicating that the expression of one gene is marginally positively correlated with the expression of the other.

2) Additionally we wanted to assess whether a count of *Lyve1* larger than zero (indicating expression of *Lyve1*) is independent of *Cdh5* being zero (indicating no expression of *Cdh5*) or larger than zero (indicating expression of *Cdh5*) and vice versa. Therefore, we created a contingency table of sum for spots (Review Table 4) showing expression of *Cdh5* and *Lyve1* and performed a Fisher's exact test. The resulting p-value of 9.635×10^5 (raw count data, count threshold > 0), accepts the null hypothesis that *Cdh5* and *Lyve1* are expressed independently.

Review Table 4: Contingency table for *Lyve1* and *Cdh5* expression across spots.

Gene expression	Cdh5		
	Spot sum	0	> 0
Lyve1	0	2760	1399
	> 0	413	291

These results indicate that expression of *Cdh5* and *Lyve1* is independently expressed across spots and exhibit a weak monotonic relationship. Therefore, *Cdh5* and *Lyve1* expression coincides by random chance within the same capture region. If the expression of both genes would represent a characteristic signature for one specific cell type, we would expect a stronger correlation between the investigated genes. Thus, we conclude that *Cdh5* and *Lyve1* are not characteristic markers for the same cell type.

Based on the obtained results we changed the description of *Lyve1* as a marker for a subset of endothelial cells to the following in the main manuscript:

Line [153 - 155]

“Lymphatic liver endothelial cell and liver midlobular endothelial cell marker Lyve1^{30–32} showed expression in a smaller fraction of 698 spots (~14%).”

We hope the reviewer agrees with our adjusted interpretation of *Lyve1* and *Cdh5* expression across the liver and is convinced of our argumentation.

- [1] <https://pubmed.ncbi.nlm.nih.gov/17626278/>
- [2] <https://pubmed.ncbi.nlm.nih.gov/33340713/>
- [3] <https://pubmed.ncbi.nlm.nih.gov/30027142/>
- [4] <https://pubmed.ncbi.nlm.nih.gov/11278811/>

4) For the deconvolution the authors may want to consider other papers establishing cell-type references from single cell datasets that had more comprehensive coverage of liver NPCs compared to Tabula Muris, e.g. PMID 31398325, PMID 30222169 and PMID 31722201.

Response:

We fully agree with the reviewer that it's of interest to explore other single cell data sets that would offer a more comprehensive coverage of the cell types we are interested in, this is actually something we wanted to do already before the review process. We also appreciate that the reviewer compiled a list of suggested publications that may contain relevant single cell data.

After contacting the corresponding authors of all the three suggested publications, two of them granted us access to the necessary expression and metadata (clustering results), these being PMID 30222169 [1] and PMID 31722201 [2]. Using the two provided data sets we could map each of them onto our spatial transcriptomics data, according to the procedure described in the Materials and Methods section (original manuscript). A summary of the results, in the shape of correlation matrices, from each of these mappings are shown in Review Figure 16.

We would however like to explain why we did not include these data sets in our initial submission, nor feel inclined to do so in this second iteration. The issue here lies not in the quality of the single cell data, but rather in its composition. To elaborate, in most probabilistic methods that are designed to use single cell data in order to deconvolve/decompose spatial transcriptomics data, one tries to explain the *observed* spatial gene expression using the expression profiles learnt from the single data. In short, what is meant with "learning expression profiles for single data", is to *associate* a certain expression profile to each of the cell types/states present in our single cell data, for example by learning the parameters for a statistical distribution that describes the gene expression.

Of course, we rarely expect a perfect match between the single cell and spatial data in terms of which cell types/states that are present in the single cell data, but we operate with the assumption that the single cell data to some extent is *representative* of the cell type population in the spatial data. This assumption reduces the problem of spatial data decomposition into a much simpler one, namely to find the combination of cells (from the cell types/states defined in the single cell data) that most likely generated the spatial expression data. Violations of this assumption are to some extent accepted, the *stereoscope* method for example accounts for asymmetries between the two modalities by introducing an artificial cell type that (with some restrictions) adapts to the data. Still, if the discrepancies between the cell type/state population in the single cell and spatial transcriptomics data is too large, there is a risk of generating unreliable results since one tries to explain gene expression using missing or incorrect cell

types. As an analogy, this would be similar to an attempt of recreating a painting using a palette with different or missing colors compared to that of the original artist.

It is based on this desire to have relatively well-matched data, that we argue that none of the two data sets that we were given access to [1,2] constitute as good of a representation of the liver cell composition as the Mouse Cell Atlas (MCA). Notable, [2] only lists 3 cell types in the metadata while [1] hosts a slightly larger number of 7 cell types. In addition to having fairly broad cell type labels, both these data sets - in contrast to MCA - lack important cell types such as: Hepatocytes (zoned and non-zoned) as well as additional immune cells such as macrophages. Especially the lack of hepatocytes makes the results in the single cell integration unreliable, as they are the most abundant cell type in the liver, by numbers as well as by volume.

We hope that, in the light of the above discussion, the reviewer finds it justifiable to not include the proposed data sets. Still, we would like to emphasize that we recognize the reviewer's attempt to help us present a more holistic view of the spatial cell type organization in our data, and are thankful for it.

Review Figure 16 | Visualization of cell type co-localization by Pearson correlations. **a** Depiction of cell type co-localization in liver tissue based on stereoscope integration of single cell data set from [1]. Positive correlation values indicate spatial co-localization of cell types while negative values represent spatial segregation. **b** Co-localizations of cell type proportions annotated in [2], interpreted the same way as in **(a)**.

[1] <https://pubmed.ncbi.nlm.nih.gov/30222169/>

[2] <https://pubmed.ncbi.nlm.nih.gov/31722201/>

5) Line 117- Something in these numbers does not add up, how can each spot contain 30 cells, yet only 5-10 of them are hepatocytes? Hepatocytes take up 80% of liver mass but 60% of the number of cells. Please check.

Response:

We fully agree with the reviewer that it is necessary to investigate the number of cells - in particular hepatocytes - within one spot in further detail. We would like to refer the reviewer to the histogram, showing the number of cells per spot across the tissue, which we performed upon request of reviewer 3 (Review Figure 18). These results show that the majority of spots are estimated to contain between 30 and 60 cells. Based on the reviewers comment we sought to determine the number of hepatocytes in comparison to other cell types in our data.

To this end we performed additional sectioning and DNA staining of frozen livers used for our ST experiments. Manual counting of stained cells resulted in an average of approximately 42 cells per spot (110 μm diameter). However, the imaging quality of cryo-preserved tissue sections used for ST experiments remains inferior to e.g. imaging quality paraffin preserved tissue sections. Thus, in tissue sections from ST experiments it remains challenging to differentiate hepatocytes from other non parenchymal cells (NPCs) with absolute certainty. However, to investigate the number of hepatocytes compared to NPCs further we also performed manual counting on paraffin embedded liver sections of slightly younger (5 weeks old) female mice. We observed between 14 and 40 cells per 100 μm diameter spot, of which 60-70% were hepatocytes. Only around regions of the portal vein hepatocytes constituted about 50% of total cells.

The results obtained from the additional analysis on the number of hepatocytes confirm and agree with the reviewer that the estimation of the number of hepatocytes should be adjusted to the observed percentages reported here. Hence, we adjusted the respective section in the original manuscript to the following:

Line [119 - 122]

“Each spot is covered by a small mixture of liver cells. From the hematoxylin-stained nuclei we estimated that a majority of spots contain between 30-50 cells, of which 60-70% are considered to be hepatocytes.”

We hope the reviewer is satisfied with the performed adjustments and agrees with the performed changes to the manuscript.

Reviewer #2

With their work entitled Spatial Transcriptomics to define transcriptional patterns of zonation and structural components in the liver, Hildebrandt et al make an important contribution to the basic understanding of the zonation of the liver via using Computational models. By applying the latest methods in the field of systems biology using spatial transcriptomic analysis, the authors were able to show a very deep insight into the expression signature depending on the spatial allocation of hepatocytes using cryosections of liver material. The assignment of structural functions to specific regions in the parenchyma is particularly interesting. However, the work also has some small weaknesses that I think should be taken into account in a Revision process.

1) Discuss dimorphism and influence on data and prachymal morphology
liver tissue from female mice was used as material. This fact is not included in the discussion of the results. Since it is known that the dimorphism of the liver can also have a strong influence on the zonation (also on the morphology within the parenchyma), this would be important to discuss.

The reviewer draws attention to the very interesting and relevant point of the influence of sexual dimorphisms in liver tissue. As the reviewer highlights, previous transcriptomics studies discussed the influence of sexual dimorphisms on zonation, e.g. [1]. Thus, we agree with the reviewer that we should include the discussion of the possible influence of sexual dimorphism, in light of the fact that we are only investigating material of female mice in our study. Still, albeit interesting, the main focus of our study is not to investigate potential differences in spatial gene expression and zonation between male and female individuals. Therefore, we addressed the reviewer's request by emphasizing the fact that we exclusively investigated female liver material, while providing future implications for the potential of using Spatial Transcriptomics to studying the impact of sexual dimorphisms on metabolic zonation in liver tissue:

Line [99 - 100]

“Here, we perform ST on female mouse liver tissue sections, assessing spatial factors contributing to spatial liver heterogeneity at the transcriptional level.”

Line [114-115]

“We used a total of 8 sections of wild type adult, female mouse livers from the caudate and right liver lobe for histological staining, library preparation and sequencing.”

and

Line [453-457]

“This study constitutes a compelling initial exploration of the benefits that spatial transcriptomics provides for studies of the liver and we consider it a valuable data resource for the hepatology field. We further anticipate that ST will be highly beneficial for future studies addressing liver development, sexual dimorphisms of liver zonation, immunity and general pathology in the mammalian liver, including humans.”

We thank the reviewer for his comment to complement our discussion. We hope the revised discussion addressed the reviewer’s comment sufficiently by highlighting the limitations of our study to address spatial sexual dimorphisms. We anticipate that our study will provide a valuable resource for future studies addressing this research question, which we consider important to mention in this context.

[1] <https://pubmed.ncbi.nlm.nih.gov/23791742/>

2) Validation of cluster 5 structure (comparison with single cell data)

it would be good if the authors would carry out some additional experiments to confirm some of their results with regard to the uncharacterised structures found.

We have conducted further computational experiments based on the reviewer’s suggestion to support our characterization of cluster 5. For an in-depth statement on the reason why additional laboratory experiments on the liver sections used in our study are inaccessible, we would like to refer the reviewer to our response to comment 9 of reviewer 3. However, peer-reviewed and elaborate scRNA-seq studies, including thorough orthogonal validations of their findings, are already available. Therefore, we considered it appropriate to perform additional computational experiments to address the reviewer’s comment.

Well annotated and extensive single cell data sets of healthy liver mesenchyme [1] and liver endothelial cells (LECs) [2] were generously shared with us by the respective authors, enabling us to perform additional single cell comparisons with the spatial data of cluster 5, as suggested by the reviewer.

To the extent of comparing annotated single cell data with the uncharacterised structure identified in our spatial data, we first extracted the intersection of marker genes of cluster 5 (Appendix Table 8) and all marker genes of each annotated cell type in [1] and [2]. As expected, the marker gene signature of cluster 5 did not consist of a signature that can be attributed to one single cell type, since each spot consists of a small mixture of cells. However, some cell type marker genes are more highly represented in cluster 5 (e.g., fibroblasts of liver capsular macrophages (LCMs) than others (e.g., T cells, Kupffer cells, neutrophils or plasmacytoid dendritic cell (pDCs) (Review Table 5).

Review Table 5 | Intersection of marker genes of cluster 5 and annotated NPC types and mesenchymal cell types. The list of cluster 5 markers from the ST data (yellow) was compared to markers of annotated cell types from [1] in blue and [2] in green. Shared markers between cluster 5 and cell types of [1] and [2] are marked with a cross in the respective cell.

	PMID: 30222169							PMID: 31722201		
Gene	Endothelial cells	T cells	Kupffer cells	B cells	LCMs	pDCs	Neutrophils	HSCs	Fibroblasts	VSMCs
Gsn						X			X	
Dpt									X	
Mgp									X	
Col1a1									X	
Tagln										X
Col3a1									X	
Vim									X	
Col1a2									X	
H2-Eb1				X	X					
Crip1					X					X
Acta2										X
Ahnak									X	
Tmsb4x					X					
Timp2									X	
Dcn								X		
H2-Aa				X	X					
Lum									X	
H2-Ab1				X	X					
Cd74				X	X					
Igfbp7	X							X		
Sparc	X									
Bgn	X							X		
Col14a1								X		
Spp1										
Timp3	X								X	X
Txnip									X	

Next, we wanted to expand further on the degree to which individual marker genes of cluster 5 were expressed in the annotated cell types in the single cell data. To this end, we visualized the 12 marker genes of cluster 5 with the highest log-fold change in the scRNA-seq t-SNE embeddings conducted in [1] and [2] (Review Fig 17a and 17b). Our results show that most cluster 5 markers, if expressed in the liver endothelial cell data set generated by Halpern et al., exhibit the highest values in LCMs or B cells (e.g. *H2-Eb1* and *Crip1*), while all 12 genes show relatively high expression in the annotated cell types in the mesenchyme single cell data set by Dobie et al. Especially high expression can be observed for *Tagln*, *Crip1* and *Acta2* in vascular smooth muscle cells and *Gsn*, *Dpt* and *Vim* in fibroblasts.

In summary, marker genes of cluster 5 are expressed to varying degrees in annotated cell types of both datasets and share markers with cell types of at least one of the datasets, with the largest overlap in LCMs and B cell in liver endothelial cells and fibroblasts, vascular smooth

muscle cells and hepatic stellate cells in the liver mesenchyme. These results indicate and support that cluster 5 indeed consists of a mixture of cells which constitute the environment of the mesenchyme and/or liver capsular environment.

We believe this addition contributed significantly to our ability to interpret the nature of the uncharacterised structure underlying cluster 5. Consequently, we are including the results depicted in table 5 as Supplementary Figure 18 of the original manuscript and modified the text in the manuscript accordingly:

Line [340-345]

“Moreover, we compared markers of cluster 5 with markers of annotated cell types of scRNA-Seq data and found high overlap with mesenchymal cell types (fibroblasts, HSCs and vascular smooth muscle cells (VSMCs) (Supplementary Figure 20).

Taken together, correlation analysis of cluster 5 markers, histological morphology in the respective tissue area and high overlap with mesenchymal cell markers advocates for the spatial organization of cluster 5, independent of liver zonation.”

and

Line [419-421]

“Cluster 5 consists of a small number of spots with distinct spatial localization, which exhibit expression of mesenchymal cell marker genes^{13,27} and are associated with “collagen fibril organization” pathways.”

a

b

Review Figure 17 | Expression of cluster 5 marker genes displayed in t-SNE embeddings of annotated scRNA-seq data. **a** Projection of the 12 DE markers of cluster 5 exhibiting the highest log fold-change in t-SNE embedding of the annotated NPC types described in the paired-cell sequencing from Halpern et al. [2]. **b** Projection of 12 markers of cluster 5 with highest log fold-change in t-SNE embedding of the annotated mesenchymal cell types described in [1].

[1] <https://pubmed.ncbi.nlm.nih.gov/31722201/>

[2] <https://pubmed.ncbi.nlm.nih.gov/30222169/>

3) Discuss relevance for experiments on human material

it would be important to know whether this method could also be applied to human material in a relatively high throughput manner. Such a statement would certainly encourage other researchers in the field of liver zonation to apply this method to answer still open questions.

We are pleased about the interest of the reviewer in the possibility to apply spatial transcriptomics to investigate human material. In his/her comment he/she additionally inquires about the possibility to implement the method in a relatively high-throughput manner.

To answer the first part of the question we would like to first confirm that ST can be applied to human material and has already been performed in multiple human organs [1-10]. A recently published pre-print even includes spatial transcriptomics data of the human liver [11].

To further extrapolate on the possibility of applying the method in high-throughput, we would like to briefly reiterate at the current technical limitations of the method: These include the number and size of the sub-arrays on each ST slides (6 subarrays à 6,4 x 6,4 mm, see Methods and [12]). With the introduction of commercially available experimental kits, we anticipate an even broader spread of the technique to a more diverse set of laboratories that could apply this to various tissue types and address a high variety of biological questions [13]. In addition, we would like to highlight the currently high costs of performing ST experiments, which may further complicate the performance of ST experiments of human material at high throughput levels, given the aforementioned current technical limitations of the method.

To address the reviewers remark on the applicability, specifically the future applicability of the method presented in our study on human liver material, we modified the manuscript text to the following:

Line [455 - 457]

“We further anticipate that ST will be highly beneficial for future studies addressing liver development, sexual dimorphisms of liver zonation, immunity and general pathology in the mammalian liver, including humans.”

[1] <https://www.sciencedirect.com/science/article/pii/S0092867419312826>

[2] <https://science.sciencemag.org/content/364/6435/89>

[3] <https://www.biorxiv.org/content/10.1101/2020.07.14.200600v1>

[4] <https://www.nature.com/articles/s41551-020-0578-x>

[5] <https://pubmed.ncbi.nlm.nih.gov/32579974/>

[6] <https://www.nature.com/articles/s41598-019-55441-y#Sec20>

[7] <https://cancerres.aacrjournals.org/content/78/20/5970.long>

[8] <https://www.nature.com/articles/s41467-018-04724-5>

- [9] <https://www.nature.com/articles/s41598-018-27627-3>
[10] <https://www.nature.com/articles/s41598-017-13462-5>
[11] <https://doi.org/10.1101/2021.03.27.436882>
[12] <https://pubmed.ncbi.nlm.nih.gov/27365449/>
[13] <https://www.10xgenomics.com/products/spatial-gene-expression>

Reviewer #3

This paper performed the spatial transcriptomics on a total of 8 sections of wild type adult mouse livers consisting of 19,017 genes across 4,863 spots (each covering 5-10 hepatocytes and up to 30 cells), to study the zonation of liver lobules. In this paper, the unsupervised clustering on spot-level gene expression revealed six unique clusters, where two correspond to periportal and pericentral regions, respectively, and one is claimed to be a previously uncharacterized liver structure. The further cell type identification using scRNA-seq-derived cell type signatures found the co-localization of multiple cell types, but differing from the previous scRNA-seq-based observation, the liver is found to be predominantly constituted by zoned hepatocytes. Moreover, authors observed the dependency between spatial distance and gene expression along the lobule axis, which further motivated the computational prediction of portal and central veins from spatial gene expression.

This study is non-trivial. However, the low resolution (non-single cell level) of spots is the major concern, which may result in suspicious conclusions when conducting all clustering, biological interpretation and cell type identification. Further, this paper lacks in-depth analysis and validation, and some of the observations are over-interpreted, making the result and conclusion less solid. Followed are the detailed comments.

Before addressing the reviewer's comments in depth below, we would like to start by thanking the reviewer for correctly pointing out the current lack of single cell resolution of the ST method. We would like to clarify that our ambition is not to put forth ST as a method to challenge scRNA-seq, but rather to highlight the benefits of applying ST to liver tissue, which can be largely attributed to the spatial component this method provides. Further, we describe ST and scRNA-seq as two highly complementary methods, which we believe benefit highly from being analyzed in tandem.

In our study we are not attempting to perform cell type identifications across the tissue, as the current resolution of ST does not allow us to make such interpretations. We rather provide a - in our opinion - strong complementary method to scRNA-seq, as we retain spatial information of small mixtures of cells across tissue and exemplify how this data can be combined with pre-existing scRNA-Seq data.

Although ST currently lacks the resolution of analysis down to the single cell level, we strongly believe that ST represents an important tool for transcriptomics analyses of liver and other

tissues in general, and an appropriate method to answer the research questions posed in this study. It is our firm opinion that the analyses of ST data performed in this study provide reliable clustering results and biological interpretations, which we are addressing in more detail in the reviewer's comments below. We realize now that certain parts of the manuscript could have been more clear in highlighting the fact that we are not investigating gene expression at the single cell level. We apologize for these unclarities and have provided clarification by implementing the following modifications in the manuscript:

Line [70 - 73]

“However, all previous studies either performed laser capture microdissection¹⁵ or perfusion techniques^{11,13}, ultimately requiring tissue dissociation prior to sequencing, resulting in single cell resolution but also altering the physiological transcriptional landscape¹⁶⁻¹⁸.”

Line [92 - 95]

“Hence, the generation of Spatial Transcriptomics data from liver sections in their bona fide tissue context, together with pre-existing knowledge of liver zonation enables spatial annotation of structures consisting of small mixtures of cells in the liver microenvironment (lobule) and liver macroenvironment (tissue section).”

Line [108 - 110]

“We anticipate that our results from small mixtures of cells, complement previous findings of different cell types constituting the overall transcriptional landscape of liver tissue and enhance our current understanding of liver tissue organization.”

Line [360 - 363]

“Here, we estimated cell type information in the spatial data in two different ways. First, we assessed expression of characteristic marker genes within a wide range of expression levels and investigated zonation patterns of established cell type markers^{13,27}. Secondly, we deconvolved gene expression profiles of the mixed cells in spots using stereoscope.”

We hope the reviewer regards these modifications as appropriate and that they are already able to resolve some of the reviewer's concerns, which we are addressing carefully in response to the detailed comments below.

1) Low cell resolution of spots:

The technical artifacts of “doublet” in scRNA-seq leads to suspicious biological conclusions, while this phenomenon is much more severe in the spatial transcriptome of this paper, where

“each spot contains between 5-10 hepatocytes and up to 30 cells in total per spot”. It may largely bias the unsupervised clustering, resulting in suspicious cluster interpretation.

Response:

We appreciate the reviewer’s comment and understand his/her worry regarding the presence of doublets, or rather “multiplets”, in our spatial transcriptomics data given how this is often listed as a common source of unwanted artifacts in single cell analysis. However, we would like to elaborate on why we do not believe that this is an issue when conducting our data analysis which is inherently different from scRNA-seq data and ask the reviewer to consider our analysis in the light of these arguments.

First, for the sake of context, we see it fit to recapitulate on why doublets are necessary to remove in single cell data. The very premise of single cell RNA-seq is that each data point represents the transcriptome of an individual cell, and if the concentration of cells in the droplet loadings is correctly adjusted such is the case for a majority of the observations, meaning doublets are *rare* occurrences deviant from the majority of the collected data points. If the doublets are homotypic (host two cells of similar states and types), they aren’t of too much concern as the captured transcripts would be uniformly sampled from both cells thus being near equivalent to sampling from a single cell of said type. In contrast, heterotypic doublets (containing cells of different states and/or types) will give a transcriptional profile that is a hybrid of the two which would emerge as a potentially new (relatively rare) cell state/type. Such heterotypic doublets may cause issues in several downstream analyses such as clustering and differential gene expression analysis, or even in the initial stages of dimensionality reduction if their abundance is high enough, as seen in [1,2].

While single cell RNA-seq and spatial transcriptomics data share many similarities on a superficial level, one key difference is how we **expect** and assume that each observation in the latter consists of contributions from multiple cells, as the captured transcripts are sampled from all cells covering the spot (capture location). This means that observations that represent a mixture of multiple cells are actually the rule rather than the exception. Next, when one analyzes the spatial transcriptomics data, we **do not seek to identify groups of observations (clusters) that represent cell types or states**, instead the aim is to find observations that seem to exhibit similar expression profiles, indicating that these have a **similar composition of cell types**. Of course, the clusters that emerge from the analysis of spatial transcriptomics can’t be taxonomically indexed with the same ease as cell types in single cell data, but they still represent biological entities that are informative of the tissue being studied, and allows one to identify transcriptionally similar regions in the tissue.

Furthermore, we also make sure to regress out the total number of UMI’s observed in each spot using the SCTransform, which will account for differences in cell abundance at each spot and potentially more transcriptionally active cells. Also, in [3] the authors describe how the total UMI count is largely dependent on the number of cells present in a spot, which suggests that

regressing out the former should account for any bias that could be expected by the latter. We will nevertheless pursue this question of the cell count's influence on the downstream analysis more in the answers below.

As a final comment, we would like to add that the Spatial Transcriptomics (ST) method we are using has been published in a peer-reviewed journal and featured in multiple publications after this, and its successor is now a commercial product sold in large masses, attesting to the validity of the results that emerge in studies based on this data [4,5,6,7].

We hope that these comments have clarified our stance and relieved the reviewer of his/hers worries about the potential negative impact of the mixed data. If we have failed to do this, we ask the reviewer to be more specific in exactly how he/she believes the mixed character of our data would confound the downstream analysis, so that we can address these worries in a more targeted manner.

[1]: <https://www.sciencedirect.com/science/article/pii/S2405471220301952>

[2]: <https://www.sciencedirect.com/science/article/pii/S2405471219300730>

[3]: <https://academic.oup.com/jmcb/article/12/11/906/5861536>

[4]: <https://academic.oup.com/jmcb/article/12/11/906/5861536>

[5]: [https://www.cell.com/cell/pdf/S0092-8674\(19\)31282-6.pdf](https://www.cell.com/cell/pdf/S0092-8674(19)31282-6.pdf)

[6]: [https://www.cell.com/cell/pdf/S0092-8674\(20\)30672-3.pdf](https://www.cell.com/cell/pdf/S0092-8674(20)30672-3.pdf)

[7]: <https://www.10xgenomics.com/products/spatial-gene-expression>

1a) cell count frequency/spot

As the very first step, the authors are suggested to demonstrate the frequency of cell count in spots, e.g. histogram or density plot.

We welcome the suggestion of the reviewer to visualize the cell count across spots in the tissue, as an approach to address the concern of the potential influence of the cell count on the observed expression profiles of each spot. As previously stated, each spot consists of multiple cells of various cell types. Hence, spots are expected to contain differing amounts of cells, which is largely influenced by the cell type composition within the spot [1].

In order to quantify distribution of cell counts across the ST spots we developed a segmentation workflow using image processing and analysis tools available from the EBImage R package [2]. The segmentation workflow is largely inspired by tutorials provided by the EBImage package developers and is available as a command line tool on GitHub [3]. A summary of the workflow is described below:

First, an RGB encoded image of an H&E stained tissue section (Review Figure 18 a) is imported and converted to grayscale by merging the red and blue color channels (these color

channels were found to be the most informative to delineate cell nuclei based on visual inspection). Intensity values in the grayscale image are then reversed, resulting in a representation where nuclei appear bright on a dark background (Review Fig 18 b). Segmentation of nuclei is conducted on the inverted image using adaptive thresholding, a method that compares the original image with a filtered version and returns a binary image. In this binary image, nuclei are represented by a value of 1 and the background by a value of 0 (Review Fig 18 c). From the binary image, merged nuclei were split and labelled using a watershed transformation (Review Fig 18 d). At this step, a nucleus is represented by a set of connected pixels sharing the same label.

To quantify the number of nuclei per ST spot, we first calculated the pairwise distances between ST spot centers and nuclei centroids. Then, for each spot we counted the number of nuclei within a distance of 50 microns, corresponding to the radius of an ST spot (Review Fig 18 e and f). Review Fig 18 g depicts an example of the cell count distribution across a whole liver tissue section. For a more detailed description of the segmentation workflow and cell counting, we refer to the documentation in the markdown notebook provided on GitHub.

Review Figure 18 | Visualization of cell segmentation workflow. The original H&E image in **a** is first converted to grayscale and inverted resulting in **b**. **c** The binary image created using adaptive thresholding to distinguish nuclei in white and the background in black. **d** Each nuclei is labeled in color using watershed transformation. **e** shows the ST spots projected on the HE image (black circles) to identify which nuclei should be considered for the cell counting. **f** depicts the spot spot selection of **e** in the inverted binary image, where positions that are outside of spots are converted to black background color. **d** represents an example of the resulting cell count distribution with the original H&E image to the left and the corresponding cell count distribution to the right, where the number of cells are indicated by a color gradient from light (low cell count) to dark (high cell count).

To ensure that our segmentation workflow produced reliable results, we compared our approach with a peer-reviewed and published cell segmentation method relying on machine learning (Ilastik + Fiji) [4] (Review Figure 19), and found a high correlation (Pearson) between the results.

Review Figure 19 | Pearson correlation of different cell segmentation and cell count distribution methods. The machine learning based cell segmentation tool (Ilastik + Fiji) (y-axis) shows a strong correlation (0.96) with the segmentation method designed in the results presented here (EBImage) (x-axis).

This cell segmentation approach resulted in an estimation of the number of cells for each spot, which we then used to generate a histogram depicting the frequency of the cell number per spot across the tissue.

We observed the highest frequency for spots containing between 30 and 60 cells, followed by spots containing between 10 - 20 or > 60 cells (Review Figure 20). We expect most spots to have a similar amount of cells. Nonetheless biliary ducts, venous structures and other potentially uncharacterised structures and/or smaller cells of the same type may result in a higher cell density within some spots. Consequently, the same principle applies to spots which might contain larger cells, resulting in smaller count of cells in the respective spot.

To further answer the reviewer's comment, we consider it important to note that the tissue integrity can vary between different tissue sections as seen in H&E images (Review Figure 21). Other technical artifacts might also be present in the HE images, for example folds, bubbles and

cracks. All such technical artifacts increase the risk of biases in the cell count estimates and further support that the inclusion of cell counts in the normalization procedure can introduce an unwanted source of technical noise.

Review Figure 20 | Histogram depicting the frequency of cell numbers across spots under the tissue for all analyzed samples. The number of counted cells per spot, received by the cell segmentation tool designed for this study is shown on the x-axis. The frequency of cell counts per spot across the tissue is shown on the y-axis, showing that more than 1000 spots across tissue sections contain 40 or 50 cells.

a

sample 1 (caudate lobe)

b

sample 3 (right lobe)

Review Figure 21 | Comparison between HE stainings of different sections used for ST experiments. Representative subsets (100x100 pixels) of H&E images of two different sections used in the ST experiment to exemplify differences in staining quality between different sections. Due to technical differences during sample preparation, the cellular cytosol can look more pitted and hollow (a), or more even (b).

We hope we have addressed the reviewer's request to demonstrate the number of cells counted in individual spots sufficiently by:

- 1) implementing a method to assess the number of spots.
- 2) Visualizing the frequency of cells per spot across the tissue in a histogram.

[1] <https://academic.oup.com/jmcb/article/12/11/906/5861536>

[2] <https://bioconductor.org/packages/release/bioc/html/EImage.html>

[3] https://github.com/ludvigla/liver_cell_segmentation

[4] <https://doi.org/10.1093/jmcb/mjaa028>

1b) Investigate the influence of cell count on clustering

It is crucial to investigate whether the cell count in spots can influence the clustering. The authors are suggested to overlay the cell count in spots on the spot clustering. It is expected that the clusters with small cell count may give the more confident and accurate cluster interpretation, compared to those with high cell count, which are of the average expression of multiple cell types.

Response:

The reviewer highlights the legitimate concern about the influence of a variety of factors on clustering of sequencing data, including the number of cells per spot. We consider it beneficial to briefly reiterate on the normalization and clustering approach performed in this study. First the data was normalized to account for potential batch effects of different sections and sequencing depth for each experiment. This step was performed for all sections of each experiment (referring to one ST slide). Next, we performed data integration for all three experiment datasets using CCA (canonical correlation analysis). CCA finds a gene correlation structure that is conserved between datasets and aligns these datasets into a low-dimensional space represented by a set of correlation vectors [1]. We then used the canonical correlation vectors to perform shared-nearest-neighbor (SNN) graph based clustering. This approach, described in further detail in the materials and methods section of the original manuscript, resulted in 6 clusters across all investigated liver sections.

To address the reviewer's concern about the specific influence of the cell count we performed additional analyses, starting with his/her suggestion to overlay the cell count in spots on the spot clustering. To this end, we investigated the distribution of cell counts across spots, derived from the cell segmentation described in the response to comment 1a) across each cluster and the dimensionality reduction projection of spots (UMAP). Our results exhibit a uniform distribution of spots with different cell counts across all identified clusters. This indicates that spots do not cluster based on the number of cells present within spots (Review Figure 22 a,b).

Review Figure 22 | Distribution of cell counts across clusters. **a** depicts the distribution of cell counts (ranging from 0 - 150 on the y-axis) grouped by cluster (0-5) (x-axis). The color for each cluster corresponds to the respective color in the UMAP projection in the main manuscript. **b** Depiction of the number of spots across clusters in the UMAP embedding with cell count values ranging from 0 (low cell count, light) to 150 (high cell count, dark).

[1] <https://www.nature.com/articles/nbt.4096>

1c) Follow up on question 1b

To further reduce the bias, authors are suggested to also directly use the spots with low cell count to conduct clustering, to see how many clusters can be made, followed by cell type identification and biological interpretation.

We appreciate the reviewer's comment to address his/her concerns about the influence on cell count on the downstream clustering of our data.

To answer the reviewer's request we first would like to emphasize again that our spatial data is not equivalent to scRNA-seq data as each observation (spot) represents a composition of multiple cells and thus various types. The results requested by the reviewer show that the great majority of spots contain between 20 and 60 cells (Review Figure 20). As we are suggested to conduct clustering on spots exhibiting low cell count, followed by cell type identification and biological interpretation, we would like to give a detailed explanation why we believe that the suggested analysis would not result in the reduction of bias from differences in cell-count across spots expected by the reviewer.

As previously mentioned, we show that the majority of spots contain between 20 and 60 cells. If we only included spots with low cell counts we would discard a substantial amount of data, for example; excluding spots with more than 5 cells would leave only 1% of the data to be analyzed. Even when considering 10 cells per spot as the threshold, more than 98% of our transcription data would be lost. The clustering strategy employed in this study is based on the assumption that the constructed nearest-neighbour graph represents an adequate approximation of the data manifold. Removing the vast majority of data points would make the constructed graph more sparse and thus would not be able to reflect this assumption anymore. Hence, clustering of spots containing only few cells would not render an accurate depiction of the inherent groups found in the data. In our opinion, given the relatively small size of our dataset, any analysis where we exclude between 95-99% of the data would not produce results that could be compared to the results obtained from the complete data set.

Nevertheless we believe that the remaining analysis performed to address the reviewer's concern of the impact of the cell count in the previous and following responses will be able to relieve the reviewer from her/his worry.

2) Extrapolate on Cluster 5 characterization

previously uncharacterized structure Cluster 5. Based on unsupervised clustering, authors found the cluster 5, which may suggest an uncharacterized liver structure. However, given the above concern about the low resolution, the cluster may not be a novel structure, but result from the mixed existing cell population within spots. Thus, the average expression of various cell types drive the high similarity within cluster 5 and make them different from the others.

We value the reviewer's comment and hope we understand his/her concern accurately, that cluster 5 might not result from the expression of a single cell type. As correctly stated by the reviewer and described in our manuscript, each spot consists of a mixture of contributions from multiple cells. Therefore, we can never assume that one spot consists of only one cell type. We would like to use this opportunity to emphasize again that we do not make the assumption that a spot in the spatial data refers to a single cell or single cell type for this matter. Hence, the reviewer is correct in his/her statement that the average expression of various cell types is driving the transcriptional similarities between spots within cluster 5, distinguishing it from the remaining clusters.

Consequently, when referring to the uncharacterized liver structure we are describing a structure with a distinct composition of multiple cells and cell types rather than single cells and/or cell types. To further illustrate this structural pattern forming the part of the tissue annotated as cluster 5, we refer the reviewer to additional results from comparative analysis of cluster 5 and two different single cell studies (Review Figure 17, Review Table 5), requested by reviewer 2. These additional results will be included in the supplementary material of the original manuscript.

To recapitulate the relevance of these results presented here in brief, we were asked to include additional single cell comparative analyses to further validate our results and conclusions on the identification and interpretation of the uncharacterized structure in cluster 5. We gladly incorporated the suggested additional analysis and consider the results highly informative and relevant for this reviewer's comment.

Incorporating single cell data sets, which are each focusing on a different subset of cell types, we can explore intersecting marker genes expressions of cluster 5 and all annotated cell types. Based on the presence and frequency of intersection cluster 5 and cell type markers, we i) imply the presence of several cell types within the tissue defined by the cluster 5 expression profile and ii) hint on the putative overall contribution of this cell type to the structure (Review Table 5).

To further interpret the importance of the marker genes present in cluster 5 in single cell data, we can visualize the cluster 5 marker gene expression in the respective t-SNE expression (Review Figure 17). This result marks i) the confinement of each marker to the annotated cell types and ii) the expression levels of each marker.

The additional analysis performed to characterize cluster 5 in more detail will be included in Supplementary figure 18 and we address the obtained results in the main manuscript by modifying the manuscript text as shown in the response to *question 2 of reviewer 2*.

Taken together, the additional results presented here contribute - in our opinion - significantly to the understanding of the composition of the tissue within cluster 5. We hope the reviewer agrees with us and considers our answer sufficient to address his/her concerns.

3) *Spot gene expression normalization - refer to 1a,b*

Considering multiple cells in each spot, the Reviewer is curious about the gene expression normalization? Does it normalize to the cell count? This is not a standard step in the traditional single cell RNAseq gene expression normalization, but if not done, the gene expression of spots might be misestimated, i.e., overestimated for spots with higher cell count and underestimated for spots with lower cell count. This may cause problems when conducting differential gene expression analysis.

We appreciate the reviewer's suggestions and understand why normalizing by cell count - or at least accounting for this number - seems appealing when working with spatial transcriptomics (ST). As the reviewer points out, the observations in ST data consist of contributions from several cells, where the number of cells varies between the different locations; a feature that one might suspect to influence the observations. In the following discussion, we seek to explain why we deem it more appropriate to normalize by the total number of unique transcripts rather than cell count.

In short, the main objective for any normalization process - designed for transcriptomics data - is to remove technical noise and biases in the data while preserving true biological signals. One such bias, and perhaps the most obvious, is the sequencing depth (for which total count of unique transcripts is often used as a proxy); if two samples are sequenced at varying depth, comparing their raw read counts could lead to erroneous conclusions, such as upregulation of genes in the sample with more reads. In addition to sequencing depth, one may also attempt to correct for other - unwanted - sources of variation by adding more covariates to the normalization process. These covariates are dependent on the specific study but common examples are: batch id, individual, and disease state. Taken together, we seek to produce normalized data where effects from non-informative sources of variation have been removed, or at least significantly reduced.

Next, we will examine what technical biases that need to be considered in ST data *compared to* single cell data. In single cell data we aim to account for differences in sequencing depth between cells that may arise as a consequence of the experimental setup, where it's hard to obtain consistent library preparation with the minimal starting material found in each cell. Since the ST technique uses a capture based approach, where all probes are sequenced simultaneously, varying sequencing depth between spots is not a confounding factor. However, we expect other sources of bias in our data, such as:

- i) the capture efficacy of our spatial capture locations (spots)
- ii) inhomogeneous transcript density across the tissue

For (i) we might assume that there are some slight differences in the capture efficacy of each spot, which for example could be due to irregularities in the printing process or array damage. However, the differences in capture efficacy could be considered negligible as the method's robustness have been proved in previous publications[1]. Furthermore, these differences - even if they were significant - are something we can't quite account for a priori to analysis, since they aren't systematic and would vary between each array. Hence, we will not discuss (i) from a normalization perspective.

Of more interest is (ii), where the transcript density may vary across the tissue. Three main contributors to this variation are: a) differences in the number of cells contributing with transcripts to a given location, b) differences in tissue permeability during the experimental procedure, and c) cells exhibiting different levels of transcriptional activity. Regardless of the origin of the variation in transcript density, we still want to account for it in our analysis, since it allows us to investigate changes in relative gene expression between regions.

Normalizing with respect to cell counts may reduce some bias introduced by the stated sources a)-c), but it is accompanied by a certain degree of uncertainty and limitations. Firstly, the cell count is not exact, but only an estimate of the number of cells that contributes with transcripts to a spot. It is well-known that different methods of cell segmentation and counting strategies can render fairly different estimates, which also are highly dependent on image quality and resolution. Next, only viable cells that were properly permeabilized will actually contribute to the transcript count of a spot, meaning that the observed cell count does not correspond to the actual number of contributing cells. Since permeability and viability is not homogeneous across the tissue, we are *not* justified to state that the cell count is proportional to the number of contributing cells. Finally, the number of cells cannot by any means capture the permeabilization aspect of the transcript density, hence this would have to be accounted for by other means. Thus, cell count is not only an *inexact* measure of our true covariate (number of contributing cells), but by using it we are also at risk of *introducing further uncertainty* (from the estimation step) into our data.

In contrast, the total UMI count is immediately calculated from the observed data and could act as a proxy for all of the three aforementioned contributors (a-c), as all of them affect the total amount of captured transcripts.

The method we apply for normalization, *sctransform*, is designed to both normalize the data as well as to apply a variance stabilizing step; it is widely used in the single cell community and implemented in the Seurat suite [2]. We refer to the main publication for a detailed outline of the *sctransform* method, but in its most basic form, it uses a GLM-like (Generalized Linear Model) approach to model the count data, according to:

$$\log (E[x_{ij}]) = \beta_{0j} + \beta_{1j} \log (m_i)$$

Where x_{ij} is the expression of gene j in cell i , m_i is the total count of unique transcripts in cell i , and the error function is taken to be the Negative Binomial. The normalized counts are then obtained by calculating the Pearson residuals, which are supposed to represent the part of the expression that the cell's total number of unique transcripts can't account for. In the original publication, the total unique transcript count figures as a proxy for the sequencing depth. However, in the context of ST data it acts as to account for differences in transcript density; which - as mentioned above - is exactly what we aim to do.

As per the reviewer's request, we conducted several additional analyses to investigate the influence and effect of including cell counts in our normalization process (Review Figure 23, Review Table 6), and made the following observations:

- Upon regression of cell count during normalization we can obtain the same number of clusters, namely 6. Upon visual inspection the clustering results after normalizing for cell count per spot look very similar to the original clustering results.
- Upon more detailed inspection of the differences in cluster annotations between data normalized for cell count per spot and original normalization, the number of spots assigned to each cluster is very similar. Spots which were assigned to belong to cluster 0 show the highest number of spots being either assigned to cluster 1 (428 spots) or cluster 2 (106 spots), making cluster 1 the largest cluster in size (Review Table 6).

Thus, we can acknowledge differences for cluster annotations when considering the cell count for normalization but we do not believe they improve the clustering results, as we observe the same number of clusters as without normalization. In addition, most spots observed to switch cluster identity originally belong to cluster 0. Cluster 0 is the biggest cluster in our original analysis and also describes the most unspecific cluster, as we don't observe significantly differentially expressed genes to the same extent as for the remaining clusters (Figure 2c, original manuscript, attached here as Review Figure 10). Since we validated cluster 1 and cluster 2 in depth as central and portal clusters, for instance by DGEA of well established landmark genes, comparison to published and peer-reviewed spatial gene expression data on zonation [3,4,5] and histological annotation, we are confident that our clustering approach reflects the spatial information of the transcriptome accurately.

In summary, we found that the inclusion of cell count in the normalization process, requested by the reviewer, did not substantially improve our results. In addition, we provide a more theoretical argumentation for our approach and why we still consider it superior to cell count-based normalization. To us, these arguments provide a strong justification for our choice of analysis strategy, but are open for further discussion if the reviewer is of a different opinion. We would nevertheless, then ask him/her to further elaborate exactly why cell counts would be preferable to include in the analysis.

Review Figure 23 | Comparison of clustering results considering cell-count as a covariate for clustering analysis. a Depiction of original UMAP embedding as shown in the main manuscript. b UMAP embedding of clustering results upon regression of the observed cell counts per spot during normalization.

Review Table 6 | Intersection of spot cluster annotation with and without cell count normalization. To compare the cluster annotations across spots when cell counts are included as a variable to regress during normalization, we intersected the spots with original cluster annotations (A) and cluster annotations after cell count normalization (B). Matching cluster cells are colored in the original colors used in the UMAP projection (Review Figure 23). Cluster cells depicting the highest differences in non-matching spot count for the cluster annotation are labeled in grey.

		B					
		cluster 2	cluster 0	cluster 5	cluster 3	cluster 4	cluster 1
A	cluster 2	661	44	0	13	0	2
	cluster 0	106	1568	13	63	5	428
	cluster 5	0	7	100	0	0	3
	cluster 3	29	8	1	357	0	69
	cluster 4	13	3	1	1	67	78
	cluster 1	6	34	2	10	0	1171

[1] <https://science.sciencemag.org/content/353/6294/78>

[2] <https://genomebiology.biomedcentral.com/articles/10.1186/s13059-019-1874-1>

[3] <https://pubmed.ncbi.nlm.nih.gov/32579974/>

[4] <https://pubmed.ncbi.nlm.nih.gov/32637622/>

[5] <https://pubmed.ncbi.nlm.nih.gov/28166538/>

4) Elaborate on gene set enrichment for liver metabolic processes along the lobular axis

Gene sets of interest It would be interesting to see if the well-known gene set enrichment can be reproduced along the lobule axis, for example, the gradually decreasing Nutrient- and oxygen-rich condition, Mitochondrial β -oxidation, Gluconeogenesis, Glycogen synthesis, and the gradually increasing WNT signaling, Glycolysis, Lipogenesis from zone1 to zone3. The similar expectation also includes the well-known cell types along the lobular axis.

Response:

We appreciate the reviewers interest in the enrichment of genes with previously described differential expression between the periportal (zone1) and pericentral (zone3) zone. To answer his/ her comment on the enrichment of the requested set of genes (Nutrient- and oxygen-rich condition, Mitochondrial β -oxidation, Gluconeogenesis, Glycogen synthesis, WNT signaling, Glycolysis and Lipogenesis) we performed additional analyses and generated results investigating the zonation of these gene sets in our spatial data.

As a first step we compared the requested pathways against the KEGG database. We considered this database a good fit as it contains a collection of manually drawn pathway maps and genes representing our knowledge of the molecular interaction, reaction and relation networks [1], including gene sets of various metabolic pathways which are active and have been described to be zoned in mouse liver tissue [2].

To this end we used the R package *EnrichmentBrowser* to extract a list of mouse specific KEGG pathways and their respective enriched gene sets. We then selected individual pathways of interest and determined the proportions of gene sets in the central (cluster 2) and portal area (cluster 1) of our spatial data for each pathway.

Upon request of the reviewer and in comparison with previous publications investigating zonation of gene set enrichment of metabolic pathways [3], we included the analysis of the following KEGG pathways:

- WNT signaling
- glycolysis/gluconeogenesis (representative for gluconeogenesis, glycogen synthesis, glycolysis)
- fatty acid metabolism (representative for mitochondrial β -oxidation and lipogenesis)
- glycerolipid metabolism (representative for lipogenesis)
- oxidative phosphorylation and pentose phosphate pathway (representative for oxygen-rich conditions)
- glucagon signaling pathway and fructose and mannose metabolism (representative for energy metabolism)

In agreement with the data from Ben-Moshe and colleagues, our results show that KEGG pathways representative for oxygen-rich conditions (oxidative phosphorylation and pentose phosphate pathway) and energy metabolism (glucagon signaling pathway and fructose and mannose metabolism) show enrichment in the portal area (cluster 1). While central pericentral processes such as high WNT signaling, gluconeogenesis, glycogen synthesis and glycolysis (glycolysis/gluconeogenesis), lipogenesis (fatty acid metabolism, glycerolipid metabolism), mitochondrial β -oxidation (fatty acid metabolism) are enriched in the central areas of the tissue (cluster 2) (Review Figure 24).

Review Figure 24 | Gene set enrichment of selected KEGG pathways for periportal and pericentral regions. Enrichment of established zoned metabolic pathways was determined and proportions of enriched genes sets were compared between the PP zone (cluster 1) and the PC zone (cluster 2) in our data. Negative values (red bars) represent enrichment in the central zone, while positive values (blue bars) represent enrichment in the portal area.

To expand further on the zonation of metabolic pathways between the central and portal venous area we would like to refer the reviewer to additional *bivariate* expression by distance analysis of selected wnt, glucagon, ha-ras, chronic hypoxia and pituitary hormone activated genes, performed in Review Figure 3 - 5. For a detailed description of the results and their interpretation we refer to response 1 of the reviewer 1. In brief, we observed expected

expression by distance trends for selected wnt markers and glucagon activated genes in the central and portal area, respectively. Similar observations were made for ha-ras, chronic hypoxia and pituitary hormone marker genes. The investigation by bivariate plots allowed us to investigate the influence of the distance to portal and central veins simultaneously and shows that distances to both vein types are instrumental to understanding gene expression profiles of these genes and therefore also these metabolic pathways.

Further, we appreciate the reviewer's suggestion to look deeper into the cellular composition along the periportal-pericentral (PP-PC) axis. It should be highlighted again that our data allow us only to infer the relative proportion of cell-type-specific transcripts found within the respective region (spot with known distance from the closest vein) and not the annotation of cell types to individual spots. To address the reviewer's question we would like to refer him/her to the analysis performed for comment 4 of the reviewer 1. In short, we selected marker genes of different non parenchymal cell (NPC) types, some of which have been reported to be typically zoned. The most prominent NPC types exhibiting zonation include liver endothelial cells and Kupffer cells but also hepatic stellate cells and cholangiocytes.

Our results confirm zoned expression of a number of zoned markers for each cell type (Review Figure 13 - 14a, response 4, reviewer 1 and Review Figure 25). The bivariate distance analysis we performed also allowed us to investigate in more detail how the distance to both veins can explain the expression profiles of marker genes along the lobular axis. For instance the hepatic stellate cell (HSC) marker *Hsd22b1* is expressed in close proximity to the portal vein and far from the central vein, while *Lye6e* exhibits high expression when the central and portal vein are in close proximity to each other (within 400 μ m). *Fgfr2* expression is highest in close proximity to the portal vein but is also expressed if a central vein is located in close proximity (Review Figure 14 a, Appendix Table 17).

We can observe equally interesting expression patterns of zoned marker genes of Kupffer cells (Review Figure 14 a, reviewer 1, response 4), endothelial cells (Review Figure 13 c, reviewer 1, response 4), and for the cholangiocyte marker *Spp1* (Review Figure 25 b). As Kupffer cell and endothelial cell marker zonation is described in detail in response 4 of the reviewer 1, we would like to refer reviewer 3 to response 4 of reviewer 1. We would like to point out that *Ctsc* is a shared marker between Kupffer cells and HSCs and shows portal zonation with highest elevation in close proximity to the portal vein while being absent close to the central vein (Review Figure 14a, reviewer 1, response 4).

The cholangiocyte marker *Spp1* [7] exhibits high expression close to the portal vein but is also expressed along the central axis. This indicates that the close distance to the portal vein is necessary for *Spp1* expression irrespective of the distance to the next central vein (Review Figure 25 b, Appendix Table 18). As cholangiocytes are constituting cells of the bile duct, which only form next to the portal vein [8], we expect that the distance to the portal vein for markers of this cell type to be the stronger explanatory variable in comparison to the central vein distance.

In addition, we investigated zoned expression of additional NPC markers previously reported to exhibit zoned expression. These included additional markers for endothelial cells [5] and hepatic stellate cells [4] with the periportal markers *Ltbp4* and *Ngfr* as well as pericentral *Thbd*

and *Adamtsl2*, shown in response 4 of the reviewer 1 (Review Figure 13 a,b, reviewer 1 answer 4). We also attempted to investigate the zonation profile of the endothelial cell markers *Cdh13* for portal and *Efnb2* for central zonation. However, we were unable to detect these genes in our spatial data, most likely due to their relatively scarce abundance in comparison to the remaining transcripts in the tissue. This observation highlights the importance and high relevance of spatial transcriptomics and scRNA Seq data integration, allowing for a more complete understanding of the tissue landscape. Apart from these endothelial cell markers, we also attempted to investigate zonation of smaller NPC populations of the liver, which are expected to be enriched in the portal area [6]. Therefore we sought to explore zonation of the following markers:

- *Cd3d, Cd4* and *Cd8* for T-cells
- *Cd19, Cd79a/b* for B-cells
- *Nkg7, Cd69, Cd7* for NKT-like cells
- *Ly6g* for Granulocytes

Similar to the expression of the endothelial markers *Cdh13* and *Efnb2* we were not able to detect these markers in our expression data, due to their relatively scarce expression when compared to the remaining transcripts in the liver tissue.

a

Review Figure 25 | zonation of Hepatic stellate cell markers present in spatial data and Cholangiocyte marker *Spp1*. **a** Heatmap displaying DE hepatic stellate cell markers between cluster 1 (portal) and cluster 2 (central) (left). Markers with highest expression elevation in cluster 1 or cluster 2 are surrounded by a red box. This gene selection was subjected to expression by distance analysis (middle) and within 400 μm of the vein border. The blue line shows expression by distance from the portal vein border while the red line shows expression from the central vein border. Ribbons around the lines indicate standard deviations of the smoothed curves. Bivariate expression by distance analysis was performed within the same distance (400 μm) to the portal vein (y-axis) and central vein (x-axis) simultaneously (right). Numbers in curly brackets after the gene name indicate that {1} the full model does not perform significantly better than the reduced portal model, {2} the full model does not perform significantly better than the reduced central model, {3} the full model does not perform significantly better than either of the reduced models to explain gene expression along the lobular axis, {4} the full model is outperformed by the baseline intercept value, i.e. not significantly ($p > 0.05$) influenced by either covariate. Relative expression values for each gene are depicted in a color gradient ranging from low (dark) to high (light). **b** Expression by distance (left) and bivariate expression by distance (right) for cholangiocyte marker *Spp1*. Plots can be interpreted as described in sub-figure a.

Collectively, we would like to express our appreciation of the reviewer's suggestions to expand on the zonation of metabolic pathways and cell type markers along the lobular axis. We hope the reviewer can agree with us that we are able to validate the previously observed zonation of liver metabolism and cell type marker genes. In our view, the suggested analysis yields new insights, and provides additional validation of our data.

Therefore, we are including the barplot depicting pathway enrichment (Review Figure 24) as Supplementary figure 10 in the main manuscript and refer to these results as follows:

Line [238 - 241]

“These described genes belong to a small subset of liver metabolic processes. However, we were also able to confirm that a general trend of enrichment of known zoned metabolic pathways^{6,45} can be observed between the PPC and PCC (Supplementary figure 10).”

We further include the heatmaps, illustrating the results of the DGEA for HSCs (Review Figure 25a (left)) in Supplementary Figure 8, as well as the bivariate expression by distance analysis of HSCs and *Spp1* in Supplementary Figure Supplementary Figure 14 and Supplementary table 5. As these results fall in line with requests from reviewer one we would like to refer reviewer 3 to the answer of *comment 4 of reviewer 1* for the modifications made to the manuscript regarding the results for NPC zonation.

[1] <https://www.kegg.jp/kegg/pathway.html>

[2] <https://pubmed.ncbi.nlm.nih.gov/31535084/>

[3] <https://pubmed.ncbi.nlm.nih.gov/30936469/> review

[4] <https://www.ncbi.nlm.nih.gov/pmc/articles/PMC6856722/>

[5] <https://www.ncbi.nlm.nih.gov/pmc/articles/PMC6546596/>

[6] <https://www.ncbi.nlm.nih.gov/pmc/articles/PMC6197289/>

[7] <https://www.ncbi.nlm.nih.gov/pmc/articles/PMC5715535/>

[8] <https://www.ncbi.nlm.nih.gov/pmc/articles/PMC4483763/>

5) Elaborate on statistical method used in Fig 2a)

inappropriate statistic: In Fig 2a), authors calculated the Pearson correlation between cell type proportions that do not follow normal distribution. This is inappropriate, since the Pearson correlation might be largely biased by the outlier values from the data.

We commend the reviewer for looking through our data and rigorously assessing the statistics that we apply in our methods, and are most appreciative of the given feedback. However, we are inclined to argue that our use of the Pearson correlation coefficient (hereafter Pearson's r) to assess co-localization between cell types is justified, and ask him/her to allow us to explain our reasoning below:

In essence, the Pearson's r measures the strength of the relationship between two variables which we assume to have a linear relationship, i.e. $Y = aX + b$. Note that we do not impose any assumptions regarding the underlying distribution of the variables (X and Y). If we have N paired observations of two continuous variables X and Y denoted as x_i and y_i respectively. We may apply the linear transformation (often known as a z-transformation):

$$\hat{x}_i = \frac{x_i - \bar{x}}{\sigma_x}, \quad \hat{y}_i = \frac{y_i - \bar{y}}{\sigma_y}$$

without changing the shape of the distributions. Here the "bared" values represent the mean of each variable. We now assume that a similar linear relationship exists between our transformed values:

$$\hat{Y} = a\hat{X} + b$$

and aim to find the values of a and b that minimize the residual sum of squares for our estimates, that is:

$$\min_{a,b} \sum_i (\hat{y}_i - a\hat{x}_i - b)^2$$

To find these optimal values of a and b we take the partial derivatives of the objective function w.r.t, each parameter.

$$\frac{\partial}{\partial b} \left(\sum_i \hat{y}_i - a\hat{x}_i - b \right)^2 = -2 \left(\sum_i \hat{y}_i - a\hat{x}_i - b \right)$$

$$\frac{\partial}{\partial a} \left(\sum_i \hat{y}_i - a\hat{x}_i - b \right)^2 = -2\hat{x}_i \left(\sum_i \hat{y}_i - a\hat{x}_i - b \right)$$

Setting each partial derivative to zero and manipulating the expression we have:

$$\begin{aligned} \sum_i \hat{x}_i \hat{y}_i &= a \sum_i \hat{x}_i^2 + b \sum_i \hat{x}_i \\ \sum_i \hat{y}_i &= a \sum_i \hat{x}_i + Nb \end{aligned}$$

Since the transformed variables have been mean centered, their mean (and any multiple of it) will be zero, hence:

$$0 = a \cdot 0 + Nb \rightarrow b = 0$$

And

$$\sum_i \hat{x}_i y_i = a \sum_i \hat{x}_i^2 \rightarrow a = \frac{\sum_i \hat{x}_i \hat{y}_i}{\sum_i \hat{x}_i^2}$$

Where, due to division with the standard value in the transformation:

$$\sum_i \hat{x}_i^2 = \sum_i (\hat{x} - 0)^2 = N \text{Var}[\hat{X}] = N$$

Using the full expression for our transformed variables we see that:

$$a = \frac{\sum_i \hat{x}_i \hat{y}_i}{N} = \frac{1}{N} \sum_i \left(\frac{\hat{x}_i - \bar{x}}{\sigma_x} \right) \left(\frac{\hat{y}_i - \bar{y}}{\sigma_y} \right)$$

Which is one of the many ways by which Pearson's r is defined. Hence we see that, without any assumptions of a specific distribution, Pearson's r will measure the strength of a linear relationship between two variables. Which is what we seek to do in our analysis. This derivation is further discussed in the publication *"The needless assumption of normality in Pearson's r"*.^[1]

Still, the reviewer is correct in some sense that normal data is required to compute p-values and construct confidence intervals (CIs) according to the Fisher z' method (analytical).^[2] Since we fully agree with the reviewer that not only effect size but also significance are of relevance and should be taken into consideration into the analysis, we actually used a bootstrap approach to obtain confidence intervals for our correlation coefficients, but failed to describe in the methods section, something we apologize for. We have now added the following paragraph to the Method:

Line [719 - 728]

"Pearson Correlation of cell type proportions

The estimated cell type proportion values do not comply with most of the assumptions to analytically compute confidence intervals for (e.g., normality and heteroskedasticity). Therefore, we used a bootstrap approach to compute confidence intervals, and thus be able to call signals as significant (zero not being included in the CI) or not (zero being included in the CI). For each pair of cell types we generated 10000 bootstrap samples and let the mean of these samples constitute a representative correlation value, while a 95% confidence interval was constructed around this by using the 2.5th and 97.5th percentiles as lower and upper limits. Pairs where the confidence interval overlaps with zero, i.e., being non-significant, are indicated with a gray border."

The code used to generate these plots can be found in the associated github repository and is named corrplot.R.

We hope that the reviewer finds this motivation satisfying and agrees with us that the Pearson's r is an appropriate metric to use to assess potential interactions and patterns of co-localization between cell types.

[1] DOI: 10.1037/h0048216

[2] DOI: 10.3758/s13428-016-0702-8

6) *Elaborate/Justify correlation analysis of stereoscope single cell integration*

overstatement of correlation analysis

Due to mixed cells in spots, authors used the scRNA-seq-derived signatures to estimate the cell type proportions for each spot, followed by the correlation between those proportions. As mentioned in this paper, "Pearson correlation scores between cell type proportions across the spots show positive correlation, to be interpreted as spatial co-localization of non-parenchymal cells".

However, the low values of cell type proportions may be only the noise, that is, those cell types do not exist in the spot. Meanwhile, the correlation herein, may be largely driven by the similarity between the scRNA-seq-derived signatures, the gene sharing or co-expression among signatures. So, correlation may not be because of the real spatial co-localization, but just mathematical similarities. The above possibilities cannot be excluded with no solid validation. As a negative control, the same method is suggested to apply to a single cell RNAseq data. If no similar observation is made, then the co-localization could be partially supported.

The reviewer is fully correct in the statement that we use single cell data to deconvolve the mixed contributions in the spatial data, however we would like to emphasize that we *do not use any signatures* for the cell type deconvolution.

The method we are using, *stereoscope*, models both single cell and spatial data as distributed according to a Negative Binomial (NB) distribution (as is praxis when working with gene expression data [1]). For each gene and every cell type we use maximum likelihood estimation to learn the NB parameters (rate and success probability [2]) that provides the best fit w.r.t. the data; these parameters are learnt from the single cell data, where no mixing occurs, each observation has a single label (cell type) associated with it.

Importantly, the first parameter of the NB distribution is additive between variables with a shared second parameter (when parameterized as described above). We leverage this additive property to model the mixed spatial data's distribution as a linear combination of the rates inferred from the single cell data, where the objective is to find the combination of (positive) coefficients that best explain the observed gene expression. For more details, we refer to the original manuscript [3].

Indeed had we operated with gene signatures, it would have been a very strategic approach to make sure that these signatures did not spuriously correlate with each other in the single cell

data, but as such is not the case we must make a slight modification to this request. Rather than looking at correlations between gene signatures, we compiled an average expression profile for each cell type in the single cell data (mean of normalized expression values), and produced similar correlation plots as for the spatial data using these representative average profiles.

If the patterns of co-localization and segregation that we claim to be present in our tissues were solely driven by gene expression, we would expect the single-cell based correlation matrices to exhibit a similar pattern to those matrices generated from the spatial data. However, as can be seen when comparing Review Figure 26 with Figure 2a in the manuscript (included here as Review Figure 27), the two matrices exhibit very different patterns. These differences imply that our results actually capture true signals of spatial co-localization and segregation, and are driven by expression similarity between cell types.

Review Figure 26 | Correlation between average normalized expression profiles for each cell type in single cell data. Correlations between cell types based on their expression profile are depicted in red for positive and in purple for negative correlations. Grey boxes replace correlations within the same cell type.

Review Figure 27 | Figure 2a (left), original manuscript : Correlation matrix between average cell type proportions across the tissue. Positive correlation values are depicted in red while negative correlation values are shown in purple. Grey boxes replace correlations within the same cell type.

We hope that the reviewer finds the above explanation and additional analysis provides a sufficient amount of support for our use of the Pearson's r to gauge cell type co-localization.

[1] <https://genomebiology.biomedcentral.com/articles/10.1186/s13059-014-0550-8>

[2] https://en.wikipedia.org/wiki/Negative_binomial_distribution

[3] <https://www.nature.com/articles/s42003-020-01247-y>

7) Elaborate on reason for discrepancy between spatial transcriptome and MCA

As mentioned in the paper, “A large portion of spots is assigned to cluster 1 and cluster 2, and 100% of the spots contain hepatocyte markers, showing that - spatially - the liver is predominantly constituted by zoned hepatocytes, while these cells only represent a very small fraction of the MCA data. This discrepancy illustrates the power of complementing single cell transcriptome data with spatial gene expression data to thoroughly delineate liver architecture and the transcriptional landscape of liver tissue, while simultaneously demonstrating the limits of scRNA-seq data integration.”

However, the discrepancies may only result from the limit of spatial transcriptome rather than the limits of scRNA-seq. The spatial transcriptome spot is not at the single cell level, and cluster1/2 may cover both zoned and non-zoned hepatocytes. Thus, it is possible that, even with only a very small proportion of zoned hepatocytes, cluster 1/2 is still good enough to be distinguished from the clusters comprising non-zoned hepatocytes and cells other than hepatocytes. With that said, “A large portion of spots is assigned to cluster 1 and cluster 2” cannot exclude the possibility that the liver has only a small proportion of zoned hepatocytes, as suggested by MCA. More validation is needed to make the conclusion.

The reviewer is absolutely correct in the statement that more validation would be needed to estimate the actual proportion of zoned hepatocytes in the liver tissue and we understand that the phrasing we used to describe the results and conclusion of the single cell data integration can be misinterpreted. Therefore, we would like to thank the reviewer for pointing out this unclarity and elaborate on the conclusions we draw from the single cell integration accordingly.

As the reviewer highlights very importantly, the spatial transcriptomics data does not reach single cell resolution and capture regions (spot) constitute a small mixture of cells as opposed to single cell data, which on the other hand lacks spatial resolution. We would like to emphasize that our study demonstrates the power of combining spatial data and single cell data to delineate transcription profiles across the liver. The data also illustrates the limits of the aforementioned **combination** of these different approaches, and **not** the limits of scRNA-seq. To clarify, we have no intention to attribute the limitations of the integrative analysis to the scRNA-seq data, and have revised our text to make this more clear.

To elaborate on the limitations of the integration approach we believe a clear reiteration of our conclusions to be constructive.

Cluster 1 and cluster 2 of the spatial transcriptomics data each refer to the zoned regions around the portal and central vein, respectively. Each of this region comprises a number of capture regions exhibiting a similar expression profile. The reviewer is correct in the statement, that it is not possible for us to be certain that this expression profile is exclusively generated by zoned hepatocytes. Therefore, we can not assume that these clusters are mainly constituted

of zoned hepatocytes which would be in contrast to the relatively small amount of zoned hepatocytes in the MCA data.

The *stereoscope* integration performed in our study shows the highest proportion values for pericentral and periportal hepatocytes closest to the respective vein. This is visualized in the right panel of Figure 2a in the original manuscript, which we attach here (Review Figure 28 a). Furthermore, a large number of spots with a zoned expression profile (i.e. cluster 1 or cluster 2 annotation) with decreasing proportion values of zoned hepatocytes upon increasing distance to the respective vein. This is exemplified by supplementary figure 9 of the original manuscript, attached here (Review Figure 28 b).

We interpret these data to demonstrate two different possibilities: First the spots belonging to cluster 1 and cluster 2 include a small number of zoned hepatocytes, dominating the expression profile of the respective capture region, due to e.g. high numbers of mRNA contents of transcripts responsible for the observed zonation.

Secondly, it might indicate that many cells within the spots exhibit an expression profile of zoned hepatocytes annotated in the MCA single cell dataset.

Review Figure 28 | **a** Figure 2a (right), original manuscript: Quantile scales of cell-proportions annotated as pericentral and periportal hepatocytes (see methods) are mapped on spatial transcriptomics spot data (top). UMAP embedding of single-cell data of the Mouse Cell Atlas (MCA)³⁹ grouped by annotated cell types (bottom right). Numeration behind the cell types represent annotation of MCA data (B cell-1 : Fcgr high, -2 : Jchain high, Dendritic cell-1 : Cst3 high, -2 : Siglec high, Epithelial cell-1 : Spp1 high, -2 : /, Erythroblast-1 : Hbb-bs high, -2 : Hbb-bt high, Hepatocyte-1 : Fabp1 high, -2 : mt-Nd4 high, T cell-1 : Gzma high, -2 : Trbc2 high). Encircled clusters in the plot refer to pericentral or periportal hepatocytes of MCA data. Quantile scales of cell-proportions annotated as pericentral and periportal hepatocytes (Methods) are mapped on Spatial Transcriptomics spot data (top right). **b** Supplementary Figure 11, original manuscript: Expression by distance of annotated cell types of MCA single cell data from the outer

portal and central vein borders. Distances of pericentral hepatocytes from central to portal veins are depicted in the figure on the left. Distances of periportal hepatocytes from portal to central veins are depicted in the figure on the right.

Given the relatively small number of annotated pericentral and periportal hepatocytes in the MCA for liver tissue we speculate the first possibility of zoned hepatocytes giving rise to a high proportion value in the respective spots. Our conclusion highlights the limits of single cell data **integration** on current spatial transcriptomics data of liver tissue, which we are not able to resolve within the scope of this study. This emphasizes the importance of the consideration and integration of transcription data generated by diverse methods to vasten our current knowledge on tissue biology.

To address the reviewers concerns and clarify our conclusions of the scRNA-seq integration we adjusted the text in the manuscript to the following:

Line [177 - 182]

“A large portion of spots were assigned to cluster 1 and cluster 2, while these cells only represent a very small fraction of the MCA data. This discrepancy implies that a relatively small cell type population identified by scRNA-seq can constitute a large proportion of the spatially profiled cells, illustrating the power of complementing single cell transcriptome data with spatial gene expression data to thoroughly delineate liver architecture and the transcriptional landscape of liver tissue.”

and

Line [378 - 382]

“The observed discrepancies between ours and the MCA data may result from the different technical limitations that scRNA-seq and spatial data generation face, emphasizing the current limits of scRNA-seq data integration. For instance, transcriptionally highly active or physically large cells might mask cell types with moderate to low transcriptional levels in ST data.”

We hope the detailed explanation above and the performed changes to the manuscript text clarify the interpretation of our performed analysis and provide enough additional information for the reviewer to be able to agree with our conclusions.

8) Prediction accuracy of portal and central veins

The authors predicted the portal and central veins based on gene expression. Although conducted cross validation, the Reviewer cannot find the relevant performance evaluation, failing to see how good the prediction is.

First, regarding the prediction evaluation, the authors are suggested to provide the ROC curve; Second, regarding the prediction result, it is better to overlay the statistics of prediction, eg., the log ratio on Fig 3c, so that others can see how confident the vein prediction is.

Based on the reviewer's request we here provide additional performance metrics for the binary vein type classifier presented in our study. More precisely we show the ROC (receiver operating characteristic) curve and report the AUC score.

In line with our previous analyses, we use a "leave one out cross validation" (LOOCV) scheme to evaluate the model's performance. We describe the cross validation more thoroughly in the Methods section of the main text, but in brief: in each iteration we train the classifier on the annotated veins from all sections except one, to then predict the vein type of the veins within the left-out section.

To summarize the results from each fold, we assembled an *average* ROC curve by - for each fold - interpolating the TPR (true positive rate) over a set of FPR (false positive rate) values in the closed interval [0,1], results are shown in Review Figure 29. The average AUC score, here approximately 0.85, is the arithmetic mean of all the individual AUC scores, see Review Table 7.

Both the AUC and, unexpectedly, the accuracy was fairly high (>0.8) for a majority of the samples except for the sample CN65-E1. We have not fully established why the accuracy drops so severely for CN65-E1, since the other samples from the same individual CN65-D1 and CN65-D2 have much higher accuracies of 1.0 respectively 0.8. Possible reasons include poor sample quality, which can also complicate manual annotations considerably.

We hope the reviewer finds this extended evaluation of the classifier satisfactory. As we agree with the reviewer that extensive performance evaluation is important to include, we added the requested ROC curve and AUC scores as well as the previously conducted cross validation in the supplementary material as Supplementary Figure 15. We refer to the results of the classifier evaluation as follows:

Line [299 - 302]

"The model constructed in this study (Methods) corresponds convincingly to manually annotated central and portal veins based on the expression profile of their respective neighborhood across

all sections from different biological origins (caudate and right liver lobe) (Supplementary Figure 15).”

Review Figure 29 | ROC curve illustrating the performance of the expression-based vein type classifier. The blue line represents the average AUC taken over all the folds in the cross validation analysis. The red dashed line corresponds to the curve obtained from a completely random classifier. The gray shaded area represents the interval of the mean plus/minus one standard error. Av, AUC stands for average AUC and is the arithmetic mean taken across all folds.

Review Table 7 | Performance validation of binary vein classifier | For each sample prediction performance was evaluated by training according to an LOOCV. The performance is illustrated by the results of the accuracy of the cross validation analysis and AUC. A value of 1 denotes highest accuracy, while a value of 0 denotes lowest accuracy.

predict on	train on	accuracy	AUC
CN73-D1	CN65-E1, CN73-E2, CN65-D1, CN73-C1, CN16-D2, CN65-D2, CN16-E2	0.9333	0.9444

CN16-E2	CN65-E1, CN73-E2, CN65-D1, CN73-C1, CN16-D2, CN65-D2, CN73-D1	1	1
CN65-D2	CN65-E1, CN73-E2, CN65-D1, CN73-C1, CN16-D2, CN16-E2, CN73-D1	0.7143	0.8
CN16-D2	CN65-E1, CN73-E2, CN65-D1, CN73-C1, CN65-D2, CN16-E2, CN73-D1	0.9473	0.9886
CN73-C1	CN65-E1, CN73-E2, CN65-D1, CN16-D2, CN65-D2, CN16-E2, CN73-D1	0.9375	0.9524
CN65-D1	CN65-E1, CN73-E2, CN73-C1, CN16-D2, CN65-D2, CN16-E2, CN73-D1	1	1
CN73-E2	CN65-E1, CN65-D1, CN73-C1, CN16-D2, CN65-D2, CN16-E2, CN73-D1	0.8077	0.8333
CN65-E1	CN73-E2, CN65-D1, CN73-C1, CN16-D2, CN65-D2, CN16-E2, CN73-D1	0.2222	0.2857

9) Immunostaining of liver tissue

Considering the above concerns, in addition to H&E staining, some other Immunostaining with antibodies against liver zonation and cell types are also suggested, for example, CD73 for pericentral zonation and E-cadherin for periportal zonation. These might be overlaid on all clustering, cell types and vein prediction, which may serve as the orthogonal validation for multiple observations.

We agree with the reviewer that an additional cross-validation of the vein prediction and clustering on the tissue sections we used for the Spatial Transcriptomics experiment would be beneficial to additionally confirm the observation made in our study. To address this comment properly we consider it important to briefly reiterate the most crucial steps of the peer-reviewed ST protocol, carried out in this study. This protocol is described more extensively in the main manuscript and in [1].

The preparation of spatially resolved sequencing libraries begins with fixation for RNA preservation and Hematoxylin and Eosin (H&E) staining for visualization. Subsequently, the tissue is permeabilized allowing the mRNA transcripts to hybridize to the spatially and individually barcoded probes on the array surface under the tissue. Then, after the cDNA synthesis, the tissue is **enzymatically digested from the array**.

Thus, immunostaining of the same tissue sections used in the experiments presented here is not feasible to optimize for this platform.

The reviewer mentions the additional validation via immunostaining on two distinctive levels; first the liver zonation, i.e. immunostaining for protein markers (CD73 and E-cadherin) at the central and portal vein. Secondly, he/she suggests immunostaining for specific cell types.

We would like to emphasize that we have considered performing additional orthogonal validations using immunostaining for zonation markers and/or cell types during generation of the data presented in our study. Unfortunately, staining protocols for the ST platform presented here are not readily available and optimized. However, we would like to address these suggestions and elaborate in detail why we believe the current validations presented are sufficient for the scope of our study.

1) Immunostaining against pericentral and periportal zonation markers

The reviewer requests the potential overlay of immunostaining against pericentral and periportal protein markers to serve as an orthogonal validation for clustering and vein predictions. Generally, the ability of the methods presented here to reliably capture the transcripts in the capture regions has been validated extensively in multiple peer-reviewed papers [eg. 2-9]. Based on the aforementioned limitation of the ST method used in this study we performed the following validations of our observations made for pericentral and periportal zonation:

- As a first validation we performed histological annotations of central and portal veins on the H&E stained images. These annotations were performed by a trained histology expert and the process is described in detail in the materials and methods section of the main manuscript.
- Secondly we overlaid the clustering results from our analysis to validate the histological and clustering observations as observed in Figure 2b of the main manuscript (attached here as Review Figure 30 a). These cluster annotations aligned well with the performed histological annotations.
- Further, we explored marker gene expression profiles of cluster 1 and cluster 2 shown in the heatmap in Figure 1c of the main manuscript (attached here as Review Figure 10) . Pearson correlation analysis of cluster 1 and cluster 2 markers, revealed strong anticorrelation of markers of cluster 1 and cluster 2, interpreted as spatial segregation between marker gene expression of annotated central and portal areas, demonstrated in Figure 2c of the original manuscript, attached here as Review Figure 30 b.
- Importantly the reported markers for the central and portal area have been validated extensively by previous spatial studies on transcription along the centrilobular axis. For instance, studies performed by Halpern et al. [10] have used smFISH experiments for orthogonal validation of expression of portal and central marker genes in various distances to the central and/or portal veins. Thus we are able to use this peer-reviewed external data to cross-validate gene expression along the lobular axis of our data. To illustrate this further we would like to refer to the Supplementary figure 8 a,b of our manuscript attached here as Review Figure 31 and Review Figure 32. Here, we visualized the periportal (cluster 1) and pericentral (cluster 2) marker genes identified by unsupervised clustering and DGEA of our spatial data on the reconstructed spatial layers of the single cell data. Our data is highly similar to the data by Halpern et al. showing that the portal markers of our data display the lowest expression in layer 1 (the most central layer in the single cell study) with increasing expression towards the most

portal layer (layer 9). The opposite expression gradient applies to central ST markers along the reconstructed single cell layers.

Therefore the comparison of our data to the single-cell reconstruction work by Halpern et al. represents an indirect orthogonal validation of our spatial data by smFISH and the performed histological annotations provide direct validation on the tissue analysed here. This - in our opinion - provides sufficient validation of pericentral and periportal clustering and computational vein prediction based on gene expression.

Review Figure 30 | **a** Figure 2b: Visualization of spots representing gene expression profiles of cluster 1 (portal vein, blue) and cluster 2 (central vein, red) on H&E stained tissue (right), compared with visual histology annotations of central- (red circles) and portal- (blue circles) veins (left) (scale bar indicates 500 μ m)., **b** Figure 2c: Pearson correlations of genes expressed in cluster 1 and 2 ordered by their first principal component (see methods). Genes with high expression in the pericentral cluster (cluster 2) show negative correlation with genes highly expressed in the periportal cluster (cluster 1) and vice versa. Genes present within cluster 1 or cluster 2 exhibit positive correlation with genes in the same cluster.

Review Figure 31 | Supplementary Figure 8a : Visualization of expression of periportal marker genes identified by unsupervised clustering and DGEA of ST data across reconstructed spatial layers (1-9) of single cell data zonation matrix [11].

Review Figure 32 | Supplementary Figure 8b : Visualization of expression of pericentral marker genes identified by unsupervised clustering and DGEA of ST data across reconstructed spatial layers (1-9) of single cell data zonation matrix [11].

2) Immunostaining against cell types

In the context of cell type validation, we would like to emphasize that we do not assign cell type labels to the spots, which constitute transcripts of mixed cell populations. The single cell integration performed in this study rather provides information about cell type proportions across the tissue than annotating a cell type to each spot. For instance, if the single cell integration predicts a high proportion of Kupffer cells in a single spot, this **does not imply** that this spot either consists exclusively of Kupffer cells nor that this spot contains unusually high numbers of Kupffer cells compared to other cell types. It rather implies that based on the provided single cell data set the spot in question exhibits a proportional similarity to single cell dataset's cell type annotations between 0 and 1 when compared to the remaining tissue. Thus, we agree with the

reviewer that immunostaining against cell types would confirm the general presence of cell types across the tissue. However, the binary results on cell type presence received from immunostaining are not expected to correlate directly with the observed cell type proportions based on gene expression patterns across the tissue [11,12].

Given the comment raised above, the incentive of this study does not include the detailed annotation of cell type distributions within spots and therefore across the tissue. Therefore, we believe the requested cell type validation is outside the scope of this study. Nonetheless, conclusions on the distributions of cell type proportions presented in this study are still highly informative for future research on e.g. further detailed scRNA Seq studies of the liver.

Taken together, we hope the reviewer is satisfied with the detailed explanation of the undertaken steps in this study to validate the observed portal and central zonation in our data and can agree with us that immunostaining against cell types will be more suitable for potential future spatial studies with resolution on the single cell level and targeted single cell studies. To highlight the importance of the external validation (e.g. by using spatially reconstructed scRNA-seq data by Halpern et al. [10]) we revised the original manuscript as follows:

Line [256 - 258]

“These results are in agreement with the observed expression gradients in spatially reconstructed layers in Halpern et al. 11, which are orthogonally validated by smFISH (Supplementary figure 12).”

[1] <https://pubmed.ncbi.nlm.nih.gov/27365449/>

[2] <https://science.sciencemag.org/content/353/6294/78>

[3] <https://www.nature.com/articles/s41598-018-27627-3>

[4] <https://www.nature.com/articles/s41467-018-04724-5>

[5] <https://science.sciencemag.org/content/364/6435/89>

[6] [https://www.cell.com/cell/fulltext/S0092-8674\(19\)31282-6?rss=yes](https://www.cell.com/cell/fulltext/S0092-8674(19)31282-6?rss=yes)

[7] <https://www.sciencedirect.com/science/article/pii/S0092867420306723?via%3Dihub>

[8] <https://www.sciencedirect.com/science/article/pii/S0092867420308151?via%3Dihub>

[9] <https://www.sciencedirect.com/science/article/pii/S2589004220307483>

[10] <https://www.nature.com/articles/nature21065>

[11] <https://pubmed.ncbi.nlm.nih.gov/32709985/>

[12] <https://pubmed.ncbi.nlm.nih.gov/21179022/>

Appendix

Appendix Table 1 | LRT results for central and portal area markers. Central_sig and portal_sig columns indicate whether including the central respectively portal vein distance covariate significantly improves a reduced model that only uses the intercept and the other distance covariate.

Gene	p-value only central	p-value only portal	p-value only intercept	Central sig	Portal sig	Intercept
Sds	1.39E-14	8.77E-05	6.31E-21	TRUE	TRUE	TRUE
Cyp2f2	1.49E-13	1.03E-03	2.69E-18	TRUE	TRUE	TRUE
Hal	7.91E-09	3.21E-05	6.63E-15	TRUE	TRUE	TRUE
Hsd17b13	8.10E-08	2.42E-07	2.37E-16	TRUE	TRUE	TRUE
Aldh1b1	1.14E-10	4.39E-02	1.07E-12	TRUE	TRUE	TRUE
Glul	2.58E-05	3.00E-20	5.80E-28	TRUE	TRUE	TRUE
Oat	2.32E-10	1.38E-13	8.27E-27	TRUE	TRUE	TRUE
Slc1a2	1.91E-05	1.34E-12	1.48E-19	TRUE	TRUE	TRUE
Cyp2e1	3.90E-16	1.68E-10	6.83E-30	TRUE	TRUE	TRUE
Cyp2a5	1.88E-07	1.72E-14	2.77E-24	TRUE	TRUE	TRUE

Appendix Table 2 | LRT results for selected glucagon targets. Central_sig and portal_sig columns indicate whether including the central respectively portal vein distance covariate significantly improves a reduced model that only uses the intercept and the other distance covariate.

Gene	p-value only central	p-value only portal	p-value only intercept	Central sig	Portal sig	Intercept
Mup20	0.6703	0.3110	0.5898	FALSE	FALSE	FALSE

Sds	1.39E-14	8.77E-05	6.31E-21	TRUE	TRUE	TRUE
Hal	7.91E-09	3.21E-05	6.63E-15	TRUE	TRUE	TRUE
Ctsc	3.12E-10	7.62E-06	2.42E-17	TRUE	TRUE	TRUE
Aldh1b1	1.14E-10	4.39E-02	1.07E-12	TRUE	FALSE	TRUE
Hsd17b6	3.06E-07	7.36E-02	1.44E-08	TRUE	FALSE	TRUE
Etnpl	7.36E-05	7.51E-06	6.39E-11	TRUE	TRUE	TRUE
Slc7a2	7.73E-06	6.05E-03	1.81E-08	TRUE	TRUE	TRUE
Apoa4	6.50E-09	5.46E-06	5.55E-16	TRUE	TRUE	TRUE
Gls2	2.24E-05	1.10E-02	1.34E-07	TRUE	FALSE	TRUE
Cyp17a1	6.67E-03	1.91E-04	8.47E-07	TRUE	TRUE	TRUE
Mmd2	6.88E-04	1.18E-01	0.0001	TRUE	FALSE	TRUE

Appendix Table 3 | LRT results for selected wnt targets. Central_sig and portal_sig columns indicate whether including the central respectively portal vein distance covariate significantly improves a reduced model that only uses the intercept and the other distance covariate.

Gene	p-value only central	p-value only portal	p-value only intercept	Central sig	Portal sig	Intercept
Axin2	0.7522	0.0365	0.0762	FALSE	TRUE	FALSE
Lgr5	0.9487	0.1700	0.3731	FALSE	FALSE	FALSE
Slc1a2	1.91E-05	1.34E-12	1.48E-19	TRUE	FALSE	TRUE
Cyp2a5	1.88E-07	1.72E-14	2.77E-24	TRUE	TRUE	TRUE

Mup17	5.73E-06	0.3668	3.68E-06	TRUE	FALSE	TRUE
Cyp2e1	3.90E-16	1.68E-10	6.83E-30	TRUE	TRUE	TRUE
Gulo	1.03E-11	1.93E-09	1.58E-23	TRUE	TRUE	TRUE
Slc22a1	6.73E-06	1.15E-11	4.51E-19	TRUE	TRUE	TRUE
Lect2	6.88E-11	2.59E-11	8.70E-25	TRUE	TRUE	TRUE
Cyp2c37	2.76E-15	1.90E-06	6.54E-24	TRUE	TRUE	TRUE
Aldh1a1	7.53E-05	0.0001	2.10E-09	TRUE	TRUE	TRUE
Cyp1a2	3.54E-09	1.98E-06	7.61E-17	TRUE	TRUE	TRUE

Appendix Table 4 | LRT results for selected ha-ras targets. Central_sig and portal_sig columns indicate whether including the central respectively portal vein distance covariate significantly improves a reduced model that only uses the intercept and the other distance covariate.

Gene	p-value only central	p-value only portal	p-value only intercept	Central sig	Portal sig	Intercept
Cyp2f2	1.49E-13	0.0010	2.69E-18	TRUE	TRUE	TRUE
Apoa4	6.50E-09	5.46E-06	5.55E-16	TRUE	TRUE	TRUE
Mup17	5.73E-06	0.3668	3.68E-06	TRUE	FALSE	TRUE
Oat	2.32E-10	1.38E-13	8.27E-27	TRUE	TRUE	TRUE

Appendix Table 5 | LRT results for selected chronic hypoxia targets. Central_sig and portal_sig columns indicate whether including the central respectively portal vein distance covariate significantly improves a reduced model that only uses the intercept and the other distance covariate.

Gene	p-value only central	p-value only portal	p-value only intercept	Central sig	Portal sig	Intercept
Hal	7.91E-09	3.21E-05	6.63E-15	TRUE	TRUE	TRUE
Pck1	2.55E-05	0.1737	6.34E-06	TRUE	FALSE	TRUE
Gstm3	0.0069	4.55E-08	6.08E-11	TRUE	TRUE	TRUE
Slc1a2	6.73E-06	1.34E-12	1.48E-19	TRUE	TRUE	TRUE

Appendix Table 6 | LRT results for selected pituitary hormone targets. Central_sig and portal_sig columns indicate whether including the central respectively portal vein distance covariate significantly improves a reduced model that only uses the intercept and the other distance covariate.

Gene	p-value only central	p-value only portal	p-value only intercept	Central sig	Portal sig	Intercept
Fmo3	0.0003	0.0526	2.16E-05	TRUE	FALSE	TRUE
Igfbp2	0.1386	0.0542	0.0197	FALSE	FALSE	TRUE
Cyp4a10	0.0306	0.0002	6.42E-06	TRUE	TRUE	TRUE
Slc22a1	6.73E-06	1.15E-11	4.51E-19	TRUE	TRUE	TRUE

Appendix Table 7 | Results of differential gene expression for cluster 5

	p_val	avg_logFC	pct.1	pct.2	p_val_adj
Gsn	1.56E-109	1.69	0.96	0.21	1.49E-105
Dpt	2.73E-107	0.94	0.67	0.08	2.59E-103
Mgp	3.66E-78	0.58	0.39	0.03	3.48E-74
Col1a1	4.90E-76	0.93	0.67	0.12	4.66E-72
Tagln	2.90E-75	0.68	0.51	0.06	2.76E-71
Col3a1	4.05E-75	1.61	0.94	0.36	3.85E-71
Vim	4.02E-54	1.02	0.81	0.27	3.82E-50

Col1a2	7.19E-53	0.99	0.71	0.20	6.83E-49
H2-Eb1	3.53E-52	0.76	0.59	0.12	3.35E-48
Crip1	2.35E-50	0.64	0.56	0.11	2.24E-46
Acta2	9.76E-49	0.54	0.32	0.04	9.28E-45
Ahnak	7.97E-48	0.75	0.66	0.17	7.57E-44
Tmsb4x	1.28E-42	0.96	0.99	0.86	1.21E-38
Timp2	2.28E-41	0.57	0.52	0.12	2.17E-37
Dcn	1.43E-40	0.75	1.00	0.94	1.36E-36
H2-Aa	5.48E-36	0.61	0.59	0.17	5.21E-32
Lum	1.29E-35	0.64	0.64	0.20	1.23E-31
H2-Ab1	1.13E-30	0.57	0.55	0.16	1.07E-26
Cd74	5.89E-30	0.65	0.66	0.25	5.59E-26
Igfbp7	1.25E-28	0.72	0.95	0.71	1.19E-24
Sparc	1.89E-28	0.72	0.84	0.49	1.80E-24
Bgn	6.64E-25	0.73	0.78	0.45	6.32E-21
Col14a1	3.47E-23	0.51	0.64	0.27	3.29E-19
Spp1	2.48E-21	1.08	0.56	0.23	2.36E-17
Timp3	5.35E-19	0.54	0.50	0.20	5.08E-15
Txnip	4.99E-18	0.55	0.78	0.51	4.74E-14

Appendix Table 8 | LRT results for selected non-parenchymal cell markers. Central_sig and portal_sig columns indicate whether including the central respectively portal vein distance covariate significantly improves a reduced model that only uses the intercept and the other distance covariate.

Gene	p-value only central	p-value only portal	p-value only intercept	Central sig	Portal sig	Intercept
Thbd	0.2214	0.0779	0.0475	FALSE	FALSE	TRUE
Ltbp4	0.0076	0.0041	0.0023	TRUE	TRUE	TRUE
Adamtsl2	0.3904	0.0286	0.0869	FALSE	TRUE	FALSE
Sox4	0.0204	0.9431	0.0587	TRUE	FALSE	FALSE
Ngfr	0.6859	0.2514	0.3991	FALSE	FALSE	FALSE
Tagln	0.0701	0.2849	0.1596	FALSE	FALSE	FALSE

Appendix Table 9 | Results of differential gene expression of endothelial marker genes

(PMID: 30222169) between cluster 1 and cluster 2.

	p_val	avg_log 2FC	pct.1	pct.2	p_val_adj
Sepp1	1.69E-81	0.28	1.00	1.00	2.82E-79
Aass	2.15E-15	0.27	0.94	0.86	3.59E-13
Ctsl	3.17E-20	0.22	1.00	1.00	5.29E-18
Tcn2	1.26E-04	0.15	0.66	0.62	2.11E-02
Ntn4	6.10E-13	0.15	0.17	0.05	1.02E-10
Man2a1	8.46E-05	0.15	0.78	0.75	1.41E-02
Adam23	9.70E-07	0.11	0.18	0.10	1.62E-04
Ramp2	2.09E-04	-0.08	0.12	0.18	3.50E-02
Lamp2	3.16E-05	-0.09	1.00	1.00	5.28E-03
Egfl7	4.08E-05	-0.10	0.13	0.20	6.81E-03
Slc43a3	9.14E-05	-0.11	0.14	0.21	1.53E-02
Stab1	2.63E-04	-0.11	0.26	0.34	4.38E-02
Mylip	6.43E-07	-0.11	0.11	0.19	1.07E-04
Calcr1	2.45E-05	-0.12	0.18	0.26	4.09E-03
F2r	6.35E-07	-0.12	0.16	0.25	1.06E-04
Crip2	2.57E-04	-0.13	0.41	0.49	4.29E-02
Dhrs3	4.49E-05	-0.14	0.90	0.92	7.49E-03
Slc29a1	1.09E-05	-0.14	0.77	0.83	1.83E-03
Xdh	1.73E-05	-0.15	0.38	0.47	2.88E-03
Cd36	1.34E-06	-0.16	0.36	0.46	2.24E-04
Fermt2	1.62E-05	-0.16	0.45	0.53	2.71E-03
Ptprb	2.97E-04	-0.16	0.53	0.58	4.96E-02
Gas6	7.24E-07	-0.16	0.24	0.34	1.21E-04
Eng	4.95E-08	-0.19	0.36	0.48	8.26E-06
Kit	9.99E-23	-0.24	0.09	0.25	1.67E-20
Tsc22d1	2.12E-34	-0.44	0.27	0.52	3.54E-32
Lifr	2.46E-55	-0.5146	0.89	0.974	4.12E-53
Ndrp1	3.05E-55	-0.5298	0.215	0.544	5.09E-53

Appendix Table 10 | LRT results for selected zonated endothelial cell markers. Central_sig and portal_sig columns indicate whether including the central respectively portal vein distance covariate significantly improves a reduced model that only uses the intercept and the other distance covariate.

Gene	p-value only central	p-value only portal	p-value only intercept	Central sig	Portal sig	Inter- cept
Sepp1	0.0004	6.02E-07	2.64E-11	TRUE	TRUE	TRUE

Aass	0.0005	0.9113	0.0013	TRUE	FALSE	TRUE
Ctsl	0.0774	0.1662	0.0349	FALSE	FALSE	TRUE
Tsc22d1	0.6480	0.0150	0.0510	FALSE	TRUE	FALSE
Lifr	0.0056	0.0248	0.0002	TRUE	TRUE	TRUE
Ndrg1	0.0179	0.0266	0.0009	TRUE	TRUE	TRUE

Appendix Table 11 | Results of differential gene expression of plasmacytoid dendritic cell (pDC) marker genes (PMID: 30222169) between cluster 1 and cluster 2.

	p_val	avg_log2FC	pct.1	pct.2	p_val_adj
Atp1b1	1.78E-14	0.01	0.61	0.47	8.00E-13
Lgals1	1.14E-08	0.00	0.08	0.17	5.13E-07
Upb1	2.52E-07	-0.01	0.83	0.87	1.13E-05
Rnf187	3.51E-06	-0.01	0.62	0.71	1.58E-04
Mpeg1	2.00E-04	0.00	0.41	0.32	9.00E-03
Dirc2	4.59E-04	0.00	0.22	0.28	2.07E-02
Ctsh	1.06E-03	-0.01	0.88	0.91	4.75E-02

Appendix Table 12 | Results of differential gene expression of neutrophil marker genes (PMID: 30222169) between cluster 1 and cluster 2.

	p_val	avg_log2FC	pct.1	pct.2	p_val_adj
Dgat2	5.75E-12	-0.04	0.96	0.98	1.73E-10
Grina	1.90E-08	-0.01	0.62	0.72	5.69E-07
Gsr	3.59E-05	-0.01	0.60	0.67	1.08E-03

Appendix Table 13 | Results of differential gene expression of Kupffer cell marker genes (PMID: 30222169) between cluster 1 and cluster 2.

	p_val	avg_log2FC	pct.1	pct.2	p_val_adj
Ctsc	5.31E-129	0.03	0.93	0.63	5.74E-127
Blvrb	5.35E-60	-0.01	0.50	0.78	5.77E-58
Plbd1	5.22E-41	0.00	0.25	0.54	5.63E-39
Hpgd	1.66E-33	-0.01	0.89	0.96	1.79E-31
Creg1	2.46E-14	-0.02	0.98	1.00	2.66E-12
Ctsb	4.63E-11	0.02	0.99	0.99	5.00E-09

Igf1	3.86E-09	0.03	1.00	1.00	4.17E-07
C6	5.16E-09	0.00	0.05	0.12	5.57E-07
Cd81	1.63E-08	-0.01	0.96	0.98	1.76E-06
Lpl	4.09E-07	0.00	0.21	0.31	4.42E-05
Slc40a1	5.07E-07	0.00	0.64	0.56	5.48E-05
Cd302	1.43E-05	-0.01	0.99	1.00	1.54E-03
Cd5l	1.83E-05	0.00	0.43	0.33	1.98E-03
Lipa	3.01E-05	0.00	0.91	0.92	3.26E-03
Apoe	3.13E-05	-0.10	1.00	1.00	3.38E-03
St3gal5	1.07E-04	0.00	0.27	0.20	1.15E-02
Smpdl3a	1.63E-04	0.00	0.46	0.38	1.76E-02
Mpeg1	2.00E-04	0.00	0.41	0.32	2.16E-02
Lgmn	2.07E-04	0.00	0.53	0.45	2.24E-02
Axl	4.52E-04	0.00	0.18	0.12	4.88E-02

Appendix Table 14 | LRT results for selected zonated Kupffer cell markers. Central_sig and portal_sig columns indicate whether including the central respectively portal vein distance covariate significantly improves a reduced model that only uses the intercept and the other distance covariate.

Gene	p-value only central	p-value only portal	p-value only intercept	Central sig	Portal sig	Intercept
Ctsc	3.12E-10	7.62E-06	2.42E-17	TRUE	TRUE	TRUE
Igf1	0.0011	0.0010	6.37E-07	TRUE	TRUE	TRUE
Ctsb	0.0063	0.8824	0.02	TRUE	FALSE	TRUE
Hpgd	0.3060	0.1117	0.09	FALSE	FALSE	FALSE
Creg1	0.0722	0.0526	0.01	FALSE	FALSE	TRUE
Apoe	0.9284	0.0017	0.01	FALSE	TRUE	TRUE

Appendix Table 15 | Results of differential gene expression of immune marker genes (GO:0002376 , immune system processes) between cluster 1 and cluster 2.

	p_val	avg_log2FC	pct.1	pct.2	p_val_adj
Arg1	4.79E-161	0.72	1.00	1.00	1.26E-158

C9	1.36E-87	0.70	0.97	0.83	3.58E-85
Hc	1.71E-35	0.34	1.00	0.99	4.50E-33
Cfi	4.48E-40	0.34	1.00	1.00	1.18E-37
H2-Q10	7.53E-60	0.32	1.00	1.00	1.98E-57
Fgb	1.91E-63	0.27	1.00	1.00	5.04E-61
C3	3.57E-47	0.25	1.00	1.00	9.40E-45
Cfh	5.20E-23	0.25	1.00	1.00	1.36E-20
Lbp	7.97E-09	0.24	0.67	0.58	2.09E-06
Fga	2.79E-40	0.20	1.00	1.00	7.34E-38
Hp	1.21E-12	0.19	1.00	1.00	3.20E-10
H2-K1	2.28E-16	0.19	1.00	1.00	6.00E-14
Fgg	5.14E-30	0.17	1.00	1.00	1.35E-27
Alcam	1.63E-06	0.17	0.46	0.36	4.30E+01
Pglyrp2	1.11E-05	0.16	0.53	0.43	2.90E+01
C2	4.03E-05	0.15	0.64	0.56	0.01
Zap70	1.88E-06	0.13	0.21	0.13	4.90E+01
H2-Q7	1.34E-05	0.13	0.28	0.20	3.50E+01
Cd5l	5.37E-05	0.13	0.39	0.30	1.40E+01
C8g	3.95E-06	0.11	1.00	1.00	1.00E+00
B2m	1.78E-08	0.08	1.00	1.00	4.68E-06
C6	5.16E-09	-0.10	0.05	0.12	1.35E-06
Spon2	3.42E-06	-0.11	0.11	0.18	8.90E+01
Irgm1	3.47E-05	-0.13	0.33	0.42	9.10E+01
Cd81	4.67E-14	-0.19	0.99	1.00	1.23E-11
Msrb1	1.88E-23	-0.28	0.97	0.99	4.96E-21
Psm1	3.69E-14	-0.30	0.67	0.77	9.72E-12
Mbl1	1.17E-20	-0.34	0.73	0.85	3.07E-18
C4bp	4.45E-43	-0.36	1.00	1.00	1.17E-40

Appendix Table 16 | LRT results for selected zonated immune marker genes (GO:0002376 . immune system processes). Central_sig and portal_sig columns indicate whether including the central respectively portal vein distance covariate significantly improves a reduced model that only uses the intercept and the other distance covariate.

Gene	p-value only central	p-value only portal	p-value only intercept	Central sig	Portal sig	Intercept
Arg1	0.0002	3.33E-05	1.06E-09	TRUE	TRUE	TRUE
C9	0.0012	0.0004	2.60E-07	TRUE	TRUE	TRUE

Hc	0.1794	0.0201	2.60E-07	FALSE	TRUE	TRUE
Psm1	0.8961	0.4787	0.7386	FALSE	FALSE	FALSE
Mbl1	0.4480	0.213E-05	1.99E-05	FALSE	TRUE	TRUE
C4bp	0.0379	0.1573	0.0155	TRUE	FALSE	TRUE

Appendix Table 17 | Results of differential gene expression of hepatic stellate cell (HSC) markers (PMID: 31722201) between cluster 1 and cluster 2.

	p_val	avg_log2FC	pct.1	pct.2	p_val_adj
Ctsc	2.29E-172	1.24	0.95	0.60	4.05E-170
Hsd11b1	8.70E-96	0.45	1.00	1.00	1.54E-93
Ly6e	1.48E-15	0.31	0.82	0.69	2.62E-13
Fgfr2	2.23E-12	0.24	0.32	0.19	3.95E-10
Cp	9.67E-16	0.23	1.00	0.99	1.71E-13
Slc40a1	2.62E-08	0.22	0.60	0.50	4.63E-06
G0s2	3.31E-07	0.20	0.63	0.52	5.86E-05
H2-K1	2.28E-16	0.19	1.00	1.00	4.04E-14
H2-D1	2.15E-04	0.15	0.82	0.77	3.80E-02
Lgmn	2.72E-04	0.14	0.49	0.40	4.82E-02
H2-Q7	1.34E-05	0.13	0.28	0.20	2.38E-03
Rasgrp2	1.09E-07	0.11	0.16	0.07	1.93E-05
Tmem141	2.28E-05	0.10	0.18	0.11	4.04E-03
B2m	1.78E-08	0.08	1.00	1.00	3.15E-06
Tmem47	1.93E-05	-0.08	0.07	0.13	3.41E-03
Lamp2	3.16E-05	-0.09	1.00	1.00	5.60E-03
ApoE	3.07E-19	-0.09	1.00	1.00	5.43E-17
Pam	3.48E-05	-0.10	0.18	0.25	6.16E-03
Calcr1	2.45E-05	-0.12	0.18	0.26	4.33E-03
Acaa2	9.01E-09	-0.14	1.00	1.00	1.59E-06
Tgfbi	7.73E-06	-0.15	0.28	0.38	1.37E-03
Fermt2	1.62E-05	-0.1592554459	0.451	0.533	0.00287230
Agtr1a	4.39E-06	-0.1625722413	0.553	0.644	0.0007774
Ecm1	2.33E-05	-0.1630426595	0.621	0.679	0.00412169
Dnaja1	3.29E-05	-0.16	0.44	0.53	5.82E-03
Abcc9	2.80E-09	-0.1663416632	0.124	0.224	4.96E-07
Eng	4.95E-08	-0.1940398674	0.363	0.476	8.76E-06
Dusp6	3.94E-12	-0.2143529298	0.257	0.4	6.98E-10

Reln	1.72E-14	-0.2542355471	0.335	0.499	3.05E-12
Rspo3	2.89E-30	-0.2669879087	0.051	0.219	5.12E-28
Lifr	2.46E-55	-0.5146051212	0.89	0.974	4.36E-53
Dcn	2.33E-45	-0.5241378902	0.899	0.963	4.12E-43

Appendix Table 18 | LRT results for selected zonated hepatic stellate cell (HSC) markers and Cholangiocyte marker *Spp1*. Central_sig and portal_sig columns indicate whether including the central respectively portal vein distance covariate significantly improves a reduced model that only uses the intercept and the other distance covariate.

Gene	p-value only central	p-value only portal	p-value only intercept	Central sig	Portal sig	Intercept
Hsd11b1	1.24E-05	3.81E-05	5.18E-11	TRUE	TRUE	TRUE
Ly6e	0.0008	0.1644	0.0033	TRUE	FALSE	TRUE
Fgfr2	0.2158	0.8261	0.3985	FALSE	FALSE	FALSE
Rspo3	0.1655	1.01E-05	2.15E-06	FALSE	TRUE	TRUE
Lifr	0.0056	0.0248	0.0002	TRUE	TRUE	TRUE
Dcn	0.1962	0.0787	0.0423	FALSE	FALSE	TRUE
Spp1	1.88E-20	0.9632	1.19E-20	TRUE	FALSE	TRUE

Reviewers' Comments:

Reviewer #1:

Remarks to the Author:

The authors have done a great job in improving their paper. Particularly, the addition of the bivariate model and the thorough analysis of the impact of signaling pathways greatly strengthen the paper. I only have one comment that should be addressed before final acceptance:

The analysis of the NPC markers (Supplementary figures 8, 14) is problematic. The authors find deviations from previous reported zonation profiles but many of these discordances can be explained by the fact that the markers selected are not specific enough, given that the spatial transcriptomics method averages multiple cell types within the same spot. For example, even though Sox4 is pericentrally zoned in HSCs, it is most highly expressed in the liver in cholangiocytes, explaining the portal zonation the authors observe (particularly since cholangiocytes are spatially clustered, leading to dominating effects in portal spots). Furthermore, many of the NPC markers in Supplementary Figures 8 and 14 are expressed at very high levels in hepatocytes, including Arg1 and Apoe. Arg1 is expressed more than an order of magnitude higher in hepatocytes compared to any other NPC. For genes like Apoe, which are expressed in hepatocytes at lower fractional amounts than in immune cells, the mRNA content of each spot would still be dominated by the hepatocytes, as their volumes and mRNA contents are more than 20-fold higher than NPCs (see <https://pubmed.ncbi.nlm.nih.gov/30222169/>). Consequently, any zoned NPC analysis can only be performed on genes that are expressed at lower than 20-fold in hepatocytes compared to the other NPCs (to avoid biases such as for Apoe and Arg1), and expressed at sufficiently high levels in the relevant NPC compared to any other NPC cells type (to avoid biases such as for Sox4). Given the spatial resolution issue, I believe it is better to focus on the very few genes that are massively more highly expressed in the NPC of interest than in any other liver cell type (e.g. see the selection of endothelial genes in <https://pubmed.ncbi.nlm.nih.gov/30222169/>). Alternatively, the authors should remove this analysis altogether and leave it for future work, I think the paper is sufficiently strong without this section.

Reviewer #2:

Remarks to the Author:

With reference to the extensive investigations carried out in the revised manuscript, my questions have also been sufficiently clarified.

Reviewer #3:

Remarks to the Author:

The Reviewer appreciated the authors' hard work and detailed response, but there is still concern about the cell type proportion-related discoveries. The detailed comments are as following:

Comment 1): cell count frequency

It is good that authors estimated the cell count within each spot and excluded the impact of cell count on the clustering. It would also be helpful if authors can discuss or explain how the spot with diameter of 100um can by average host ~50 hepatocytes with diameter of 30um, from the angles of slide thickness, nuclei detection, etc.

Comment 5): Elaborate on statistical method used in Fig 2a)

It is good that authors provided the detailed mathematic derivation, and the Reviewer totally agrees that the calculation of Pearson correlation coefficient does not need to satisfy the normal distribution, except when computing p value analytically. Therefore, it is good that authors alternatively proposed to use bootstrap to evaluate the significance.

However, without assumption of normal distribution, the Pearson correlation coefficients might not be comparable between statistical tests, for example, the r_1 is larger than r_2 , but by bootstrap, the corresponding p_{val1} is less significant than p_{val2} . Moreover, the large Pearson correlation might not even be significant, either. As such, the visualization of Pearson correlation would be

misleading, i.e., Fig2a. The Reviewer would suggest authors directly visualize the signed $-\log_{10}$ p-values to make it easier to interpret.

Btw. Authors mentioned that "Pairs where the confidence interval overlaps with zero, i.e., being non-significant, are indicated with a gray border." But the Reviewer failed to find those borders.

Comment 6): Elaborate/Justify correlation analysis of stereoscope single cell integration

Due to mixed cells in spots, authors used the scRNA-seq-derived signatures to estimate the cell type proportions for each spot, followed by the correlation between those proportions. As mentioned in this paper, "Pearson correlation scores between cell type proportions across the spots show positive correlation, to be interpreted as spatial co-localization of non-parenchymal cells". However, the low values of cell type proportions may be only the noise, that is, those cell types do not exist in the spot. Meanwhile, the correlation herein, may be largely driven by the similarity between the scRNA-seq-derived signatures, the gene sharing or co-expression among signatures. So, correlation may not be because of the real spatial co-localization, but just mathematical similarities. The above possibilities cannot be excluded with no solid validation. To this end, Reviewer suggested that "as a negative control, the same method and same scRNAseq-based signature should be applied to the scRNAseq data." On the one hand, when estimating cell type proportion, the single cell as an individual cell type is supposed to have no small value, but if small values are indeed observed, it may suggest those values estimated by the tool is not confident or just noise; on the other hand, if similar correlation patterns between the spatial transcriptome and the single cell data were observed, it might suggest that the spatial transcriptome data might not provide additional information beyond scRNAseq data, or that the cell type correlation could be simply driven by the correlation or underlying indirect correlation between cell type signatures.

Authors make a modification to the request: "Rather than looking at correlations between gene signatures, we compiled an average expression profile for each cell type in the single cell data (mean of normalized expression values), and produced similar correlation plots as for the spatial data using these representative average profiles."

The Reviewer appreciate the modification. But it still cannot fully address the concern. The cell type proportion is learned from cell type signatures, and therefore, the correlation of cell type proportions is largely driven by cell types signatures i.e., "a small group of genes", while the gene expression correlation between single cell types proposed by authors is based on "the whole transcriptome". The underlying correlation between signatures might be largely diluted by a large number of other genes. Therefore, the difference of two correlation analysis might not result from the difference between gene expression and spatial co-localization (as claimed by the authors), but from the difference between a small group of signatures and whole transcriptome. Without excluding the possibility of the latter, it is still not convincing to reach the conclusion that "the cell type proportion correlation matrix capture the true signals of spatial co-localization and segregation".

Comment 9) Immunostaining of liver tissue

Authors discussed the technical challenge of immunostaining of liver tissue, and clarified some of the concepts, both of which are appreciated by the Reviewer. However, some points in the response are still unclear and unconvincing:

a) The authors mentioned that "The single cell integration performed in this study rather provides information about cell type proportions across the tissue than annotating a cell type to each spot. ... It rather implies that based on the provided single cell data set the spot in question exhibits a proportional similarity to single cell dataset's cell type annotations between 0 and 1 when compared to the remaining tissue."

If the Reviewer understand the response correctly, the estimation in the paper is not the cell type proportion for a given spot but "a proportional similarity to single cell dataset's cell type annotations between 0 and 1", which sounds more like a likelihood. But by looking into the original paper of the tool used in the paper (stereoscope, Andersson et al. Communications Biology, 2019), the Reviewer noticed that the tool aims to estimate the parameter " $W_{sz} = N_{sz}/\sum_z(N_{sz})$ ", where N_{sz} is the number of cells from cell type z at capture location s , and based on W_{sz} , the Pearson correlation was used to estimate the cell-type co-localization. By the definition of the formula, W_{sz} is exactly the proportion of cell count of type z out of all cells within the spot s .

b) The authors mentioned that "Given the comment raised above, the incentive of this study does not include the detailed annotation of cell type distributions within spots and therefore across the

tissue”.

However, according to a), authors indeed estimated the W_{sz} of cell type z within each spot z , inferred the cell-type co-localization, and accordingly, claimed the contribution of spatial transcriptome, which should be one of the major discovery of the paper.

c) Authors mentioned that “we agree with the reviewer that immunostaining against cell types would confirm the general presence of cell types across the tissue. However, the binary results on cell type presence received from immunostaining are not expected to correlate directly with the observed cell type proportions based on gene expression patterns across the tissue [11,12].” It is good that authors agreed with the benefits of immunostaining against cell types. But when claiming the challenge that “immunostaining is not expected to directly correlate with the observed cell type proportions”, two papers cited by the authors are not relevant, both of which talked about the correlation between mRNA and protein levels. Of note, the requested validation is to investigate consistent cell type estimation by two methods: mRNA expression (proposed by authors) and immunostaining (suggested by the Reviewer). It does not necessarily rely on the correlation between the mRNA and protein markers, and even does not require the mRNA and protein of the same gene.

Put together, the clarification and discussion are still not sufficient to convince the Reviewer. The Reviewer suggests authors do the solid independent validation to confirm the cell type-related observation. Two detailed comments are as followed:

a) immunostaining: it is understood that it is technically challenging to do the immunostaining on the same slide section, but it would also be helpful to investigate the serial section.

b) cell count estimation: Immunostaining density of cell types are proportional to the cell count, while stereoscope-estimated cell type proportion within each spot is a percentage, and not comparable to the cell count. However, the total cell count within spots can be estimated (claimed in comment #1), i.e., $\sum_z(N_{sz})$, and therefore, the cell count of cell type N_{sz} can be simply calculated via $N_{sz} = W_{sz} * \sum_z(N_{sz})$.

Thus, the correlation between two independent methods can be calculated. The good correlation will validate the cell type proportion estimation and to a great extent support discovery of cell type co-localization in the paper.

Additional comment 1):

The liver spatial transcriptome data is a good resource for other researchers to explore and utilize. But it seems that the link (<https://zenodo.org/deposit/4399655>) cannot be accessed with permission requested.

a) As for the data sharing, authors are recommended to share the comprehensive data and information, for example, fastq files, raw images, and point out the spaceranger parameter settings, like slide serial, capture area arguments, etc.

b) In addition, the manually annotated and predicted petriportal and petricentral are also necessary.

c) As for the code sharing, authors should provide the code, which can be run on the above data to reproduce the observations in the paper.

ST Liver

-

2nd Revision

Reviewer 1

Comment 1

The authors have done a great job in improving their paper. Particularly, the addition of the bivariate model and the thorough analysis of the impact of signaling pathways greatly strengthen the paper. I only have one comment that should be addressed before final acceptance:

The analysis of the NPC markers (Supplementary Figures 8, 14) is problematic. The authors find deviations from previous reported zonation profiles but many of these discordances can be explained by the fact that the markers selected are not specific enough, given that the spatial transcriptomics method averages multiple cell types within the same spot. For example, even though Sox4 is pericentrally zoned in HSCs, it is most highly expressed in the liver in cholangiocytes, explaining the portal zonation the authors observe (particularly since cholangiocytes are spatially clustered, leading to dominating effects in portal spots). Furthermore, many of the NPC markers in Supplementary Figures 8 and 14 are expressed at very high levels in hepatocytes, including Arg1 and Apoe. Arg1 is expressed more than an order of magnitude higher in hepatocytes compared to any other NPC. For genes like Apoe, which are expressed in hepatocytes at lower fractional amounts than in immune cells, the mRNA content of each spot would still be dominated by hepatocytes (see <https://pubmed.ncbi.nlm.nih.gov/30222169/>). Consequently, any zoned NPC analysis can only be performed on genes that are expressed at lower than 20-fold in hepatocytes compared to the other NPCs (to avoid biases such as for Apoe and Arg1), and expressed at a level not dominated by the hepatocytes, as their volumes and mRNA contents are more than 20-fold higher than sufficiently high levels in the relevant NPC compared to any other NPC cell type (to avoid biases such as for Sox4). Given the spatial resolution issue, I believe it is better to focus on the very few genes that are massively more highly expressed in the NPC of interest than in any other liver cell type (e.g. see the selection of endothelial genes in <https://pubmed.ncbi.nlm.nih.gov/30222169/>). Alternatively, the authors should remove this analysis altogether and leave it for future work, I think the paper is sufficiently strong without this section.

We would like to thank the reviewer for his constructive suggestions and feedback to the revised version of our manuscript and very happy to hear that the vast majority of the additionally performed analyses were satisfactory.

We agree with the reviewer that the mixed cell type population within spots represents a major caveat to reliably define non-parenchymal cell (NPC) zonation within our ST data, as the expression of certain marker genes such as *Apoe* and *Arg1* are not restricted to NPCs but shows high levels of expression in hepatocytes, described in detail in the reviewer's comment. We also

agree with the reviewer that it will be more beneficial and of high interest to address zoned NPC expression profiles in future ST studies. Therefore, to avoid confusion and provide clarity with regard to the conclusions we are able to make in this study, we decided to omit the presented data on NPC zonation in Supplementary Figures 8 and 14 as suggested by the reviewer. However, we believe the analysis of the zonation profile of the GO-term "immune system processes" (GO: 0002376) and metabolic pathways presented in Supplementary Figure 8 and/or Supplementary Figure 14, remains informative, as it not based on specific cell types but a biological process, similar to the general observed zonation between central and portal veins presented in our study. Consequently, we only kept the analysis on the general zonation of immune system processes and metabolic pathways in Supplementary Figure 8 and Supplementary Figure 14, of which we attach the revised versions here as Review figure 1 and Review figure 2. We also revised Supplementary table 5 and the manuscript text accordingly:

Line [228 - 232]:

"Based on these observations, we further investigated the zonation of reported marker genes in the context of reported immune zonation⁴². To this end, we investigated DEGs associated with immune system processes (GO:0002376) and found more genes with periportal than pericentral zonation (Supplementary figure 8)."

Line [381 - 385]:

"While our data does not indicate elevated Kupffer cell proportions in the periportal cluster compared to the remaining clusters, we found more genes related to immune system processes with periportal enrichment in comparison to the pericentral zone providing initial support for implications of previously proposed immune zonation⁴²."

and the methods describing the zonation based differential gene expression from line 615-624.

We thank the reviewer again and hope the implemented modifications are to the reviewers satisfaction and that they could remove any further objections to the data presented in the manuscript.

Supplementary Figure 8: a Heatmap depicting differentially expressed markers, associated with the GO-term "immune system process" (GO:0002376). Genes exhibiting highest elevation in either the portal area (cluster 1) or central area (cluster 2) are surrounded by a red box. These highlighted genes were selected to be analyzed with the bivariate expression by distance model. In **b** selected markers from **a** were subjected to bivariate expression analysis (methods). Numbers in curly brackets after the gene name indicate that 1: the full model does not perform significantly better than the reduced portal model, 2: the full model does not perform significantly better than the reduced central model, 3: the full model does not perform significantly better than either of the reduced models, 4: the full model does not outperform the most reduced model (intercept only), i.e., the gene expression can be taken as constant across the tissue w.r.t vein distances. Relative expression values for each gene are depicted in a color gradient ranging from low (dark) to high (light). The results from the likelihood-ratio tests (LRT) are presented in **Supplementary Table 5**.

a

b

Review figure 1 | Revised version of Supplementary Figure 8.

Supplementary Figure 14: bivariate expression by distance analysis (methods) of selected genes. **a** glucagon target genes, **b** Wnt pathway target genes, **c** Ha-ras target genes, **d** chronic hypoxia target genes and **e** pituitary hormone target genes. For a detailed description of the plot interpretation see **Supplementary Figure 8**. The results of likelihood-ratio tests (LRT) are presented in **Supplementary Table 5**.

a Glucagon target genes

c WNT pathway target genes

d Ha-ras target genes

e Chronic hypoxia target genes

f Pituitary hormone target genes

Review figure 2 | Revised version of Supplementary Figure 14.

Reviewer 2

With reference to the extensive investigations carried out in the revised manuscript, my questions have also been sufficiently clarified.

We are delighted to hear that the reviewer's questions were sufficiently clarified during the first revision. We would like to thank the reviewer again for his/her constructive comments and suggestions during the revision, which we believe have increased the quality of our study substantially.

Reviewer 3

Comment 1 - cell count frequency

It is good that authors estimated the cell count within each spot and excluded the impact of cell count on the clustering. It would also be helpful if authors can discuss or explain how the spot with diameter of 100µm can by average host ~50 hepatocytes with diameter of 30µm, from the angles of slide thickness, nuclei detection, etc.

We are glad to hear that the reviewer appreciates the cell count estimation we performed and agrees with us to exclude this covariate from our normalization and clustering approach.

We are also happy to elaborate on the question asked by the reviewer on the reason for a relatively high average cell density within 100 µm spot diameter. Indeed, when speaking about areas the 50 hepatocytes with an estimated diameter of 30 µm each would not be able to fit into the spot area with a 100 µm diameter.

This can be easily estimated when considering a section being a 2D plane with zero thickness:

Area of one 100 µm spot $3,14 \times (50)^2 = 7853.98 \mu\text{m}^2$

Area of one 30 µm hepatocyte = $3,14 \times (15)^2 = 706.86 \mu\text{m}^2$

This would mean that only 11.11 hepatocytes would fit into the spot.

And the average cell diameter of 50 cells/spot would need to be around 14.14 µm

Or when calculating in more realistic 3D volumes:

Volumes for sphere - for cells : $[V] = 4/3 * \pi * r^3$, where $r = \text{diameter} / 2$.

Volume of a cylinder - for each spot: $[V] = \pi * r^2 * h$, where $r = \text{diameter} / 2$, and h is the height (10µm)

Hepatocytes: $r=15$; $[V] = 14\ 130 \mu\text{m}^3$ needs to be divided by 3 as the tissue (including Hepatocytes) is sectioned at 10µm thickness, thus $[V] = 42\ 390 / 3 = 14\ 130 \mu\text{m}^3$

Tissue over spot: $r=50$; $78\ 500 \mu\text{m}^3$

Which leads to an estimate of ~16,6 hepatocytes per spot.

It should be noted that the true shape of hepatocyte in unaltered tissue is much closer to the liquid crystal shape, rather than a sphere [1]. And that the average volume of the mouse hepatocyte is approximately $8000 \mu\text{m}^3$ [2], with asymmetrical “crystal-like” dimensions of $\sim 30 \times 10 \mu\text{m}$ [1], getting the number of hepatocytes per spot to $\sim 9.8\text{-}28.4$ depending on the angle of the $10 \mu\text{m}$ section and therefore orientation of the cells (Review figure 3).

Review figure 3 | Schematic illustrating the potential effect of **a)** the dimensionality of the liver sections used in our experiments and for cell segmentation and estimation. The nuclei visualized in the images potentially originate from multiple z-planes. In addition cell-size and cytoplasmic size between hepatocytes can differ as illustrated in **b)** which can increase the cell count additionally while simultaneously considering the 3-dimensional structure of the section. $10 \mu\text{m}$ thick sections are indicated as red boxes. Each blue dot denotes a hepatocyte with the surrounding cytoplasm in grey.

It is worth mentioning that the observed 30-50 cells/spot do not only include hepatocytes but also other cell types, such as highly compressed endothelial cells and a variety of other non-parenchymal cells such as Kupffer cells, neutrophils and other blood cells with small diameter (diameter $10\text{-}12 \mu\text{m}$), as we are estimating the total cell count by the presence of nuclei and not only the presence of hepatocytes.

Thus, starting from the range of 30-50 cells per spot we report in the manuscript, the corrected average estimation of hepatocytes per spot should be 21-35, based on the assumption that hepatocytes comprise $\sim 70\%$ of the total liver volume. Nevertheless, we were not satisfied with the argumentation based on available literature, and decided to quantify the cells manually using the consecutive cryosections of the same cryopreserved tissue used for the spatial transcriptomics experiments.

To this end, we performed immunofluorescence assays (IFAs) for established marker proteins of three different cell types. These cell types included: Kupffer cells (F4/80 +), hepatocytes (HNF4a +) and endothelial cells (CD31+). The Kupffer cell marker F4/80 is encoded by *Adgre1*, the hepatocyte marker HNF4a by *Hnf4a* and the endothelial marker CD31 by *Pecam1*. As a result of the technical limitations posed by the limited number of fluorescence filters of the microscope

used in our experiment and the cross-reactivity of some antibody combinations, we were unable to perform multi-immunostainings exceeding two antibodies simultaneously and were limited in the combination of cell type stainings. Thus, we performed a dual-immunostaining of the following cell type combinations:

Kupffer cells and Hepatocytes (F4/80 +, HNF4 α +) as well as Kupffer cells and endothelial cells (F4/80 +, CD31 +). For both assays we also performed a DNA counterstain, using Hoechst (Review figure 4 a,b). 100 μ m spots were randomly assigned across the imaged sections to estimate the total number of cells. In total 224 spots were counted for nuclei and Kupffer cells and 110 and 114 spots were counted for Hepatocytes and Endothelial cells, respectively. Counts were visualised as histograms (Review figure 4 c-f). Quantification revealed the range of total cell counts (Hoechst+) per spot is 10-60 with a mean value of 32.1 cells per spot . For HNF4 α + hepatocytes specifically, these numbers are 2-30 hepatocytes per spot with mean value of 18.3. These numbers are also in strong agreement with the data used for the volumetric calculation [1,2].

Review figure 4 | Quantification of cell type proportions by immunofluorescence assay (IFA). a depicts a representative fluorescence image of a spot used for manual counting of Kupffer cells and hepatocytes. The white circle indicates the area of a spot on an ST array. DNA (left) was stained using Hoechst. Kupffer cells (middle left in green) were counted based on a positive signal for DNA and F4/80 signal. Hepatocytes were counted based on a positive signal for HNF4α and Hoechst signal (middle, right,

red). The merged image to the right depicts the overlay of all three signals within the region of interest. **b** similar to **a**, depicts a representative fluorescence image of a spot used for manual counting of Kupffer cells and endothelial cells in the region of interest (white circle). DNA is represented in blue to the left, Kupffer cells are represented in green (middle left). Endothelial cells were counted based on a positive signal for CD31 and Hoechst signal (middle, right, red). The merged image to the right depicts the overlay of all three signals within the region of interest. **c** shows the distribution of the manual counting of 100 μm spots for nuclear Hoechst staining. 250 spots were randomly selected, 26 spots were excluded due to tissue damage or spots assigned outside of the tissue area, resulting in 224 counted spots. staining for **d** HNF4 α /Hoechst-positive hepatocytes were counted in 110 spots, **e** the Kupffer cells were counted based on a positive signal for DNA and F4/80 signal (224 spots), **f** the endothelial cells counted based on a positive signal for CD31 and Hoechst signal (114 spots).

Thanks to the reviewer's comment we have carefully considered the volumetric calculations $\sim 9.8\text{-}28.4$ hepatocytes/spot, hepatocyte to other cell type average ratio of 60-70% per spot, as well as our new quantifications of stained sections and decided to modify the manuscript in the following way.

We changed the sentence in lines [120 - 121] *"From the hematoxylin-stained nuclei we estimated that a majority of spots contain between 30-50 cells, of which 60-70% are considered to be hepatocytes."* to *"For a select set of cell types, we used immunofluorescence staining to estimate the number of cells present in a subset of projected spot areas in liver cryosections. We performed stainings for nuclei (Hoechst), hepatocytes (HNF4 α), Kupffer cells (F4/80), and endothelial cells (CD31). Quantification of Hoechst+ nuclei revealed the range of cell count per spot is 10-60 cells with a mean value of 32.1 \pm 8.73 cells per spot, out of which 56.9% \pm 15.8% are hepatocytes, 12.7% \pm 7.4% are Kupffer cells, and \sim 30.8% \pm 17.0% endothelial cells (Supplementary figure 21, Supplementary Table 9)"* lines [126 - 132] and added the corresponding figure to the supplementary materials, and all the accompanying information to the methods part of the manuscript in lines [519 - 526].

[1]<https://elifesciences.org/articles/44860>

[2]<https://elifesciences.org/articles/11214>

Comment 5 - Elaborate on statistical method used in Fig 2a)

It is good that authors provided the detailed mathematic derivation, and the Reviewer totally agrees that the calculation of Pearson correlation coefficient does not need to satisfy the normal distribution, except when computing p value analytically. Therefore, it is good that authors alternatively proposed to use bootstrap to evaluate the significance.

However, without assumption of normal distribution, the Pearson correlation coefficients might not be comparable between statistical tests, for example, the r_1 is larger than r_2 , but by bootstrap, the corresponding p_{val1} is less significant than p_{val2} . Moreover, the large Pearson correlation might not even be significant, either.

As such, the visualization of Pearson correlation would be misleading, i.e., Fig2a. The Reviewer would suggest authors directly visualize the signed $-\log_{10}$ p values to make it easier to interpret.

Btw. Authors mentioned that “Pairs where the confidence interval overlaps with zero, i.e., being non-significant, are indicated with a gray border.” But the Reviewer failed to find those borders.

We are delighted to hear that the reviewer agrees with us that the Pearson correlation coefficient is an appropriate measure to apply to our data, despite it (our data) not following a normal distribution, as long as p-values are not computed analytically.

In addition we would like to highlight the fact that we are actually **not computing p-values** in our bootstrap procedure, but rather confidence intervals; though, of course the bootstrap samples could easily be used to produce p-values as well. However, we would like to argue that p-values are not superior to the correlation values due to the following two reasons:

1. **Usefulness** : A p-value is always associated with a hypothesis test, and in the case of the Pearson correlation coefficient the two competing hypotheses would be:
 - $H_0 : r = 0$ (null)
 - $H_A : r \neq 0$ (alternative)

The p-value thus gives the probability of obtaining the observed correlation value if the true population correlation value was actually zero, while the alternative hypothesis states that the true correlation coefficient is not zero. The p-value does not convey information about the magnitude of the effect, i.e., effect size, the only thing it informs us about is **whether an effect is present or not**. A more thorough account of this argument can be found in “*Using Effect Size—or Why the P Value Is Not Enough*” by Sullivan and Feinn.[1] Hence, by definition - the p-value does not convey the information we seek to present to our readers.

2. **Interpretability** : The reviewer states that displaying the $\log_{10}(p\text{-values})$ would increase the interpretability of our results, here we would humbly like to disagree. As mentioned in (1) the p-value only reflects the answer to whether there is an effect or not, meaning that notions of directionality are completely lost, to clarify: an anticorrelation with a p-value of 0.001 would be given the same representation in the suggested approach as a positive correlation signal with a p-value of 0.001. Since an anticorrelation (indicative of spatial segregation) and a correlation signal (indicative of spatial co-localization) have vastly different interpretations, we believe it’s important to include this information of directionality in our results.

The reviewer also states that

“[...]the Pearson correlation coefficients might not be comparable between statistical tests, for example, the r_1 is larger than r_2 , but by bootstrap, the corresponding $pval_1$ is less significant than $pval_2$.”

We believe the two arguments above address this concern, but to give a direct answer: we fail to fully see how the p-values would be more relevant to display — given the character of the

hypothesis test — than the actual correlation values together with information regarding their significance (here presented through the confidence intervals).

Finally, the reviewer mentions how no gray borders can be seen in Figure 2A; in fact, the gray borders are there (Review figure 5), but we fully admit that they are hard to discern from the “normal” borders and easily could give an impression of homogenous border color. Therefore, we’ve replaced the gray coloring with magenta - hoping that this makes the presentation of our results more clear. Revisiting the analysis in detail also revealed a minor mistake in our data, which does not impact the main conclusions of our study and for which we accounted for in the revised figure and by changing the manuscript text in lines [366] - [372] from “*While, our data does not indicate elevated Kupffer cell proportions in the periportal cluster compared to the remaining clusters, we found a number of Kupffer cell marker genes exhibiting portal but also central zonation. In addition, we found more genes related to immune system processes with periportal enrichment in comparison to the pericentral zone and colocalization of neutrophils and periportal hepatocytes, already in unperturbed conditions, all supporting implications of previously proposed immune zonation*⁴⁰.” to “*While our data does not indicate elevated Kupffer cell proportions in the periportal cluster compared to the remaining clusters, we found more genes related to immune system processes with periportal enrichment in comparison to the pericentral zone providing initial support for implications of previously proposed immune zonation*⁴².” lines [381-385]). We thank the reviewer for bringing this to our attention, allowing us to improve our figures. Consequently, we exchanged the original correlation plot in Figure 2a with the updated figure (attached here as Review figure 6, for ease of inspection and adjusted the figure legend as follows:

Line [1054]

“[...] Non-significant correlations are highlighted with magenta borders. [...]”

Review figure 5 | Left: excerpt from the original correlation plot. Right : Excerpt from the updated correlation plot. Red arrow points to the borders in question.

a

Review figure 6 | Updated version of Figure 2a in the revised manuscript.

[1] : <https://doi.org/10.4300/JGME-D-12-00156.1>

Comment 6 - Elaborate/Justify correlation analysis of stereoscope single cell integration

Due to mixed cells in spots, authors used the scRNA-seq-derived signatures to estimate the cell type proportions for each spot, followed by the correlation between those proportions. As mentioned in this paper, “*Pearson correlation scores between cell type proportions across the spots show positive correlation, to be interpreted as spatial co-localization of non-parenchymal cells*”. However, the low values of cell type proportions may be only the noise, that is, those cell types do not exist in the spot. Meanwhile, the correlation herein, may be largely driven by the similarity between the scRNA-seq-derived signatures, the gene sharing or co-expression among signatures. So, correlation may not be because of the real spatial co-localization, but just mathematical similarities. The above possibilities cannot be excluded with no solid validation. To this end, Reviewer suggested that “as a negative control, the same method and same scRNAseq-based signature should be applied to the scRNAseq data.”

On the one hand, when estimating cell type proportion, the single cell as an individual cell type is supposed to have no small value, but if small values are indeed observed, it may suggest those values estimated by the tool is not confident or just noise; on the other hand, if similar

correlation patterns between the spatial transcriptome and the single cell data were observed, it might suggest that the spatial transcriptome data might not provide additional information beyond scRNAseq data, or that the cell type correlation could be simply driven by the correlation or underlying indirect correlation between cell type signatures.

Authors make a modification to the request: "Rather than looking at correlations between gene signatures, we compiled an average expression profile for each cell type in the single cell data (mean of normalized expression values), and produced similar correlation plots as for the spatial data using these representative average profiles."

The Reviewer appreciate the modification. But it still cannot fully address the concern. The cell type proportion is learned from cell type signatures, and therefore, the correlation of cell type proportions is largely driven by cell types signatures i.e., "a small group of genes", while the gene expression correlation between single cell types proposed by authors is based on "the whole transcriptome". The underlying correlation between signatures might be largely diluted by a large number of other genes. Therefore, the difference of two correlation analysis might not result from the difference between gene expression and spatial co-localization (as claimed by the authors), but from the difference between a small group of signatures and whole transcriptome. Without excluding the possibility of the latter, it is still not convincing to reach the conclusion that "the cell type proportion correlation matrix capture the true signals of spatial co-localization and Segregation".

We appreciate that the reviewer is so meticulous in his/her examination of the methods we apply, it's evident that much thought has been given to this issue. We therefore apologize if we've been unclear in our description of *stereoscope* and wish to clarify certain aspects of the method.

Firstly, *stereoscope* **does not operate with any form of signatures** specific to each cell type, as other deconvolution (e.g., MIA by Moncada et al.).[1] Instead, from the single cell data we learn the rate (r) and success probability (p) parameters for a negative binomial distribution describing the expression of all genes included in the analysis, for every cell type, that is:

$$y_{gc_z} \sim NB(r_{gz}, p_g)$$

Where y_{gc_z} is the expression of gene g in a cell (c) of cell type z . Having learnt these parameters from the single cell data, where each observation belongs to a single cell type (no mixing), we then use them to infer the cell type proportions in each spatial location. This is done by leveraging the additive property of the first argument in the NB distribution when the second parameter is shared among all components, that is:

$$x_{gs} = \sum_{c \in C_s} x_{gc} \leftrightarrow x_{gs} \sim NB\left(\sum_z v_{sz} r_{gz}, p_g\right)$$

Where C_s is the set of cells residing at spot s . Here estimates of v_{sz} are obtained through MAP (maximum a posteriori) estimation. The proportion values are then derived from these MAP

estimates. For a full account of the model we refer to the original *stereoscope* publication, which the reviewer seems to be already familiar with.

Secondly, we would like to express our appreciation of the reviewer's previous suggestion, that is to deconvolve the scRNA-seq data with *stereoscope*. However, we must admit that we did not fully understand the request and had no intentions of "modifying" this, but rather thought we acted according to the proposed procedure. Our main confusion stemmed from the fact that we were requested to apply a method designed for deconvolution, to a non-convoluted data set, but understand the reviewer's intention and present analyses in line with this suggestion below. To us, deconvolution of transcriptional profiles obtained from single-cell data would be a task slightly different from what *stereoscope* initially was designed for, and performance would not be fully transferable to the scenario where the data is mixed.

Therefore, in an attempt to show that the correlations we observe are not simply due to similarities between expression profiles among the cell types we generated synthetic *mixed* data based on the single cell data. The synthetic data also highly resembles the real data presented in our manuscript, which we believe makes results from any evaluation using this data more applicable to our discussion. We then analyzed the synthetic data using *stereoscope* in a fashion similar to what we present in the paper. Below, we will first describe the generative process and then continue with an account of our analysis:

Synthetic Data Generation

We here aimed to generate synthetic data with a character similar to the spatial transcriptomics data, but with a random (patternless) distribution of cells across the tissue. Hence, we devised the following procedure to generate such a data set:

1. Sample a number of cells (n_s) that should be present at spot s according to: $n_s \sim Normal(50,10)$. With the additional criteria that n_s needs to be larger or equal to ten; effectively this means that we draw samples until this criteria is fulfilled.
2. Sample n_s members from the Z cell types present in the data, by drawing a vector \mathbf{m}_s from a multinomial distribution according to: $m_s \sim Mult(n_s, p), p = [1/Z, \dots, 1/Z]$. Element z of \mathbf{m}_s (m_{sz}) represents how many cells of cell type z that are found at spot s .
3. From every cell type z randomly pick m_{sz} cells (with replacement), add these to the set of cells belonging to spot s (C_s)
4. Compute the relative abundance (\mathbf{p}_s) of UMIs from each gene within C_s , that is:

$$p_{gs} = \bar{p}_{gs} / \sum_{g'} \bar{p}_{g's} \text{ with } \bar{p}_{gs} = \sum_{c \in C_s} y_{gc}$$
Where y_{gc} is the expression of gene g in cell c .
5. Sample the number of UMI's ($n_{s,UMI}$) that should be observed in spot s from the discrete uniform distribution according to: $n_{s,UMI} \sim IntUnif(2000,10000)$
6. Finally, sample the transcription observed in spot s from a multinomial distribution according to:

$$x_s \sim Mult(n_{s,UMI}, p_s)$$

7. Repeat step 1-6 for every spot in the synthetic data set

Here the cells (c) and their associated expression vectors (\mathbf{y}_c) are taken from the single cell data. The data set we generated consisted of 5000 synthetic spots in total.

Analysis of synthetic data

The generative process produces a set of artificial “spots” where cell types are evenly and randomly distributed, meaning that any form of co-localization analysis should only give very weak signals (due to spurious correlations) and share no obvious similarity with the observed correlations analysis that we present in the manuscript. Therefore, to confirm that this was indeed the case, we first ran our synthetic data set through *stereoscope* using the exact same parameters for the proportion estimation as was used to generate the results presented in the main text. We then performed the same co-localization analysis using bootstrap estimates of the Pearson correlation value. The result from this analysis is presented in Review figure 7, where it is evident that the correlation values from the synthetic data oscillates around zero with no major peaks or troughs, while the corresponding observed values associated with the real data show large fluctuations and much larger magnitudes. Also discernible from Review figure 7 is how the correlation values in the synthetic and observed data are independent of each other, as we expected.

Review figure 7 | Comparison of Pearson correlation values for the real observed (black solid line) and the synthetic data (red dashed line). The y-axis represents the calculated correlation coefficient, while pairs of cell types are found along the x-axis.

We also want to emphasize that several significant signals with a confidence interval excluding zero (i.e., being significant) were observed in the analysis of our synthetic data. Still, we do not

consider this a major flaw of our approach — with large sample sizes even the smallest of effects will be considered significant (including spurious ones) — instead we take this as a strong argument in favor of the choice to mainly focus on effect size in our discussion. As already mentioned and being observable in Review figure 7, the effect sizes are small for all pairs in the synthetic data and would not have been given much attention if they had emerged in analysis of any real data.

We hope that these efforts are sufficient to convince the reviewer that the co-localization patterns are not driven by similarities in gene expression between cell types. A notebook outlining the generation step and the analysis step has been added to the github repository.

[1] <https://www.nature.com/articles/s41587-019-0392-8?proof=t>

Comment 9 - Immunostaining of liver tissue

Authors discussed the technical challenge of immunostaining of liver tissue, and clarified some of the concepts, both of which are appreciated by the Reviewer. However, some points in the response are still unclear and unconvincing:

a) Stereoscope cell type estimation

The authors mentioned that “The single cell integration performed in this study rather provides information about cell type proportions across the tissue than annotating a cell type to each spot. ... It rather implies that based on the provided single cell dataset the spot in question exhibits a proportional similarity to single cell dataset’s cell type annotations between 0 and 1 when compared to the remaining tissue.”

If the Reviewer understand the response correctly, the estimation in the paper is not the cell type proportion for a given spot but “a proportional similarity to single cell dataset’s cell type annotations between 0 and 1”, which sounds more like a likelihood. But by looking into the original paper of the tool used in the paper (stereoscope, Andersson et al. Communications Biology, 2019), the Reviewer noticed that the tool aims to estimate the parameter “ $W_{sz} = N_{sz}/\sum_z(N_{sz})$ ”, where N_{sz} is the number of cells from cell type z at capture location s , and based on W_{sz} , the Pearson correlation was used to estimate the cell-type co-localization. By the definition of the formula, W_{sz} is exactly the proportion of cell count of type z out of all cells within the spots.

We would like to introduce our answer by stating that in the sections of the first revision, referenced by the reviewer in comment a) and b) we were aiming to clarify the same issue. This issue concerned highlighting the difference between cell type annotations in scRNA-seq data and cell type proportion estimations in ST data. In the previous revision, we got the impression that this difference was not clarified sufficiently, which is why we wanted to explain it repeatedly and in multiple ways. However, we realize that we seem to have created more confusion than clarity, which we will try to counteract in our answers to comment 9a and comment 9b in the following paragraphs.

The reviewer is correct in his/her observation that *stereoscope* estimates the proportions of cell types within each spot. We understand that the first statement of our response:

“The single cell integration performed in this study rather provides information about cell type proportions across the tissue than annotating a cell type to each spot. [...] It rather implies that based on the provided single cell data set the spot in question exhibits a proportional similarity to single cell dataset’s cell type annotations between 0 and 1 when compared to the remaining tissue.”

Was not clearly formulated and could be easily misinterpreted, which is why we would like to clarify what the single cell integration analysis performed in our study is able to show and what the current limit of this analysis includes.

- 1) As stated in the original manuscript and in our initial response to the reviewer, *stereoscope* estimates cell type proportions in spots across the tissue. These proportions are obtained using MAP (maximum a posteriori) estimation as described in more detail in the response to comment 6 and in the original publication of the method. The resulting proportion values **do not** represent a similarity score to the single cell data, but rather the estimated fraction of cells residing at a given spot that belong to each cell type. These fractions range from 0, indicating that the cell type is not present in a spot, and 1, indicating the spot only contains this cell type. We apologize for creating this confusion and hope we are able to clarify it here.
- 2) With our statement -
“The single cell integration performed in this study rather provides information about cell type proportions across the tissue than annotating a cell type to each spot.”
- we sought to emphasize that the *stereoscope* method applied in our manuscript **does not** assign a single cell type to the spots in the tissue (“hard classification”), but rather provides a form of “soft classification” where proportion estimates of each cell type at every location are presented. Something we were keen on clarifying, as the reviewer was concerned about the resolution of the method presented in our manuscript.

We apologize for creating this confusion and hope our answer clarifies our previous statements sufficiently and relieves the reviewer of his/her concerns about the correct estimation of cell type proportions in our study.

b) Cell type annotation

The authors mentioned that “Given the comment raised above, the incentive of this study does not include the detailed annotation of cell type distributions within spots and therefore across the tissue”.

However, according to a), authors indeed estimated the W_{sz} of cell type z within each spot z , inferred the cell-type co-localization, and accordingly, claimed the contribution of spatial transcriptome, which should be one of the major discovery of the paper.

We appreciate the reviewer's comment and thank the reviewer for drawing our attention to the potentially confusing statements in our previous reply. The reviewer is correct in this statement that we are estimating the cell type distribution within every spot and therefore implicitly across all spots. This is achieved by using the *stereoscope* tool, described more thoroughly in our response to comment 6.

With our comment:

“Given the comment raised above, the incentive of this study does not include the detailed annotation of cell type distributions within spots and therefore across the tissue”

We sought to express that the incentive of this study **does not include** the assignment of a single cell type to each individual spot and would like to refer the reviewer to our answer to comment 9a for a more detailed explanation.

We hope we were able to explain our argumentation for the integration of scRNA-seq data in our study sufficiently and that this response removed the previously introduced confusion on the interpretation of our discoveries.

c) Immunostaining of consecutive/serial liver sections

Authors mentioned that “we agree with the reviewer that immunostaining against cell types would confirm the general presence of cell types across the tissue. However, the binary results on cell type presence received from immunostaining are not expected to correlate directly with the observed cell type proportions based on gene expression patterns across the tissue [11,12].”

It is good that authors agreed with the benefits of immunostaining against cell types. But when claiming the challenge that “immunostaining is not expected to directly correlate with the observed cell type proportions”, two papers cited by the authors are not relevant, both of which talked about the correlation between mRNA and protein levels. Of note, the requested validation is to investigate consistent cell type estimation by two methods: mRNA expression (proposed by authors) and immunostaining (suggested by the Reviewer). It does not necessarily rely on the correlation between the mRNA and protein markers, and even does not require the mRNA and protein of the same gene.

Put together, the clarification and discussion are still not sufficient to convince the Reviewer. The Reviewer suggests authors do the solid independent validation to confirm the cell type-related observation. Two detailed comments are as followed:

We appreciate the reviewer's suggestion to perform immunostaining on liver sections consecutive to the sections used in our study in order to confirm cell type related observations. We agree with the reviewer that immunostainings would be supportive in validating consistent cell type estimations. However, the request is challenging since we do not have access to consecutive liver sections linked to our original sections used in this study, but merely sections in close proximity to one of our reference sections.

As pointed out by reviewer 1, *“The analysis of the NPC markers (Supplementary Figures 8, 14) is problematic. The authors find deviations from previous reported zonation profiles but many of these discordances can be explained by the fact that the markers selected are not specific enough, given that the spatial transcriptomics method averages multiple cell types within the same spot”*. In accordance with reviewer #1’s suggestion we have decided to omit the presented data on NPC zonation in Supplementary Figures 8 and 14, since this data is outside the major scope of this study, and we agree with reviewer #1 that this is better suited for a future study.

Nevertheless, to accommodate the reviewer’s suggestion we have performed the requested immunostainings of markers related to zonation and selected cell types and discuss our results and conclusions in light of our manuscript.

a) immunostaining

It is understood that it is technically challenging to do the immunostaining on the same slide section, but it would also be helpful to investigate the serial section.

First, we produced immunostainings of sections from the remaining tissues used in our ST experiments. It is important to note, that the number of consecutive high quality sections was limited and as distance to the reference sections increased, differences in morphology became more pronounced. To illustrate this observation further, the sections of sample 1, section 1 and sample 2, section 3 used in our study originated from the same tissue but differ considerably in their morphology. This is especially true when comparing the organization of venous structures (central and portal veins) across the tissue. We attached the H&E images of two representative sections to demonstrate this argument in Review figure 8. However, immunofluorescence stainings for the caudate lobe were in close enough proximity to section 3, sample 2 (Review figure 9, right). Therefore, our results will be compared to this section.

As stated above, the differences in tissue morphology between the tissue sections used in our experiments changed rapidly with increasing distance to each other and due to the three dimensional structure of the tissue, fine structures can easily escape out of the field of view. An additional factor to consider is the technical difficulty of sectioning and placement of the tissue, which represented a major challenge. For instance, the delicacy of the tissue was increased by repeated cryo-sectioning procedures and thus temperature changes of the specimen. Further, the reproduction of the same sectioning angle after refreezing the sample was technically challenging.

In brief, to perform the cross-validation requested by the reviewer, we performed immunofluorescence staining assays (IFAs) for the established central vein marker glutamine synthetase (GS), expressed by pericentrally located hepatocytes and the SOX9 transcription factor expressed by cholangiocytes in the bile ducts accompanying the portal veins [1,2] as well as a nuclear counterstain of 10 μm sections of the remaining tissue.

To visualise the positions of the veins in the tissue as well as to compare them to the manually and computationally annotated veins, we performed the immunostaining of central veins and

periportally-located bile ducts in consecutive sections and overlaid the images based on the nuclear counterstain. Generally, the GS staining resulted in a stronger signal than the SOX9 staining, which made it easier to identify central veins with high confidence. Despite the weaker SOX9 signal, we were able to show that both antibodies marked the associated vein type exclusively. (Review figure 9b).

Despite the fact that tissue sections of the ST experiments and IFAs are not fully overlapping, larger structures can be assessed and compared more easily and accurately. Here, the largest annotated venous structure in the investigated H&E images of the ST sample can be compared to the largest structure observed in the immunostained image (Review figure 9 a,b, white box). Interestingly, the manual and computational annotation differ for this structure, as it is annotated as a portal vein in the manual annotation and predicted to be a central vein by the computational annotation (Review figure 9a). The comparison to the immunological staining of this structure suggests that this structure most likely is a central vein, supporting the computational annotation of the structure. The immunological stainings of the proximate sections also agree with the computational predictions for the two ambiguously annotated veins in the sample (Review figure 9 a,b white box).

In other cases, the stainings are in agreement with the manual annotation over the computational predictions and some structures in the tissue, including mainly structures which have been annotated as portal veins in the computational annotation and which are mostly located close to the edges of the tissue, were not at all detected in the immunological stainings. (Review figure 9 a,b). There are several possible explanations for this observation. Structures close to the tissue edges are smaller and more delicate and thus more prone to escape the field of view. Further, the efficiency of immunological staining across the tissue is oftentimes not entirely uniform and general specificities of the antibody to the epitope may differ, thus smaller structures might not be stained sufficiently to be visible using the given magnification and resolution in our experiments.

The immunological staining also shows that the large identified central vein is in direct proximity to a portal vein, which was not identified as separate blood vessel in the manual annotation and is masked by the central vein profile in the computational annotation, as it not annotated as a separate vein and in very close to the larger identified central vein. For ease of inspection we enlarged the structure in Review Figure 10. The close proximity of central and portal veins presented here, illustrates a major difficulty of manual as well as computational annotations, as this produces mixed portal and central vein "signals" in the structures' neighborhoods for computational annotation.

This observation also highlights that the vein structures must be identified *a priori* to their computational annotation, i.e., *our* classifier cannot discriminate between veins and other parts of the tissue. In this case, central, portal or ambiguous veins were selected based on the H&E image. Some veins may be challenging to detect in an H&E image, while they are easily defined using immunological stainings. This observation represents an additional potential explanation for discrepancies between the annotations in the ST slide and the immunological staining

presented here. Thus, it will be desirable to generate unsupervised computational annotations in future experiments, which does not necessitate manual annotations of the new tissue specimens.

Conclusively, the results of the immunological staining of serial sections requested by the reviewer here largely support the predictions of the computational vein annotations in our study. However, even though we agree with the reviewer that stainings of consecutive sections are informative, we believe that the serial sections presented here are already too distant from the sections used in our original experiments to consider them as ground-truth of the venous architecture let alone detailed cell type observations within these structures.

Nevertheless, we hope we were able to address the reviewer's comment sufficiently and that the reviewer considers the staining of the sections presented here convincing. As we find that these results of the orthogonal validation, suggested by the reviewer, are highly informative, we decided to include these results in the original manuscript. The additional data from Review figure 9a and Review figure 9b in Supplementary figure 22.1-22.2 of the original manuscript and refer to the images in the main text as follows:

Line [316] - [319]

"For proximate tissue sections of selected samples, we also show that the majority of computational predictions is supported by immunofluorescence staining for the respective central and portal protein markers GS and SOX9, serving as an orthogonal validation of our results (Supplementary figure 22.1 - 22.2)."

caudate lobe

a

sample1, section 1

b

sample 2, section 3

Review figure 8 | Comparative HE image of distant sections of the same caudate lobe. To the left, a representative section of sample 1, which includes three sections in close proximity to each other and were placed on the same slide. The right depicts a representative section of sample 2, sectioned from the same tissue as shown to the left but in a more distant location.

Review figure 9 | Orthogonal validation of central and portal veins of proximate sections to tissue sections used in Spatial Transcriptomics experiments via IFA.

Review figure 10 | Enlarged view of orthogonal validation of selected central and portal veins of images shown in Review figure 9. For ease of inspection the figure shows an enlarged view of the area marked with a white box in Review figure 9. **a** depicts the overlay of the immunological staining for GS (left) and SOX9 (right) on the computationally annotated section 3 of sample 2(caudate lobe). **b** shows the same structure in the immunologically stained images only, with the nuclear staining (left), the staining for central veins (GS, middle left), bile ducts (SOX9, middle right) and the overlay of all stainings (merged, right).

[1] [https://www.gastrojournal.org/article/S0016-5085\(09\)00300-X/fulltext](https://www.gastrojournal.org/article/S0016-5085(09)00300-X/fulltext)

[2] <https://www-sciencedirect-com.ezp.sub.su.se/science/article/pii/S001216061500281X>

b) cell count estimation

Immunostaining density of cell types are proportional to the cell count, while stereoscope-estimated cell type proportion within each spot is a percentage, and not comparable to the cell count. However, the total cell count within spots can be estimated (claimed in comment #1), i.e., $\text{sum}_z(\text{Nsz})$, and therefore, the cell count of cell type Nsz can be simply calculated via $\text{Nsz} = \text{Wsz} * \text{sum}_z(\text{Nsz})$.

Thus, the correlation between two independent methods can be calculated. The good correlation will validate the cell type proportion estimation and to a great extent support discovery of cell type co-localization in the paper.

Assuming we understand the reviewer's comment correctly, we agree that computing correlation coefficients between cell type proportion estimates obtained from independent methods - namely *stereoscope* proportions and immunostaining assays - would serve to support

our discoveries of cell type co-localization signals.

We appreciate the reviewer's suggestion to apply the formula to compare proportion estimates between different methods. Simultaneously, we would also like to point out that this comparison is accompanied by several challenges which we will elaborate on in further detail throughout the following paragraphs.

To be able to assess cell type correlations across spots and therefore estimate co-localization, we performed dual-immunostainings of two cell types in each immunofluorescent assay as described in detail in our answer to comment 1. To be able to calculate reliable proportions we only considered images from the same antibody combination.

F4/80 and CD31 are both expressed in the cytoplasm of the respective cell type, while HNF4a is present in the nucleus of hepatocytes. The inconsistent distribution of the signal and its intensity as well as the variable size of endothelial and Kupffer cells and the potential presence of staining artifacts made it difficult to adapt the cell segmentation strategy, which was only designed to detect nuclei and not cytoplasm, which is why we decided to perform manual counting instead.

From the manually estimated cell counts we calculated the proportion of cell type z in spot s (w_{sz}) according to:

$$w_{sz} = n_{sz} / \sum_k n_{sk}$$

Where n_{sz} is the number of cells of cell type z located at spot s .

We then compared the proportion estimates of the manual counting with the *stereoscope* proportion estimates of **section 3 of sample 2**, which exhibits the shortest distance and thus similarity to the sections used in the IFAs. We could observe similar frequencies of proportion values across spots between methods for hepatocytes proportions, with higher proportions in the *stereoscope* estimates, in some cases constituting all cells present in a spot indicated by a the maximal value of 1 (Review figure 11a, left). For Kupffer- and endothelial cells we observed a difference of a magnitude of approximately 10-fold between proportions of the two methods (Review figure 11a + b).

We explain the observed difference in magnitude of the calculated proportion values between immunofluorescence and *stereoscope* data by the nature of our data. In brief, *stereoscope* estimates cell type proportions using annotated single cell data sets. It is possible that in our ST data, the NPCs' expression profiles constitute a very weak signal due to their low abundance and lower transcriptional activity in comparison to for example hepatocytes. This means that their presence easily could be underestimated by methods based on gene expression. For instance, hepatocytes which cover the majority of the surface in a spot and are present in high numbers are also highly metabolically active, resulting in a high number of hepatocyte specific transcripts. Other non-parenchymal cells (NPCs), such as Kupffer- and endothelial cells on the other hand are smaller in size, lower in number and expected to exhibit less activity, especially in the naive, i.e. unperturbed state for Kupffer cells. Hence, transcripts solely expressed by

NPCs are expected to be present at a much lower number than the average hepatocyte marker transcript in the original sequencing library [1]. This in term can lead to a potential underestimation of actual NPC proportions on the transcriptional level and in consequence lead to an underestimation of NPC numbers within a spot.

However, when we calculated the correlation between Kupffer cells and hepatocytes from the proportions of the immunofluorescent assays (IFAs) and compared it to the correlation from stereoscope proportion estimates of the reference section (section 3 of sample 2), we observed a correlation of -0.092 for IFAs with a p-value of 0.39 after bootstrapping. This indicates that these cell types do not inhabit a significant spatial correlation in either direction, i.e . spatial segregation or spatial co-localization (Review figure 11d). For the stereoscope proportions of section 3 we can perform a more distinguished correlation analysis of the reference section and generally see few significant correlations between individual cell types, including no significant correlation between Kupffer cells and any of the different types of hepatocytes (Review figure 11c), supporting the observed correlation trend of Kupffer cells and hepatocytes in immunostainings.

We observe a similar trend for the correlation between endothelial and Kupffer cell proportion values in the *stereoscope* results of section 3, where endothelial cells show almost no significant correlation to any of the remaining cell types. If significant correlations are observed, the effect size is very small with values around 0 (Review figure 11c). In contrast to this observation we observed a significant negative correlation between endothelial- and Kupffer cell proportions in the manual derived proportion estimated of the IFAs of sections in proximity to the ST experiment (Review figure 11e). In the correlation analysis presented in Figure 2a of the manuscript (Review figure 6 for ease of inspection), we observe a positive correlation trend between Kupffer cells and endothelial cells. It is important to stress that the proportion values in the original manuscript include the averaged proportion estimates of cell types across **all eight sections** of our data. The estimated cell type proportions and corresponding correlations presented here represent only individual sections and therefore too few data points to consider most cell type correlations as significant. The celltype proportion correlations in the manuscript (Figure 2a, Review figure 6), similar to our statement in the original manuscript on the description of the novel structure in cluster 5:

Line [452 - 454]

“Considering the sample size used in this study, we can provide initial indications rather than general claims of the function of this proposed structure.”

rather serve the purpose to provide a valuable resource for general trends of cell type proportions in liver tissue than general statements.

Thus, the observed differences in correlations between NPCs, namely Kupffer cells and endothelial cells can most likely be explained by differences of NPC distributions between individual tissue sections and differences in sample size. In more detail the variations in NPC distributions between tissues might arise by : 1) technical reasons based on differences in

capture efficiency across spots and staining efficiency, as explained in extensive detail in the first revision and briefly in the following paragraphs and/or II) biological differences based on differences in distribution of NPCs between sections.

As shown in our calculations in comment 1, hepatocytes in our tissue compose 56.9% +/-15.8% of cells in liver tissue while NPCs compose the remaining cells, which aligns well with previously reported numbers [2,3]. As described in our manuscript we also find a more widespread and scattered distribution of NPCs in liver tissue based on the expression of selected marker genes:

Line [171 - 173]

“These results demonstrate that highly abundant, or bigger cells are widespread, while smaller and rarer cell types are found more scattered across the liver tissue.”

In this context, in our response to comment a) we mention that tissue morphology differs upon increasing distance to the reference section which would also affect cell type compositions. Therefore, we don't consider it surprising that some sections exhibit spatial segregation between NPCs while others exhibit spatial correlation or no spatial distribution of cells.

The investigation of a potential underlying phenomenon of differences in NPC distributions between sections is without doubt interesting but beyond the scope of this study. As we already stated in the previous revision; at the current state of the technology, we do not consider the estimation of cell counts in the ST data as robust enough to include conclusions drawn from this data into the overall conclusion of our manuscript.

Nonetheless, we are grateful for the reviewer's suggestion and consider the results presented here to be informative and the research questions highly relevant in general. However, we believe that the inclusion of the results presented in this revision exceeds the scope and the research questions we aim to address within our study. Including these will - in our opinion - be more confusing than clarifying to the reader in the context of our main findings and discussion. We believe the detailed investigation of the reviewer's questions on cell type proportions or cell count estimates of different cell types is more suitable for future studies using additional and/or different methods, especially in regard to the study of NPCs. This we also emphasize in the conclusive remarks of the discussion in the original manuscript:

Line [465 - 466]

“With expected future advances in the spatial genomics field, increased resolution will promote detailed investigations of rare cell types in tissue space.”

We hope that our response is to the reviewer's satisfaction, that the results presented here suffice as the requested external validation of our description of cell type co-localization and relieved the reviewer from his/her remaining concerns.

Review figure 11 | Estimation of cell type proportions by independent immunofluorescence assay (IFA) and correlation with stereoscope proportion estimates. **a** shows the frequency (y-axis) of stereoscope proportion estimates (x-axis) across spots for sample 2 section3 of the ST for hepatocytes (left), Kupffer cells (middle) and endothelial cells (right). **b** shows the frequency (y-axis) of proportion estimates from manual counting (x-axis) across spots for immunostained tissue for hepatocytes (left), Kupffer cells (middle) and endothelial cells (right). **c** depicts cell type co-localization by Pearson correlations **exclusively** in section 3 of sample 2 of the ST data (right), red arrows mark cell types which we considered for comparison, as these cell types are expected to express the immunological markers used for the performed immunostainings (HNF4a for hepatocytes, F4/80 for Kupffer cells and macrophages and CD31 for endothelial cells). **d** depicts correlation tables with the respective p-value of the calculated correlation for proportion estimates of manually counted hepatocytes and kupffer cells and **e** depicts correlation tables with the respective p-value of the calculated correlation for proportion estimates of manually counted endothelial- and kupffer cells.

[1] <https://www.ncbi.nlm.nih.gov/pmc/articles/PMC6546596/>

- [2] <https://pubmed.ncbi.nlm.nih.gov/2472341/>
[3] <https://pubmed.ncbi.nlm.nih.gov/20159590/>

Additional comment 1 - data availability

The liver spatial transcriptome data is a good resource for other researchers to explore and utilize. But it seems that the link (<https://zenodo.org/deposit/4399655>) cannot be accessed with permission requested.

- a) As for the data sharing, authors are recommended to share the comprehensive data and information, for example, fastq files, raw images, and point out the spaceranger parameter settings, like slide serial, capture area arguments, etc.
- b) In addition, the manually annotated and predicted periportal and pericentral are also necessary.
- c) As for the code sharing, authors should provide the code, which can be run on the above data to reproduce the observations in the paper.

We would like to thank the reviewer for drawing our attention to the incomplete data and code availability. We agree with the reviewer that all data should be shared in open access format to ensure maximal data transparency and reproducibility and apologize for any missing information. Therefore we would like to provide a more detailed description on where and how the data used in this study can be accessed by the reviewer and any other researcher who is interested in our data. We updated the zenodo link and changed it in the original manuscript to a new repository link: <https://zenodo.org/record/5045689> [1].

Line [925 - 928]

“The datasets generated during and/or analyzed during the current study are available in the doi-minting zenodo repository “Spatial Transcriptomics to define transcriptional patterns of zonation and structural components in the liver” and can be accessed at <https://zenodo.org/record/5045689>.”

This link includes the H&E stained images, scaled to 10% of the original size as used in our analysis and (as per request in additional comment 1a) we added an additional folder containing the raw H&E images as well as the manually annotated periportal and pericentral images (as per request in 1b). We would also like to highlight that the created masks, necessary for any expression by distance analysis, are stored at 10% of the original size in the same Zenodo repository. We provide a detailed description of the data structure in the Zenodo repository and hope the reviewer will find this description satisfactory. In light of this revision we also included the original fluorescence images described and analyzed to answer comment 1 and comment 9.

The fastq files, requested by the reviewer can be found on the Gene Expression Omnibus (GEO) database, which complies with data availability requirements of this journal, and has the accession code GSE165141: <https://www.ncbi.nlm.nih.gov/geo/query/acc.cgi?acc=GSE165141> [2]. We thank the reviewer for drawing our attention to this missing reference in the data

availability section of the manuscript. We now have changed this part of the manuscript accordingly.

Line [928 - 930]

“The raw expression data and spot files can be accessed at the Gene Expression Omnibus database with the accession code GSE165141 (<https://www.ncbi.nlm.nih.gov/geo/query/acc.cgi?acc=GSE165141>).”

We also included files containing the selection of spots under the tissue from the spot detection output [3], described in the methods section of the main manuscript between line 533 and line 541 in the metadata of the GEO database accession code GSE165141, as our data is not generated by using *spaceranger* but the *STpipeline*, as described in the materials and methods section between line 546 and 550 of the manuscript and in more detail in [4] as well as in the repository of the package [5].

For the request of the reviewer to code availability, we would like to refer him/her to the Github repository (<https://github.com/almaan/ST-mLiver>), containing all necessary scripts to reproduce the results presented in our manuscript, which we also refer to in the code availability section in our manuscript between line [924] and line [934]. In this Github repository, for which we included an additional doi minted repository (DOI: 10.5281/zenodo.5517601), we included a detailed description of the structure of the repository and reference to external repositories containing necessary data for the analysis as well as the usage of the *hepaquery* package.

We hope this - in our opinion - comprehensive data and code source is to the reviewer's satisfaction and would like to ask him for any further specific missing data or code for the reproduction of the observations made in our study, which we will provide with pleasure.

[1] <https://zenodo.org/record/5045689>

[2] <https://www.ncbi.nlm.nih.gov/geo/query/acc.cgi?acc=GSE165141>

[3] https://github.com/SpatialTranscriptomicsResearch/st_spot_detector_singularity

[4] <https://academic.oup.com/bioinformatics/article/33/16/2591/3111847>

[5] https://github.com/SpatialTranscriptomicsResearch/st_pipeline

[6] <https://github.com/almaan/ST-mLiver>

Reviewers' Comments:

Reviewer #1:

Remarks to the Author:

The authors have addressed all of my concerns, the paper in my view is ready for publication.

Reviewer #3:

Remarks to the Author:

The paper has been significantly improved and more convincing with the substantially additional efforts, especially the immunofluorescence assays of cell types and veins. The Reviewers has only a few comments before final acceptance.

1. spot size:

The spot size is claimed to be 100nm in diameter in this paper, but by checking the 10X Genomics documents, the spot is ~50nm in diameter (55nm). Not sure if this is the version issue of spatial transcriptome, but the Reviewer would like to confirm with the authors, since a few analyses in this paper were based on the spot size, which may largely influence the result, e.g., cell counting.

2. Data access:

By checking the data, a few problems were seen, which authors should address before publication:

a) /sample_2/img/high_resolution/CN16_Liver_HE_E2.jpg seems not the high-resolution image due to very small file size.

b) supplementary Figure 2, the images do not match the data. The sample 2 and sample 3 seem to swap.

c) in /Immunofluorescence, the Reviewer saw the folder of C1, C2, D2, E2, but failed to find the sample id, indicating which sample they are from.

d) the files in /Immunofluorescence/Celltypes /manual_counting are all empty in zero file size. Authors might fail to upload.

e) Not sure if the images were modified. The area beside the tissue is blurry, and the four boundaries of spots for image alignment seems missing. Please refer to the images of "Spatial Gene Expression" from the link: <https://www.10xgenomics.com/resources/datasets>

Spatial Transcriptomics to define transcriptional patterns of zonation and structural components in the mouse liver

Hildebrandt et al.

-
3rd revision

Reviewer 1

We would like to thank reviewer 1 again for his contribution to our manuscript. We are delighted to hear that we were able to address all of his concerns and that he considers our manuscript publication ready.

Reviewer 3

First, we would like to thank the reviewer for their meticulous revision of our manuscript. We believe that the reviewer's suggestions helped improve our manuscript substantially and are glad the reviewer is convinced by the additional results we provided in response to his/her review. We further addressed all additional minor comments the reviewer has in the following.

1. Spot size

The spot size is claimed to be 100nm in diameter in this paper, but by checking the 10X Genomics documents, the spot is ~50nm in diameter (55nm). Not sure if this is the version issue of spatial transcriptome, but the Reviewer would like to confirm with the authors, since a few analyses in this paper were based on the spot size, which may largely influence the result, e.g., cell counting.

We appreciate the reviewers attention to detail in our manuscript. Assuming the reviewer refers to a diameter of 50 μm (not 50 nm), he/she is correct that the spot diameter for a 10X Genomics Visium slide is approximately 50 μm in diameter.

However, as also correctly assumed by the reviewer, we are not using 10X Visium in this manuscript but the previous version of the Spatial Transcriptomics protocol, which has a spot diameter of 100 μm , and a minimal distance of 100 μm and maximal distance of 150 μm between spot centers [1,2]. Therefore, all conclusions in the manuscript which are dependent on the spot size on the array are accurate.

[1] <https://www.sciencedirect.com/science/article/pii/S0092867420306723?via%3Dihub>

[2] <https://bmcgenomics.biomedcentral.com/articles/10.1186/s12864-020-6631-z>

2. Data access

By checking the data, a few problems were seen, which authors should address before publication:

a) `/sample_2/img/high_resolution/CN16_Liver_HE_E2.jpg` seems not the high-resolution image due to very small file size

We thank the reviewer for drawing our attention to the incorrect upload of a low resolution image instead of the high resolution image. We corrected for the mistake and updated the zenodo repository including the image with the image of the correct resolution.

The updated data can be found in a new version of the zenodo repository (doi: 10.5281/zenodo.5595907) [1].

[1] <https://zenodo.org/record/5595907>

b) supplementary Figure 2, the images do not match the data. The sample 2 and sample 3 seem to swap.

We thank the reviewer for drawing our attention to the observed inconsistency in the data description. We corrected the observed error in the zenodo repository by renaming the folders from “sample_2/” to “sample_3/” and from “sample_3/” to “sample_2/” , so that they match the corresponding data.

c) in `/Immunofluorescence`, the Reviewer saw the folder of C1, C2, D2, E2, but failed to find the sample id, indicating which sample they are from.

If we understand correctly, the reviewer failed to correlate the images of the immunofluorescence images in `/immunofluorescence` to the images in the `/immunofluorescence/manual_counting` folder. We apologize for any confusion and hope that we are able to clarify in the following.

In the case the reviewer is under the impression that the sample ids of the immunofluorescence images and the ST experiments are the same, we would like to emphasize that the sample ids (sample1, sample 2 and sample 3) correspond to the corresponding ST experiment and **not** the tissue samples used for immunofluorescence stainings.

In more detail, we used the same frozen tissue of the caudate liver lobe to generate ST sample 1 and ST sample 2 . To generate the immunofluorescence images we used this frozen caudate liver lobe again (as for sample 1 and sample 2 of the ST experiments) but the sections were in another position within the tissue (Review Figure 1a).

However, the sections used in the immunofluorescence assay are **in closest proximity to sample 2** of the ST experiments. The file descriptions “C1, C2, D2, E2”, refer to the position on

the slide used for imaging. C1 represents the section on the top left, while E2 represents the section on the bottom right (Review Figure 1b).

To clarify the description and relation to the results and images of the images in the manual counting folder we changed the file names in the zenodo folder “/immunofluorescence” from “C1”, “C2”, “D2”, “E2” to “caudate_F480_green_HNF4a_red_C1”, “caudate_F480_green_CD31a_red_C2”, “pcaudate_F480_green_CD31a_red_D2”, “caudate_F480_green_HNF4a_red_E2”. We use the same description of the selected spots of the generated overlay of the spot coordinates described in the manuscript (line [124] to line [130]) and our previous response to the reviewer. Each image in “/manual_counting” refers to the position of the selected region in the respective tissue (Review figure 2). We added the following description in the zenodo repository to clarify the content of the respective folder:

“The Celltypes folder contains a folder with the original image of each section of the IFA and a subfolder “manual counting”, with the randomly selected 100 μm openings on each of the corresponding images. The 100 μm openings are adapted from the grid of the ST experiments (Supplementary figures 1-2).”

We hope this change clarifies the file description and makes the data easier to access for the reviewer.

The updated data can be found in a new version of the zenodo repository (doi: 10.5281/zenodo.5595907) [1]

Review Figure 1 | Schematic representation of sectioning and experimental setup for ST and IFAs. **a** Schematic representation of sectioning progression for the tissue of the caudate lobe used for ST (sample 1 and sample 2), sample 2 was obtained from tissue further into the tissue block in the direction of sectioning (indicated by the grey arrow on the bottom right). The sections used for IFAs were obtained from positions further down the sectioning direction. **b** Schematic representation of slide setup with the first section placed on the top left (C1) and the last section on the bottom right (E2). arrows indicate the direction of section placement on the slide.

Review figure 2 | Example for generation of spots for manual counting of cells of IFA validations. As described in the methods section of the main manuscript and our previous reply to the reviewer, we performed additional IFAs to perform manual counting of cells and cell types. We overlaid a grid corresponding to the ST setup (including coordinates) on the image and randomly selected up to 100 spots per image for manual counting. The left image shows the sample “caudate_F480_green_CD31a_red_C2”, where stainings for Kupffer cells (F4/80), endothelial cells (CD31) and nuclei (Hoechst) were performed. Randomly selected spots are highlighted with white circles and spot-coordinates are depicted above each spot. The right panel shows an excerpt of the tissue (white box in the left panel), with an enlarged view of selected spots and their corresponding coordinates in the grid.

[1] <https://zenodo.org/record/5595907>

d) the files in /Immunofluorescence/Celltypes/manual_counting are all empty in zero file size. Authors might fail to upload.

We thank the reviewer for noticing the missing files in the folder: “/Immunofluorescence/Celltypes/manual_counting” of the zenodo repository. We apologize for failing to provide the files and have corrected this error now in an updated version of the repository (doi: 10.5281/zenodo.5595907) [1].

[1] <https://zenodo.org/record/5595907>

e) Not sure if the images were modified. The area beside the tissue is blurry, and the four boundaries of spots for image alignment seems missing. Please refer to the images of “Spatial Gene Expression” from the link: <https://www.10xgenomics.com/resources/datasets>

As stated in our response to comment 1, we did not use 10x genomics Visium slides to study spatial gene expression in our manuscript but the previous Spatial Transcriptomics protocol. Image alignment was performed as described in the materials and methods section “Spot visualization and image alignment” of our manuscript between line [553] and line [561] of our manuscript. In brief, spots were hybridized with fluorescently labeled probes for staining and imaged [1]. The brightfield images of the tissue slides and the fluorescent spot image were then loaded and aligned in the ST Spot Detector tool [2].

Therefore, we can relieve the reviewer from his/her concern that the Hematoxylin and Eosin (H&E) images were modified and can assure him/her that these are the original H&E images, as this version of the Spatial Transcriptomics slides uses different slides and a different alignment method than the slides the reviewer is referring to in <https://www.10xgenomics.com/resources/datasets>.

[1] <https://pubmed-ncbi-nlm-nih-gov.ezp.sub.su.se/27365449/>

[2] <https://pubmed.ncbi.nlm.nih.gov/29360929/>